# CHiQPM: Calibrated Hierarchical Interpretable Image Classification

**Thomas Norrenbrock, Timo Kaiser & Bodo Rosenhahn**
Institute for Information Processing (tnt)
L3S - Leibniz Universität Hannover, Germany
{norrenbr,kaiser,rosenhahn}@tnt.uni-hannover.de

**Sovan Biswas & Neslihan Kose**
Intel Labs, Germany
{sovan.biswas,neslihan.kose.cihangir}@intel.com

**Ramesh Manuvinakurike**
Intel Labs, USA
ramesh.manuvinakurike@intel.com

## Abstract

Globally interpretable models are a promising approach for trustworthy AI in safety-critical domains. Alongside global explanations, detailed local explanations are a crucial complement to effectively support human experts during inference. This work proposes the Calibrated Hierarchical QPM (CHiQPM) which offers uniquely comprehensive global and local interpretability, paving the way for human-AI complementarity. CHiQPM achieves superior global interpretability by contrastively explaining the majority of classes and offers novel hierarchical explanations that are more similar to how humans reason and can be traversed to offer a built-in interpretable Conformal prediction (CP) method. Our comprehensive evaluation shows that CHiQPM achieves state-of-the-art accuracy as a point predictor, maintaining 99% accuracy of non-interpretable models. This demonstrates a substantial improvement, where interpretability is incorporated without sacrificing overall accuracy. Furthermore, its calibrated set prediction is competitively efficient to other CP methods, while providing interpretable predictions of coherent sets along its hierarchical explanation.

## 1 Introduction

Deep Learning has made remarkable advances and is being used more widely, including in high-stakes domains like medicine [1] or autonomous driving [27]. Using more transparent models, *e.g.* those that are interpretable by-design, is a promising approach to facilitate safety, robustness, and trust [53] and is even required by law for some applications [61]. For domains like autonomous driving, with no expert present during inference, models with built-in global interpretability, that can generally explain their behavior, are valuable as their reasoning can be robustly tested and verified before deployment. QPM and Q-SENN [40, 41] follow that goal by enforcing very compact class representations, that are made up of general, diverse and contrastive features, which are properties of human-friendly explanations [35]. The generality of these features is in contrast to prototypical networks like *ProtoTree* [37], which seem inherently interpretable, as they use the similarity between image patches as crucial element of their computation. However, the space in which the similarity is

39th Conference on Neural Information Processing Systems (NeurIPS 2025).

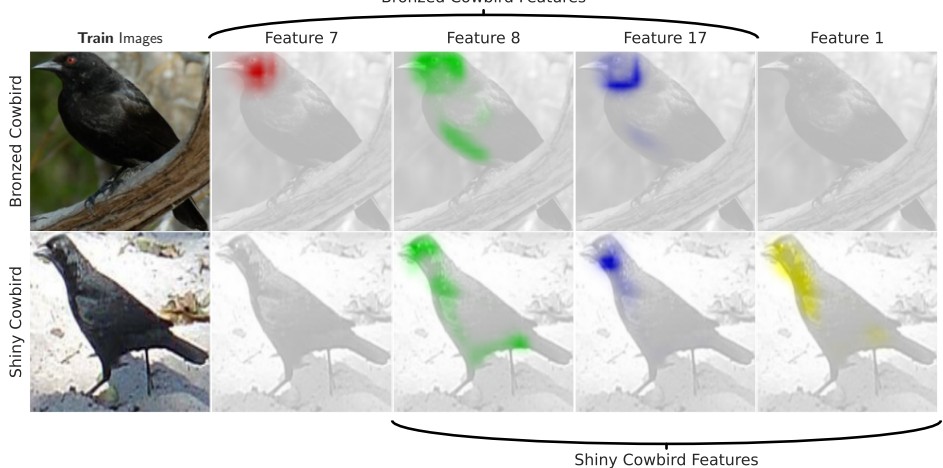

Figure 1: Contrastive global Explanation, comparing the class representations of Shiny and Bronzed Cowbirds for CHiQPM that represents every class with 3 of 30 features. The cowbirds are differentiated based on the red eye.

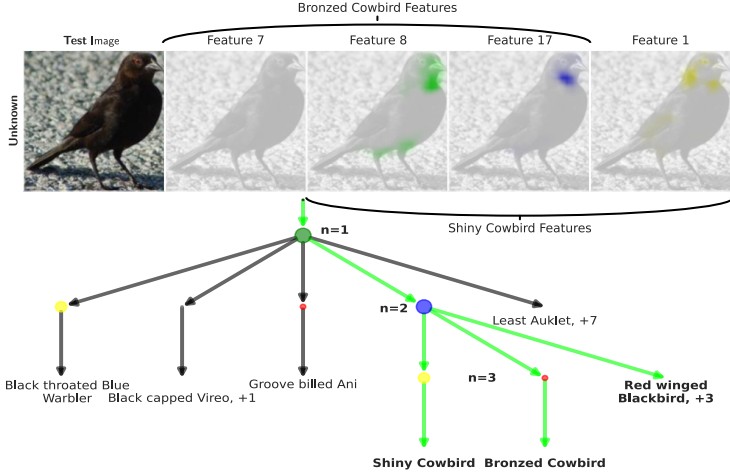

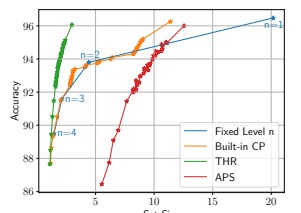

Figure 3: Coverage relative to the set size on CUB-2011 [63] for various set prediction methods applied to CHiQPM with 5 out of a total of 50 assigned features per class. The stars denote different calibration or hierarchy levels and are linearly interpolated. Traversing the hierarchical explanations (Figure 2), the built-in conformal prediction method predicts coherent sets with competitive efficiency to CP methods THR [50] or APS [46].

Figure 2: Exemplary local explanation provided by our CHiQPM, with the global explanation in Figure 1, for a difficult test image of a Bronzed Cowbird with a pale red eye that is not clearly visible. This leads to negligible activation of the red-eye detecting Feature 7. The calibrated CHiQPM provides a hierarchical explanation that communicates clear evidence for the predicted coherent set of black birds (marked in bold with green edges, including the correct label), but no sufficient evidence to differentiate between them.

computed is freely learnt which leads to class-specific prototypes [41] and similarities humans can frequently not predict [20, 24]. Due to these shortcomings, this work follows the goal of models like QPM [41] to represent classes using general features.

Considering the cognitive limitations of average humans [34], QPM represents every class with the binary assignment of very few broadly shared features, usually $\leq 5$ features. A consequence is the emergence of highly contrastive class representations. Similar to Figure 1, the difference between two classes in the model's representation space can be concretely pointed out, like differentiating the two birds via their eye, just like humans do [63]. While the contrastive representations are very helpful for QPM, they are also fairly rare, *e.g.* on average just $0.13\%$ pairs per class on CUB-2011. Other domains, *e.g.* medicine or science, can profit off of additional interpretability. When a human expert is present, they should be supported rather than replaced, a notion known as human-AI com-

plementarity. While global interpretability is still beneficial in this scenario, the value of explaining the decision for a single sample rises, known as local explanation. These typically have the form of saliency maps, such as GradCAM [54], that visualize where the explainee saw support for its decision. They can also be meaningfully computed for individual features if the globally interpretable model can be decomposed into detecting general human understandable concepts. This crucial property is a key feature of our proposed CHiQPM, as demonstrated in Figures 1 and 2. However, those heatmaps generally do not transport a notion of certainty. Therefore, predicting a set of classes with configurable guarantees on the accuracy using Conformal Prediction (CP) [62] has emerged as a promising direction for supporting experts [56, 57]. Intuitively, more classes are predicted for uncertain or less conform samples, whereas a point-predictor always predicts just one class. However, these sets typically contain a larger variety of classes, that resemble the misalignment between human and machine representations.

This work introduces the Calibrated Hierarchical QPM (CHiQPM). It improves the global interpretability of QPM while maintaining or improving the state-of-the-art accuracy by enforcing more pairs of classes with highly contrastive class representations and adapting the training pipeline to ensure class representations made of interpretable features via the proposed Feature Grounding Loss $\mathcal{L}_{\text{feat}}$ combined with ReLU activation. CHiQPM is the first model with a built-in interpretable set prediction that can be calibrated via CP, inheriting all its robust guarantees. Intuitively, CHiQPM predicts sets of classes by predicting all classes that share the dominant $n$ features with the most likely class, *e.g.* predicting all the black birds in the hierarchical explanation in Figure 2 below the blue feature at the tree level $n = 2$. Figure 3 demonstrates the already competitive efficiency of this set predictor compared to CP methods, while CHiQPM can be calibrated using CP to dynamically select the appropriate level for a concrete sample. This results in a novel way of providing hierarchical local explanations and traversing them to dynamically and understandably construct coherent prediction sets similar to how a human would reason. Considering the graph in Figure 2, the CHiQPM found the green and blue feature, that identify black birds, but no sufficient evidence to differentiate between them. Therefore, it predicts the coherent set of various black birds, including the correct class. Holistically, the novel hierarchical local explanations answer an unprecedented range of questions simultaneuosly: 1. What meaningful features of which classes are found in this image? 2. How does each feature narrow down the set of potential predictions into increasingly similar classes? 3. Which set shall be predicted to guarantee a configurable average accuracy? 4. Which features would have needed to activate stronger in order to predict a smaller set with sufficient certainty?

Our main **contributions**[1] are:

- We present the Calibrated Hierarchical QPM (CHiQPM). It is based on a heavily constrained discrete quadratic problem (QP), that selects features from a black-box model and assigns them to classes. The features of CHiQPM then adapt to the optimal solution, resulting in a globally and locally interpretable model.

- CHiQPM offers novel hierarchical local explanations and can be calibrated to reach a target coverage with competitive efficiency while ascending through its dynamically constructed interpretable class hierarchy and selecting the appropriate level. Thus, CHiQPM can be considered an interpretable conformal predictor.

- We present the Feature Grounding Loss $\mathcal{L}_{\text{feat}}$, which, alongside an additional ReLU, leads to learning more grounded and sparser features that facilitate compact hierarchical explanations along more human concepts.

- The state-of-the-art performance of CHiQPM as point- and built-in interpretable calibrated coherent set-predictor is evaluated across multiple architectures and datasets, including ImageNet-1K [48], where the gap to the black-box baseline is more than halved.

## 2 Background

### 2.1 Interpretable Machine Learning

The field of Interpretable Machine learning can be split into models with interpretability by design and methods that aim to explain models post-hoc [4, 13, 21–23, 42]. This paper introduces a model that offers interpretability by design, which is why we focus on that part. Two directions

---

[1]The code is published: `https://github.com/ThomasNorr/CHiQPM/`.

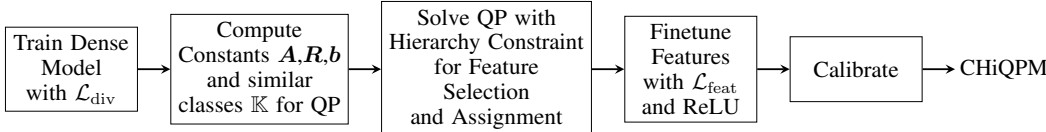

Figure 4: Overview of our proposed pipeline to obtain a CHiQPM

of interpretability can be defined: Explaining the decision for a single sample is called *local*, whereas a *global* explanation is concerned with the general behavior of a model [36]. Measuring interpretability is an unsolved task on its own, but several desired criteria have been determined. A human-friendly explanation should be diverse, general, compact and contrastive [28, 35, 45]. The SENN [2] framework, which CHiQPM can be considered as, further adds grounding as criteria, which refers to the alignability of learned representation with human concepts. One issue with measuring grounding is the prevalence of polysemantic neurons [11, 52, 60], which refers to features that detect multiple human concepts simultaneously. Nevertheless, multiple metrics have been proposed to quantify the desirable aspects [40, 41], which we evaluate in this work.

Globally interpretable models typically make their classification using a simple interpretable model applied to more interpretable features. These features are either freely learned with priors or losses to induce interpretability [38–41], supervised with human concepts in the family of Concept Bottleneck Models (CBM) [10, 25, 32, 33, 51] or restricted to closely resemble parts of the training data, so-called prototypes [6, 31, 37, 49]. Recently, the simple interpretable model is typically a very compact linear layer [38–41], but another example is a decision tree. In its most interpretable form, *Prototree* learns a static deep decision tree to classify based on similarities to learned prototypes. Like all prototype models, these similarities are typically not predictable for a human [20, 24] and the interpretability of a very deep decision tree is debatable. Apart from the static tree in *Prototree*, CHiQPM with its sample-specific low-depth hierarchical explanation is most comparable to *SLDD-Model*, *Q-SENN* and QPM [39–41]. They follow a pipeline of training a black-box model, making it compact, i.e. selecting features from the black box and sparsely connecting them to classes, and then fine-tuning the compact model to achieve a superior accuracy-compactness tradeoff. *SLDD-Model* and *Q-SENN* used *glm-saga* [65] for feature selection and their class assignment, which includes normalizing the features and is a very local optimization. QPM improves upon that by formulating the selection and assignment as quadratic problem and solving it globally optimal, improving upon accuracy and interpretability via the binary comparable class representations with a fixed number of features. The incorporated constraints on compactness enable a quantitative measurement of a metric that relates to interpretability. While prior work [15, 16, 47] primarily optimized compactness for efficiency, this and other work in interpretable machine learning [30, 38, 40, 41] focus on it for interpretability.

## 2.2 QPM

QPM [41] follows a pipeline similar to Figure 4. The first step is training a black-box dense model with an auxiliary Feature Diversity Loss $\mathcal{L}_{\text{div}}$ [39] to ensure that the initial $n_f$ feature maps activate on distinct locations of the image.

**QP** Afterwards, a quadratic problem is formulated to jointly find an optimal selection of $n_f^*$ features $s \in \{0,1\}^{n_f}$ and assignment between features and $n_c$ classes $W \in \{0,1\}^{n_c \times n_f}$, with $W^* \in \{0,1\}^{n_c \times n_f^*}$ denoting $W$ for the selected features. These variables are then optimized so that the selected features are assigned to classes they are maximally correlated to, described in the class-feature similarity matrix $A \in \mathbb{R}^{n_c \times n_f}$. Additionally, the problem formulation steers the feature selection towards distinct and local features, encoded in the feature-feature similarity matrix $R \in \mathbb{R}^{n_f \times n_f}$ or linear bias term $b \in \mathbb{R}^{n_f}$ respectively. Additionally, the quadratic problem is constrained to result in a solution with the desired level of compactness:

$$\sum_{d=1}^{n_f} s_d = n_f^* \qquad \sum_{d=1}^{n_f} w_{c,d} s_d = n_{\text{wc}} \quad \forall c \in \{1, \ldots, n_c\} \qquad (1)$$

The even sparsity can cause duplicates to arise. QPM prohibits these with an iterative optimization, that also includes an efficient incorporation of Equation (1). For more details, we refer the reader

to [41]. Finally, the features are fine-tuned with a fixed $\boldsymbol{W}^*$, so that they adapt to their assigned classes and detect shared general concepts, detailed in Appendix F.

## 2.3 Conformal Prediction

We calibrate our model using the Conformal Prediction (CP) [62] framework. For a more thorough introduction, we refer the reader to [7] which we summarize shortly. Conformal Prediction is a mathematically robust framework with minimal assumptions that can guarantee a desired error rate $\alpha$ is not exceeded. The most popular framework and the one we use is called split conformal prediction [43]. It is based on splitting off calibration data $\mathbb{D}_{\text{cal}}$ to calibrate any model and guarantees an error rate of up to $\alpha$ on data $(x_{\text{test}}, y_{\text{test}})$ that is exchangeable with $\mathbb{D}_{\text{cal}}$:

$$1 - \alpha \leq P(y_{\text{test}} \in \mathbb{Y}(x_{test})) \tag{2}$$

Here, $\mathbb{Y}(x_{test})$ describes the predicted classes for the test sample. It is constructed using a nonconformity score $s(x, c)$, that should relate to how well the predicted label $c$ fits to the input $x$. This calibration score is computed on the entire calibration set $\mathbb{D}_{\text{cal}}$, resulting in a set of scores $\mathbb{S}$ and its values are used to construct the test prediction sets:

$$\mathbb{Y}(x_{test}) = \{c \in \mathbb{C} : s(x_{test}, c) \leq \text{Quantile}(1 - \alpha, \mathbb{S})\} \tag{3}$$

Simply put, if $(1 - \alpha)$ of the true labels on the calibration data were less nonconform, i.e. had a lower nonconformity score, than a candidate class $c$ during prediction, $c$ will not be included in $\mathbb{Y}$. The nonconformity score $s(x_{test}, c)$ can be fairly arbitrarily defined and just needs to capture some notion of conformity and have sufficiently many distinct values, so that the quantile has the desired resolution. CP is typically evaluated under two aspects: a) Unconditional coverage that guarantees a minimum average accuracy on an equally distributed test set. b) Conditional coverage that aims for the desired expected accuracy for all test samples, even difficult ones. The simple method THR [50] is fairly ideal for unconditional coverage. Therefore, research has focused on conditional coverage, *e.g.* APS [46], also in difficult settings [9], or conformal training [58], which involves directly optimizing for the size of predicted sets during training. Cortes-Gomez et al. [8] even steer the prediction sets towards more coherence. To the best of our knowledge, this is the first work that proposes the notion of interpretable CP. The predicted sets are constructed by traversing hierarchical explanations, leading to coherent sets of similar classes by design, while efficiently ensuring unconditional coverage.

## 3 Method

The proposed CHiQPM is designed to classify an input image, denoted as $\boldsymbol{I} \in \mathbb{R}^{3 \times w \times h}$ into one or more of $n_c$ classes, represented as $c \in \mathbb{C} = \{c_1, c_2, ..., c_{n_c}\}$. CHiQPM consists of a deep feature extractor, $\Phi$, which processes $\boldsymbol{I}$ to generate a low-dimensional feature vector $\boldsymbol{f}^* \in \mathbb{R}^{n_f^*}$ via feature maps $\boldsymbol{M} \in \mathbb{R}^{n_f^* \times w_M \times h_M}$ of dimensions $w_M \times h_M$, and a sparse interpretable final assignment $\boldsymbol{W}^* \in \{0, 1\}^{n_c \times n_f^*}$ of these features to classes. Thus, the prediction of CHiQPM can be expressed as $\boldsymbol{y} = \boldsymbol{W}^* \boldsymbol{f}^*$, with $\hat{c} = \text{argmax}(\boldsymbol{y})$ being the predicted class. Compared to QPM, we apply an additional ReLU to the features $\mathbf{f}^* = \text{ReLU}(\hat{\mathbf{f}})$, so that CHiQPM only reasons positively, and negligible activations are suppressed. Notably, all classes in CHiQPM are represented with the same number $n_{\text{wc}}$ of features per class, which leads to easily comparable class representations. It is easiest, when two classes share exactly $n_{\text{wc}} - 1$ features, as the differentiating factor can then be concretely pointed out, as shown in Figure 1. We denote the set of these pairs with

$$\mathbb{P} = \{(i, j) : 1 \leq i < j \leq n_c \text{ and } \boldsymbol{w}_i^* \boldsymbol{w}_j^{*T} = n_{\text{wc}} - 1\} \tag{4}$$

CHiQPM follows a similar pipeline to QPM, shown in Figure 4. CHiQPM improves upon QPM (Section 2.2) with easier interpretable class representations for more of the classes via an increased $|\mathbb{P}|$, directly enforced in the QP (Section 3.1), alongside a built-in interpretable conformal prediction method traversing the novel local hierarchical explanations (Section 3.2) and an improved fine-tuning with a new Feature Grounding Loss $\mathcal{L}_{\text{feat}}$ that improves grounding and compactness of CHiQPM's explanations (Section 3.3).

## 3.1 Hierarchical Constraint

In order to ensure a higher cardinality $|\mathbb{P}|$ resulting in more easily comparable class representations, an additional constraint is added to the quadratic problem. A set $\mathbb{K}$ of pairs of classes that are highly

similar in our dense model is determined, and we ensure that most of these are part of the resulting $\mathbb{P}$ via a two-step process. Specifically, we calculate the class-class similarity $\mathbf{K} \in \mathbb{R}^{n_c \times n_c}$ based on the similarities computed for the QP $\boldsymbol{A}$ and set $\mathbb{K}$ as its $\rho \cdot n_c$ most similar class pairs:

$$\mathbf{K} = \boldsymbol{A}\boldsymbol{A}^T - \mathbf{I}_{n_c} \qquad\qquad \mathbb{K} = \{(i,j) : 1 \leq i < j \leq n_c \text{ and } \mathbf{K}_{i,j} \geq \theta\} \qquad (5)$$

Here, $\theta = \text{sort}(\mathbf{K})_{2 \cdot \rho \cdot n_c}$ describes the $(2 \cdot \rho \cdot n_c)$-th highest value in $\mathbf{K}$, $\mathbf{I}_{n_c}$ is the identity matrix and $\rho$ is a hyperparameter that controls how many classes each class should be very similar to in the representations of the resulting CHiQPM, with $\rho = 0.5$ enforcing on average one very similar class each. A higher $\rho$ increases the number of classes that can be contrastively globally explained as in Figure 1, thus improving global interpretability. Forcing similar classes to be represented very similarly by CHiQPM further induces an efficient use of the $n_f^*$ features, as shared concepts need only be detected by shared features. This improves the grounding of the hierarchical explanations and thus the efficiency of our built-in set prediction. However, with increasing $\rho$, the risk of adding classes to $\mathbb{P}$ that do not share $n_{\text{wc}} - 1$ general concepts rises. Those pairs would cause the shared features to be less robust and thus harm performance. Thus, there can be a tradeoff between point and set prediction. Therefore, $\rho$ should be set using calibration data to efficiently achieve a target coverage while balancing point prediction, which is ablated in Section 4.3 and Appendix D. After obtaining $\mathbb{K}$, the similarity of each included pair is directly added as constraints to the discrete optimization:

$$(\boldsymbol{w}_c \circ \boldsymbol{w}_{c'})^T \boldsymbol{s} = n_{\text{wc}} - 1 \quad \forall (c, c') \in \mathbb{K} \qquad (6)$$

We relax this constraint after finding the initial global solution to find the most optimal way for the CHiQPM to have sufficient, *i.e.* $|\mathbb{P}| \geq |\mathbb{K}|$, highly similar classes in its representations. This is detailed in Appendix G.

## 3.2 Set Prediction

This section describes how the proposed CHiQPM can be used to predict interpretable sets $\mathbb{Y}$ along its local hierarchical explanation. The construction of these explanations and how they can be used to predict sets at a fixed hierarchy level is first formalized. Then, we show how CHiQPM is calibrated using Conformal Prediction to predict coherent sets $\mathbb{Y}$ that contain the target label with an error rate $\alpha \in (0, 1)$ while still traversing the explaining hierarchy.

### 3.2.1 Hierarchical Explanation

Our CHiQPM enables the construction of hierarchical local explanations for a concrete test sample, as shown in Figure 2. It contains nodes for all nonzero feature activations and indicates the presence of all classes that are assigned to at least one of the shown feature-nodes. For every class $c$ with its assigned features $\mathbb{F}^c \in \{1, \ldots, n_f^*\}^{n_{\text{wc}}}$, the features are shown in the order of their activation, which can be interpreted as reasoning from the more clearly visible feature like the neck in Figure 2 to the less certain features, such as the pale red eye. The class node is then attached to its last activating feature and the class would be predicted if CHiQPM's calibration or fixed level determines that this feature and all its descendants should be predicted. To formalize predicting at a fixed level, we define the order of activations $\hat{\mathbb{F}}^c$ of the assigned features for every class $c$, where $\hat{\mathbb{F}}^c$ is ordered so that:

$$f_{\hat{\mathbb{F}}^c_i}^* \geq f_{\hat{\mathbb{F}}^c_j}^* \quad \forall \quad i, j \in \{1, \ldots, n_{\text{wc}}\} \text{ where } i < j \qquad (7)$$

Additionally, we introduce the indicator function $\delta_n^c$ at depth $n \in \{1, \ldots, n_{\text{wc}}\}$, which indicates if the class $c$ shares the same top $n$ features with the predicted class $\hat{c}$:

$$\delta_n^c = \begin{cases} 1, & \text{if} \quad \hat{\mathbb{F}}^c_i = \hat{\mathbb{F}}^{\hat{c}}_i \quad \forall i \in \{1, \ldots, n\} \\ 0, & \text{otherwise} \end{cases} \qquad (8)$$

Thus, predicting at a fixed depth of $n$ features shared with the predicted class $\hat{c}$ is:

$$\mathbb{Y}^n = \{c \in \mathbb{C} : \delta_n^c = 1\} \qquad (9)$$

For example in the graph in Figure 2, $\mathbb{Y}^1$ contains all shown class nodes, $\mathbb{Y}^2$ the five classes that share the green and blue feature and finally $\mathbb{Y}^3 = \mathbb{Y}^{n_{\text{wc}}} = \{\hat{c}\}$, as only the predicted class $\hat{c}$ shares all features with itself. Note that the stars for the fixed depth line in Figure 3 indicate predicting with a fixed $n \in \{1, \ldots, n_{\text{wc}}\}$. For more compact explanations, we restrict the graphs in the main paper to include only those classes which share the most activating feature with the predicted class. However, even for large datasets, full graphs are still informative and included in the appendix, along further visualization details in Appendix C.

### 3.2.2 Nonconformity Score

As introduced in Section 2.3, the nonconformity score $s$ is a crucial aspect of any Conformal Prediction method. In order to construct the prediction sets for CHiQPM by going up the class hierarchy, we compute the nonconformity score based on how similar the class is in the hierarchy to the initially predicted class $\hat{c}$, as we are ascending the hierarchy that led to $\hat{c}$. Note that the introduced method offers all the guarantees of CP, as only a different nonconformity score is used. For every class $c$, we propose to use the activations of the features in the shared path down the tree as nonconformity score:

$$s_{\mathrm{up}}(c) = -\sum_{i=1}^{n_{\mathrm{wc}}} \delta_i^c f_{\hat{\mathbb{F}}_i^c}^* \tag{10}$$

Note that $\delta_i^c$ ensures that all $i$ features are shared and the dependency on the input sample of all variables is omitted for brevity. We call this simple nonconformity score *up* as it goes strictly up the tree.

**Subtree Selection**  The conformal predictor can predict with more granularity and achieve its guarantees when the prediction can also go down towards only some subtrees below the feature node determined by $s_{\mathrm{up}}(c)$. That also allows CHiQPM to predict those descendants preferably that have some support beyond the shared path, *e.g.* choosing to predict only the cowbirds in Figure 2. Therefore, we extend the nonconformity score to also account for the activation of the feature at the point of diversion:

$$i^{\mathrm{div}} = \hat{\mathbb{F}}_{\mathrm{argmin}(\delta^c)}^c \qquad\qquad s_{\mathrm{sel}}(c) = s_{\mathrm{up}}(c) - f_{i^{\mathrm{div}}}^* \tag{11}$$

**Limited Level**  Finally, the maximum number of levels the set is constructed from is limited to ensure efficient sets. Towards that goal, the minimum reachable error rate $\alpha_{cal}^n$ on the calibration data for each fixed level $n$ is calculated according to Equation (9). The conformal prediction is then limited to the highest level $n^{\mathrm{limit}}$ that still reaches the target coverage defined by $\alpha$. To ensure the limitation, we limit $s_{\mathrm{up}}$ to $n^{\mathrm{limit}}$, and multiply the score with the indicator function $\delta_n^c$ indexed at $\delta_{n^{\mathrm{limit}}}^c$. This ensures all classes that were not correctly predicted under $n^{\mathrm{limit}}$ get the most nonconform score of $0$:

$$s(c) = \underbrace{\delta_{n^{\mathrm{limit}}}^c}_{\mathrm{Limitation}} \cdot \big( \underbrace{-\sum_{j=n^{\mathrm{limit}}}^{n_{\mathrm{wc}}} \delta_j^c f_{\hat{\mathbb{F}}_j^c}^*}_{\mathrm{Limited}\ s_{\mathrm{up}}} \underbrace{- f_{i^{\mathrm{div}}}^*}_{\mathrm{Subtree\ Selection}} \big) \tag{12}$$

### 3.3 Feature Grounding Loss

This section presents our novel Feature Grounding Loss $\mathcal{L}_{\mathrm{feat}}$. Its motivation is shown in the example in Figure 5. In this example, two similar classes share $2$ of their $3$ features, hence they are in $\mathbb{P}$. This is relevant, as most classes in each CHiQPM have a similar class due to the constraints described in Section 3.1. For a concrete training example, where one of these is the ground truth, Cross-Entropy loss only causes a significant gradient on the differing Ground Truth Exclusive (GTE) feature. Therefore, the distinguishing feature GTE is pushed to also activate on other concepts of the ground truth class, instead of just activating on the general concept that would differentiate them. To alleviate that issue, we propose the Feature Grounding Loss $\mathcal{L}_{\mathrm{feat}}$:

$$\mathcal{L}_{\mathrm{feat}} = -\frac{\sum_{i\in\mathbb{F}} \frac{f_i^*}{|\mathbb{F}|} - \sum_{i\in\overline{\mathbb{F}}} \frac{f_i^*}{|\overline{\mathbb{F}}|}}{\max(f^*)} \tag{13}$$

where $\mathbb{F}$ and $\overline{\mathbb{F}} \in \{1,\dots,n_f^*\}^{n_f^*-n_{\mathrm{wc}}}$ indicate the indices of the features of the ground truth class or those not assigned to it, respectively. $\mathcal{L}_{\mathrm{feat}}$ describes the difference in average activation between $\mathbb{F}$ and $\overline{\mathbb{F}}$, scaled by the maximum activation to prevent increasing activations. As shown in Figure 5, $\Delta$ has the same gradient for all positively assigned features and also has a nonzero gradient for all features in $\overline{\mathbb{F}}$. Therefore, it encourages all the features of the ground truth class to detect more general concepts while also inducing sparsity in the feature activations, enabling the visualization of the entire hierarchical explanation. We add $\mathcal{L}_{\mathrm{feat}}$ to our overall training loss during fine-tuning with weighting $\lambda_{\mathrm{feat}}$.

Table 1: Comparison on Accuracy, Compactness, Contrastiveness and Structural Grounding. Compact describes the number of features $n_f^*$ and features per class $n_{wc}$. It is binned into very compact + ($n_{wc} = 5$ and $n_f^* = 50$), medium ○, and the baseline, denoted - ($n_{wc} = 2048$ and $n_f^* = 2048$), exact figures in Table 18. Figure 9 shows the increasing gap when further raising compactness. Among more compact (○ or above) models, the best result is marked in bold, second best underlined.

| Method | Accuracy ↑ | | | Com-pact | Contrastiveness ↑ | | | SG ↑ |
|---|---|---|---|---|---|---|---|---|
| | CUB | CAR | IN | | CUB | CAR | IN | CUB |
| Dense Resnet50 | 86.6 | 92.1 | 76.1 | - | 74.4 | 75.1 | 71.6 | 34.0 |
| glm-saga$_5$ | 78.0 | 86.8 | 58.0 | ○ | 74.0 | 74.5 | 71.7 | 2.5 |
| PIP-Net | 82.0 | 86.5 | - | ○ | 99.5 | 99.5 | - | 6.7 |
| ProtoPool | 79.4 | 87.5 | - | ○ | 76.7 | 78.9 | - | 13.9 |
| SLDD-Model | 84.5 | 91.1 | 72.7 | + | 87.2 | 89.7 | 93.4 | 29.2 |
| Q-SENN | 84.7 | 91.5 | 74.3 | + | 93.0 | 94.2 | 92.6 | 23.4 |
| QPM | 85.1 | 91.8 | 74.2 | + | 96.0 | 97.7 | 89.3 | 47.9 |
| CHiQPM (**Ours**) | **85.3** | **91.9** | **75.3** | + | **99.9** | **100** | **99.9** | **75.0** |

## 4 Experiments

Following QPM (Section 2.2), we evaluate our method on CUB-2011, Stanford Cars [26] and ImageNet-1K. CUB-2011 and Stanford Cars are the most commonly used datasets for interpretability, while ImageNet-1K is suitable to demonstrate how the method scales to larger problems with more real-world applications. CUB-2011 includes human annotations of relevant concepts for every image, which makes it suitable for evaluating the alignment between human representations and the ones learned. As the proposed method can be applied to any backbone, we show results on Resnet50 [18], Resnet34, Inception-v3 [59] and Swin-Transformer-Small [29] to allow an easy comparison with previous work. The main paper focuses on results for Resnet50, always reporting the mean across 5 random seeds, with 3 for ImageNet-1K. More results are

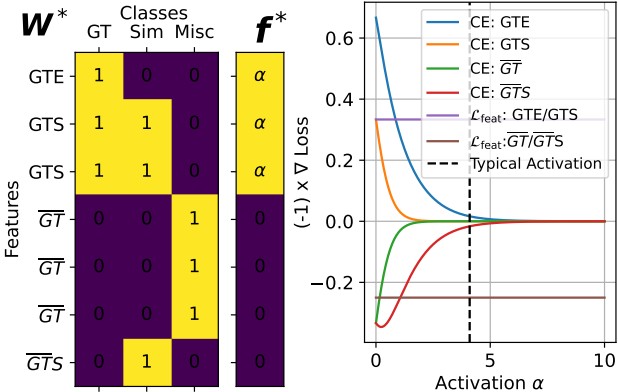

Figure 5: Gradient on features $f^*$ for a train sample labeled $GT$ for a toy example with 3 classes and 7 features, with $W^*$ shown left. At the average activation on the CUB dataset, the Ground Truth Exclusive (GTE) feature has a roughly 4000 times higher gradient than the other assigned features, which are shared with *Sim*, Ground Truth Shared (GTS).

included in Tables 7 to 9 and 18 to 25, demonstrating the robustness of our method across random seeds and architectures. For implementation details, we generally followed QPM [41], but report details and slight deviations in Appendix A. Specifically, all models that follow a pipeline similar to ours, *SLDD-Model*, Q-SENN and QPM, use the same parameters for the dense training. As usual in literature, $n_{wc} = 5$ and $n_f^* = 50$ are set if not reported otherwise. Further, we generally set the density parameter for our class hierarchy to $\rho = 0.5$, as it is sufficient to demonstrate the improvements in built-in set prediction without sacrificing accuracy as point-predictor. Finally, $\lambda_{feat} = 3$ is set as higher values cause reduced accuracy.

### 4.1 Metrics

CHiQPM is designed to improve upon QPM (Section 2.2), primarily via more easily interpretable class representations, being able to produce meaningful hierarchical explanations and by offering the built-in interpretable calibrated set prediction. Therefore, we evaluate CHiQPM across all QPM

Table 2: Average Set Size $|\mathbb{Y}|$ of CHiQPM calibrated to reach various coverages $1 - \alpha$ comparing different conformal prediction methods. All methods are very close or reach the desired coverage.

| $|\mathbb{Y}|\downarrow$ | Inter- | CUB | | | | CARS | | | INET | | | |
|---|---|---|---|---|---|---|---|---|---|---|---|---|
| Method | pretable | $\alpha$=0.12 | $\alpha$=0.1 | $\alpha$=0.075 | $\alpha$=0.05 | $\alpha$=0.075 | $\alpha$=0.05 | $\alpha$=0.0025 | $\alpha$=0.22 | $\alpha$=0.2 | $\alpha$=0.175 | $\alpha$=0.15 |
| **Ours** | ✓ | 1.22 | 1.73 | 2.94 | 9.05 | 1.05 | 1.25 | 8.25 | 1.10 | 1.42 | 3.25 | 4.58 |
| $s = s_{\text{sel}}$ | ✓ | 4.62 | 6.15 | 9.53 | 29.4 | 3.62 | 5.95 | 28.4 | 8.14 | 11.3 | 17.9 | 30.5 |
| $s = s_{\text{up}}$ | ✓ | 3.03 | 3.91 | 8.87 | 18.7 | 2.32 | 3.27 | 17.9 | 4.36 | 6.23 | 11.8 | 31.4 |
| THR | ✗ | 1.16 | 1.32 | 1.67 | 2.41 | 1.02 | 1.15 | 2.09 | 1.05 | 1.16 | 1.40 | 1.87 |
| APS | ✗ | 6.30 | 7.20 | 8.54 | 11.3 | 5.64 | 6.83 | 9.61 | 16.7 | 18.9 | 22.1 | 26.8 |

metrics in addition to the accuracy as point predictor in relation to its compactness. For evaluating the performance as set predictor, we report the size of the predicted sets, when calibrated to reach a specific accuracy or coverage. Finally, the annotations in CUB-2011 are used to compute the Set Coherence of the predicted sets and also measure the grounding of the features via the Alignment [40] metric, alongside QPM's Structural Grounding. Following QPM, we compute the ground truth class-class similarity matrix $\Psi^{gt} = \Lambda\Lambda^T$ using the annotated average class attributes $\Lambda \in [0,1]^{n_c \times 312}$ with columns $\lambda_c \in [0,1]^{312}$, where $\lambda_{c,j}$ indicates the fraction of images with label $c$, in which attribute $j$ is annotated to be present. $\Psi^{gt}$ enables quantifying the similarities of classes that are predicted together, the novel Set Coherence sc:

$$\mathbb{K}(\mathbb{Y}(x_{\text{test}})) = \{\psi^{gt}_{c,c'} \mid c, c' \in \mathbb{Y}(x_{\text{test}}), c < c'\} \quad \text{sc} \quad = \text{mean}( \bigcup_{x_{\text{test}} \in \mathbb{D}_{\text{test}}} \mathbb{K}(\mathbb{Y}(x_{\text{test}}))) \quad (14)$$

It is defined as the mean over all the class similarities of classes that the predictor predicts jointly. Thus, a higher value indicates sets with more similar classes in reality.

For brevity, we focus on a subset of more strongly affected metrics in the main paper but report results on all of them in the appendix alongside every metrics formulation in B.

## 4.2 Results

This section discusses the main quantitative results of our proposed method. Further qualitative examples are included in the appendix. Figures 14 to 20 showcase global explanations, Figures 21 to 30 include the novel local hierarchical explanations and Figures 31 to 34 demonstrate how the features of CHiQPM are general concept detectors. The accuracy as point predictor along the generally preferable qualities of Compactness, Contrastiveness and Structural Grounding is shown in Table 1. CHiQPM shows state-of-the-art accuracy for compact point predictors. Further, it scores nearly perfectly on Contrastiveness. CHiQPM learns features that can be more clearly separated between active and inactive than even the class detectors of *PIP-Net* [38], indicating a gap between the ReLU-induced minimum of $0$ and the activations where a relevant concept is found. The clear distinction between active and inactive enables our saliency maps, like in Figures 1 and 2, to also transport *activation* rather than just *location* without a reference test image and therefore enables extensive local explanations in practice. The details are explained in Appendix C.2. Finally, Structural Grounding quantifies that the additionally added pairs via Equation (31) are also similar in reality and thus lead to more grounded class representations. The state-of-the-art accuracy as point predictor paves the way for accurate set prediction along the hierarchical explanation, as the sets are conditioned on the predicted class.

For calibrating Conformal Prediction methods, the first 10 test examples per class are split off into the calibration data $\mathbb{D}_{\text{cal}}$. That way, we can use the same models for evaluating point and set prediction. Notably, applying Split Conformal Predictions requires exchangeability between calibration and test data. Our experimental setup is designed to ensure this exchangeability, as detailed in Appendix L. As comparable CP methods, THR [50] and APS [46] are used, as they are applicable without hyperparameters and broadly used [7]. Table 2 compares our built-in CP method with these and also with the two simpler nonconformity scores $s_{\text{sel}}$ and $s_{\text{up}}$. Evidently, our proposed nonconformity score that restricts the sets to be constructed by going up the hierarchical local explanations shows competitive efficiency to THR for higher error rate $\alpha$ and approaches APS for lower values. The reason can be seen when comparing our approach with the simpler $s_{\text{sel}}$ that does not restrict the tree level to $n^{\text{limit}}$: With lower $\alpha$, the gap decreases, as $n^{\text{limit}}$ has to be set more loosely, allowing larger and therefore inefficient sets, which can be gauged from Figure 3. Figure 6 visualizes the Set Coherence (Equation (14)) of multiple models and CP methods and also indicates the set size for a

Table 3: Impact of $\mathcal{L}_{\text{feat}}$ on CUB-2011. It effectively increases the Feature Alignment while reducing the fraction of nonzero features on the test data.

| Method | Acc ↑ | Feature↑ Alignment | Feature ↓ Sparsity |
|---|---|---|---|
| CHiQPM (**Ours**) | **85.3** | **3.8** | **22.3** |
| w/o $\mathcal{L}_{\text{feat}}$ | **85.3** | 3.0 | 39.9 |
| and L1-Regularization | **85.3** | 3.4 | 31.3 |
| and L1 for $\overline{\mathbb{F}}$ | 85.1 | 3.6 | 26.4 |

desired $\alpha$. As expected, CHiQPM predicts the most coherent sets, as it moves up the hierarchical local explanation constructed from class representations with Structural Grounding. CHiQPM further clearly surpasses QPM in efficiency and Set Coherence, making it the only model with efficient, calibrated, built-in, and interpretable coherent set prediction.

### 4.3 Ablation Studies

This section investigates the impact of $\rho$, $\mathcal{L}_{\text{feat}}$ and summarizes the ablations shown in the appendix. Figure 6 demonstrates how a higher $\rho$ leads to an increased $|\mathbb{P}|$, more Set Coherence and efficient sets. For example, $\alpha = 0.075$ can be reached with 6.3 classes per prediction with $\rho = 0.1$ or 2.9 with $\rho = 0.5$, as $n^{\text{limit}}$ can be higher. Note that $\rho$ can be optimized on calibration data, steering the CHiQPM towards efficiently reaching a desired $\alpha$, demonstrated in Appendix D. Further visualizations that explicitly relate $\rho$ to the efficiency are included in the appendix, Figures 7 and 8. The impact of $\mathcal{L}_{\text{feat}}$ is shown in Table 3 and also contrasted with a $\mathcal{L}1$-regularization on the features $\mathbf{f}^*$ as another form to induce sparse features. The degree of regularization was chosen so that a further increase greatly reduces the accuracy. Without sacri-

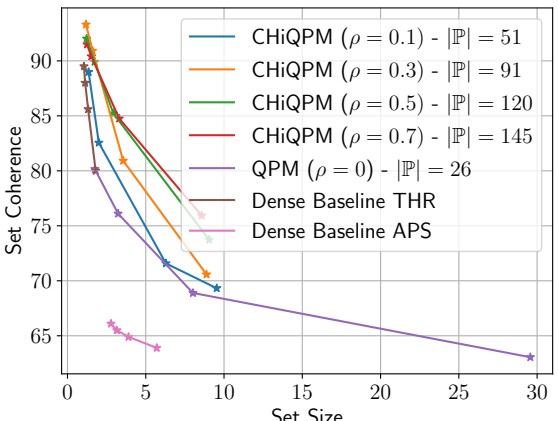

Figure 6: Set Coherence on CUB-2011 for all classes predicted together. Across the set sizes, the built-in set prediction produces the most coherent sets. The stars denote different $\alpha \in \{0.12, 0.1, 0.075, 0.05\}$ used for calibration. Note that an increased set size corresponds to a lower $\alpha$. Therefore, the competitive efficiency is also visible.

ficing any accuracy, the proposed CHiQPM reaches the highest Feature Alignment and best sparsity and thus validates $\mathcal{L}_{\text{feat}}$ as a suitable formulation to induce compact grounded hierarchical local explanations, with Appendix H visualizing the impact of $\lambda_{\text{feat}}$. As further validation for our method, Appendix J shows the benefit of using a sample-specific hierachy instead of a dynamic one. Additionally, Appendix K quantifies that CHiQPM's features quantifiably localize more on relevant regions in the image. Finally, further ablations show a larger gap between CHiQPM and QPM at even higher compactness (Figure 9) and the strong positive impact of ReLU (Appendix I).

## 5 Conclusion

This work introduces the Calibrated Hierarchical QPM (CHiQPM). Faithfully following its grounded globally interpretable class representations, CHiQPM provides hierarchical local explanations. CHiQPM is calibrated as a form of built-in interpretable Conformal Prediction to traverse the hierarchy at test time and predict a set of coherent classes, similar to how a human reasons, which can be a step towards human-AI complementarity. Finally, CHiQPM's improved global and additional novel form of local interpretability come with state-of-the-art accuracy as compact point predictor and efficiency on par with non-coherent set predictors even on ImageNet-1K, ensuring broad applicability.

**Acknowledgments**

This work was supported by the Federal Ministry of Education and Research (BMBF), Germany, under the AI service center KISSKI (grant no. 01IS22093C), the Deutsche Forschungsgemeinschaft (DFG) under Germany's Excellence Strategy within the Cluster of Excellence PhoenixD (EXC2122), the MWK of Lower Sachsony within Hybrint (VWZN4219), the European Union under grant agreement no. 101136006 – XTREME. The work has been done in collaboration and partially funded by the Intel Corporation. This work was partially supported by the German Federal Ministry of the Environment, Nature Conservation, Nuclear Safety and Consumer Protection (GreenAutoML4FAS project no. 67KI32007A).

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

# A    Implementation Details

For implementing CP, we utilized the *torchcp* [64] package and implemented the model using *Pytorch* [44]. Note that all details will also be clear in the published code.

The QP is solved as described, with the additional constraints from Section 3.1, but with two relaxations based on observations: First, the MIP-Gap of the discrete optimization in *Gurobi* [17] can be relaxed without significant effect on the resulting metrics, hence we set it to $1\%$. Second, due to the high number of classes, $n_c = 1000$, the resulting assignment for ImageNet-1K contains more pairs $|\mathbb{P}|$ than desired anyway. Therefore, we do not add any pairwise constraints to the QP for ImageNet-1K. After solving the QP, we further fine-tune the model for 70 epochs using the learning rate schedule from Q-SENN, as we observed significantly larger changes in the features during the fine-tuning as opposed to QPM, as quantified in Appendix F.1. The longer training with a higher learning rate facilitates that the features can be reorganized according to the imposed hierarchical structure. We set $\lambda_{\text{feat}} = 3$, but skip this loss during the final 10 epochs, as its desired effects are induced without reducing top-1 Accuracy. All other steps of the pipeline equal QPM [41]. By mistake, the subtrahend in Equation (13) was scaled by a factor of 2 in all experiments. We repeated the experiments in Appendix H with the correct implementation and observed the expected results with negligible impact of the mistake. Therefore, we did not rerun every experiment.

Note that for accuracy as point-predictor and for $\hat{c}$ for conformal prediction, the argmax of $\boldsymbol{y}$ is considered the predicted class, leading to the lower index in case of a tie.

In order to apply our novel CP to QPM, we subtract the minimum of all features of the features, ensuring that reasoning is restricted down the hierarchical explanation and unaffected by slightly varying $\boldsymbol{\mu}^{n_f^*}$.

# B    Metrics

This section defines the metrics that are not introduced in this work for the first time.

Following QPM [41], we measure the general qualities of learning diverse, general and contrastive features via SID@5,

$$\hat{M}_{i,j}^d = \frac{M_{i,j}^d}{\frac{1}{w_M h_M} \sum |\mathbf{M}^d|} \quad \hat{S}_{i,j}^d = \frac{e^{\hat{M}_{i,j}^d}}{\sum_{m,n} e^{\hat{M}_{m,n}^d}} \tag{15}$$

$$\text{SID@5} = \frac{\sum_{i=1}^{h_M} \sum_{j=1}^{w_M} \max(\hat{S}_{i,j}^1, \hat{S}_{i,j}^2, \ldots, \hat{S}_{i,j}^5)}{5}, \tag{16}$$

Class-Independence $\tau$,

$$\tau = 1 - \frac{1}{n_f^*} \sum_{d=1}^{n_f^*} \max_c \frac{\sum_{j=1}^{n_T} l_j^c (f_{j,d} - \min \boldsymbol{f}_{:,d})}{\sum_{j=1}^{n_T} (f_{j,d} - \min \boldsymbol{f}_{:,d})} \tag{17}$$

and Contrastiveness:

$$\text{Contrastiveness} = \sum_{d=1}^{n_f^*} 1 - \text{Overlap}(\mathcal{N}_1^d, \mathcal{N}_2^d) \tag{18}$$

SID@5 describes how diversely the top 5 weighted features activate for an average test sample. Class-Independence measures which fraction of each features' activation does not activate on its most related class and a low Class-Independence indicates that features are class-detectors. Finally, Contrastiveness measures the overlap between two Gaussian distributions, parametrized by $\mu_1^d, \sigma_1^d, \mu_2^d, \sigma_2^d$, fit into the feature distribution $\boldsymbol{f}_{:,d}$. A high Contrastiveness indicates features that can be split into activating and not activating.

Using the computed $\psi_{c,c'}^{gt}$, Structural Grounding [41] quantifies how similar the top 25 most similar classes in reality, $\boldsymbol{\Psi}^{gt} = \boldsymbol{\Lambda}\boldsymbol{\Lambda}^T$, are represented in the class representations $\boldsymbol{\Psi}^{Model} = \boldsymbol{W}\boldsymbol{W}^T$:

$$\text{StructuralGrounding} = \frac{\sum_{c,c' \in C_{\text{Sim}}} \psi_{c,c'}^{Model}}{\sum_{c,c' \in C_{\text{Sim}}} \psi_{c,c'}^{gt}}. \tag{19}$$

The Feature Alignment $r$ [40] between the learned features and these attributes is also measured to estimate if the features $\boldsymbol{F} \in \mathbb{R}^{n_T \times n_f}$s are more human-like:

$$A_{a,j}^{\text{gt}} = \frac{1}{|\rho_{a+}|} \sum_{i \in \rho_{a+}} F_{i,j}^{\text{train}} - \frac{1}{|\rho_{a-}|} \sum_{i \in \rho_{a-}} F_{i,j}^{\text{train}} \tag{20}$$

$$r = \frac{1}{n_f^*} \sum_{j=1}^{n_f^*} \frac{n_T}{\sum_{l=1}^{n_T} F_{l,j}^{\text{train}} - \min_l F_{l,j}^{\text{train}}} \max_i A_{i,j}^{\text{gt}} \quad . \tag{21}$$

## C  Visualization Details

### C.1  Graph Visualization

This section discusses the details of how the hierarchical explanations in the form of graphs are visualized, *e.g.* for Figure 2, and also the global hierarchical explanations in Figures 23 and 24. Generally, the content of the graphs is explained in Section 3.2.1. To indicate the activation of each feature $i$, the radius of each feature node scales with its fraction of the maximum activation $r_i = \frac{f_i^*}{\max(f^*)} r_{\text{scale}}$, where $r_{\text{scale}}$ is a constant.

Using colors, we add additional information to the graphs. The nodes are colored according to the saliency maps, that are used in the global explanation and also as part of the local explanation. All nodes, that correspond to features not visualized as saliency map, are represented in gray. Further, we use color to indicate the actual prediction done. Only those classes are predicted by CHiQPM, for which a colored path coming from the root of the graph connects to the class nodes.

Finally, the class labels at each node are summarised if multiple classes are at the same node, i.e. sharing the same top-$n$ features, but not having any activation on their other assigned features. To summarise the class names, the text description "*ClassName, + x*" is used. Here *ClassName* is chosen from the classes at that node, usually the name with the shortest words to reduce horizontal overlap, and *x* refers to the number of classes not called *ClassName* at this node.

### C.2  Visualizing saliency maps

This section explains how the saliency maps for the individual features as part of global and local explanation are visualized.

Typical saliency map methods and available implementations [14], like GradCAM [54], only focus on showing where support for a decision is found. As they are used to showing why a decision was made, they do not have to transport activation. Therefore, they overlay the computed saliency map $\boldsymbol{S} \in \mathbb{R}^{w \times h}$ on the image $\boldsymbol{I}$ scaled to the full range of values irrespective of activation to get the Saliency explanation $\boldsymbol{I}^s \in \mathbb{R}^{3 \times w \times h}$:

$$\boldsymbol{I}^{\text{NoActivation}} = \text{ColorMap}(\frac{\boldsymbol{S}}{\max(\boldsymbol{S})})\beta + (1 - \beta)\boldsymbol{I} \tag{22}$$

$$\boldsymbol{I}^s = \frac{\boldsymbol{I}^{\text{NoActivation}}}{\max(\boldsymbol{I}^{\text{NoActivation}})} \tag{23}$$

Here, ColorMap converts the spatial information into RGB values and $\beta$ defines a weighting between image and saliency map. While typically *jet* is used, we use individual colors in this work, scaled using gamma correction to match the impression of *jet*, as we decompose the decision process into detecting individual features or concepts. Notably, different saliency methods differ in how they compute $\boldsymbol{S}$.

Similar to GradCAM, we use the feature maps $\boldsymbol{M} \in \mathbb{R}^{n_f^* \times w_M \times h_M}$ as the basis for our visualization, as they directly cause the activations our model uses and are also evaluated and steered to be diverse. However, we visualize the map $\boldsymbol{M}^d \in \mathbb{R}^{w_M \times h_M}$ for every feature $d$ individually rather than a weighted sum because every feature can and should be interpreted on its own.

Different to prior work, CHiQPM has near perfect contrastive features, as measured by Contrastiveness. That means that for a concrete sample, the feature $d$ can be clearly sorted into the inactive

distribution $\mathcal{N}_1^d$ or the active distribution $\mathcal{N}_2^d$. We make use of that distinction and especially the mean activation of active samples $\mu_2^d$ to produce saliency maps that indeed already transport whether the feature is activated or not. Specifically, the Saliency explanation for a single feature $d$ is computed following:

$$\boldsymbol{I}_d^{\text{Activation}} = \text{ColorMap}(\frac{\boldsymbol{M}^d}{\max(\boldsymbol{M}^d)} min(\frac{f_d^*}{\mu_2^d}, 1))\beta + (1-\beta)\boldsymbol{I} \tag{24}$$

$$\boldsymbol{I}_d^s = \frac{\boldsymbol{I}_d^{\text{Activation}}}{\max(\boldsymbol{I}_d^{\text{Activation}})} \tag{25}$$

Here, the term $min(\frac{f_d^*}{\mu_2^d}, 1)$ scales the saliency map based on how active the feature is. Due to the perfect Contrastiveness, the threshold $\mu_2^d$ is a good indicator how objectively active the feature is. Notably, this form of transporting confidence can be applied to a single test image, as the active threshold $\mu_2^d$ is computed on training data. This is in contrast to how QPM [41] scaled activations for the same features across images.

In our formulas, we omit how all feature maps are linearly scaled from their feature map size to the input image size.

## D   Impact of Tree Density $\rho$

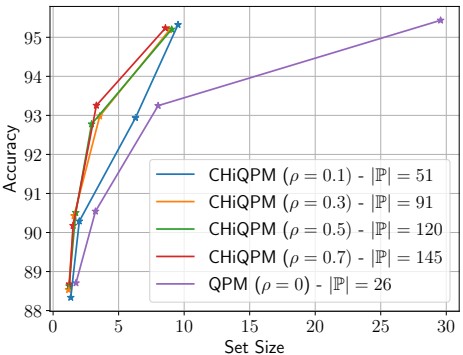

Figure 7: Accuracy over set size using the built-in conformal prediction for varying $\rho$. The stars denote different $\alpha \in \{0.12, 0.1, 0.075, 0.05\}$ used for calibration. Better fixed level performance, shown in Figure 8, leads to more efficient calibrated sets due to being able to limit the depth via $n^{\text{limit}}$.

This section discusses the impact of $\rho$ and provides the detailed relation between $\rho$ and the efficiency of the hierarchical and calibrated predictions. As noted in Section 4.3, Figure 6 demonstrates how a higher $\rho$ leads to an increased $|\mathbb{P}|$, more Set Coherence and efficient sets. Figures 7 and 8 explicitly relate $\rho$ to the efficiency of the resulting predictor and visualize the relationship between fixed level efficiency in Figure 8 and calibrated efficiency in Figure 7. As shown in Figure 8, in order to reach $\alpha = 0.075$, CHiQPM can typically be limited to $n^{\text{limit}} = 2$. However, for $\rho = 0.1$, it does not reach 92.5% accuracy with a fixed $n = 2$ and therefore has to be calibrated with $n^{\text{limit}} = 1$. This causes the significant increase in Set Size and therefore inefficiency for $\alpha = 0.075$, from 2.9 with the default configuration of $\rho = 0.5$ to 6.3 with $\rho = 0.1$, as clearly visualized in Figure 7. Therefore, optimizing $\rho$ using calibration data for a desired $\alpha$ is an effective option. However, note that the point-predicting performance starts to deteriorate slightly when further increasing $\rho$ beyond $\rho = 0.5$, as indicated by the start of the graphs in Figure 8. This is likely due to classes being forced to be represented very similarly, even though they do not share $n_{\text{wc}} - 1$ general concepts. The $|\mathbb{P}|$ is further evidence for that, as the number of unenforced pairs, $|\mathbb{P}| - \rho * n_c$, declines, with just 5 for $\rho = 0.7$. Thus, the optimal value for $\rho$ also depends on the similarities of the classes in the dataset.

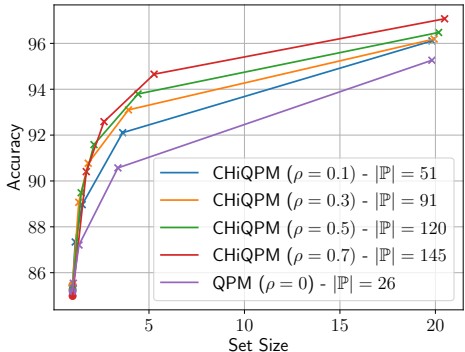

Figure 8: Set Accuracy of CHiQPM with $n_{\mathrm{wc}} = 5$ on CUB, when predicting with a fixed level in the sample specific hierarchy. Each mark represents one level.

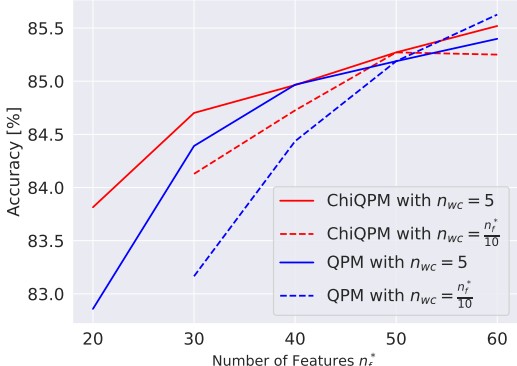

Figure 9: Accuracy on CUB in relation to Compactness: CHiQPM uses the allowed low compactnesss more efficiently, both when scaling $n_{\mathrm{wc}}$ with $n_f^*$ and when not. When $n_f^*$ becomes higher, restricting $n_{\mathrm{wc}} = \frac{n_f^*}{10}$ leads to a less efficient use of some features.

# E   Detailed Results

Table 4: Impact of Relu and Normalization on CUB-2011 with Resnet50

| Method | Acc | C-I | Contrast. | Feature Alignment |
|---|---|---|---|---|
| CHiQPM (**Ours**) | **85.3** | 94.1 | **99.9** | **3.8** |
| w/o $\mathcal{L}_{\mathrm{feat}}$ and ReLU | 84.8 | **96.7** | 97.1 | 1.9 |
| w/o Norm | 83.9 | 95.1 | 96.2 | 3.2 |

This section contains more results to demonstrate the quantifiable high performance of our CHiQPM.

First, Figure 11 visualizes how QPM performs with different CP methods. When comparing it to Figure 3, CHiQPM's drastic improvement is evident, making interpretable coherent set prediction competitively efficient. Further, Figure 12 demonstrates how the built-in interpretable coherent set prediction predicts more coherent sets than conventional CP methods, even for the same CHiQPM. However, it is notable, that THR predicts significantly more coherent sets in that scenario, as it is applied to our CHiQPM with Structural Grounding.

Finally, we include the results on Resnet50, Resnet34, Swin-Transformer-small [29] and Inception-v3 in Tables 18 to 25 with their standard deviations, computed as *np.std(x)*, where x are the individual results across the different seeds. The CP efficiency across architectures with standard deviations is shown in Tables 5 to 9. These tables demonstrate how our state-of-the-art performance is architecture independent. Note that we recreated the Q-SENN results on CUB-2011 and Stanford Cars for these tables in order to get standard deviations and results on more architectures but included the reported results in Norrenbrock et al. [41] in the main paper. Notably SID@5 has slightly decreased due to the introduction of $\mathcal{L}_{\text{feat}}$. Usually, SID@5 is high due to the $\mathcal{L}_{\text{div}}$. With the addition of $\mathcal{L}_{\text{feat}}$, especially on ImageNet-1K, its relative weighting is slightly lower. Therefore, one might want to increase the weighting for it for ImageNet-1K. For easier comparison and compute limitations, we did not optimize the weighting of $\mathcal{L}_{\text{div}}$ at all and left it at the weight used by *SLDD-Model* [39], *Q-SENN* [40] and QPM [41]. However, initial experiments show that one can increase it without significant impact on accuracy, shown in Table 10. The strong performance of CHiQPM across all the criteria is also summarized in the radar plot in Figure 10, which follows QPM and thus only includes its metrics. Even ignoring the excellent Feature Alignment, Sparsity and novel hierarchical explanations with calibrated coherent set predictions, CHiQPM clearly sets the state of the art.

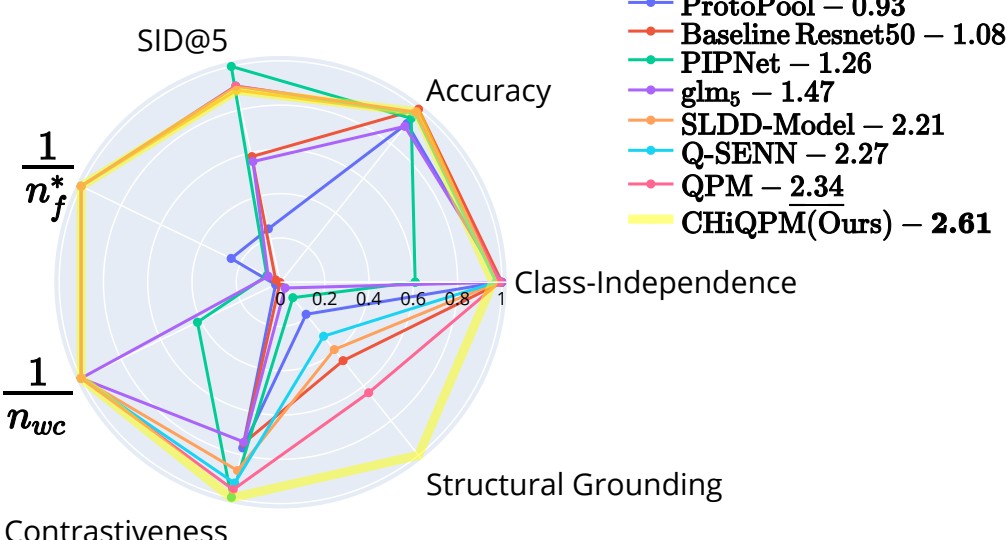

Figure 10: Radar plot across the QPM metrics for CUB-2011 on Resnet50. Every value is transformed to a fraction of the maximum, following QPM [41].

Table 5: Average Set Size $|\mathbb{Y}|$ of CHiQPM with Resnet50 calibrated to reach various coverages $1 - \alpha$ comparing different conformal prediction methods. All methods are very close or reach the desired coverage.

| $|\mathbb{Y}|\downarrow$ | CUB | | | | CARS | | | |
|---|---|---|---|---|---|---|---|---|
| Method | $\alpha$=0.12 | $\alpha$=0.1 | $\alpha$=0.075 | $\alpha$=0.05 | $\alpha$=0.075 | $\alpha$=0.05 | $\alpha$=0.0025 | $\alpha$=0.001 |
| **Ours** | $1.22 \pm 0.04$ | $1.73 \pm 0.08$ | $2.94 \pm 0.17$ | $9.05 \pm 0.20$ | $1.05 \pm 0.02$ | $1.25 \pm 0.14$ | $8.25 \pm 0.38$ | $74.4 \pm 1.83$ |
| THR | $1.16 \pm 0.04$ | $1.32 \pm 0.05$ | $1.67 \pm 0.08$ | $2.41 \pm 0.12$ | $1.02 \pm 0.01$ | $1.15 \pm 0.02$ | $2.09 \pm 0.16$ | $6.93 \pm 1.49$ |
| APS | $6.30 \pm 0.41$ | $7.20 \pm 0.65$ | $8.54 \pm 0.50$ | $11.3 \pm 0.96$ | $5.64 \pm 0.30$ | $6.83 \pm 0.55$ | $9.61 \pm 0.55$ | $18.6 \pm 1.94$ |

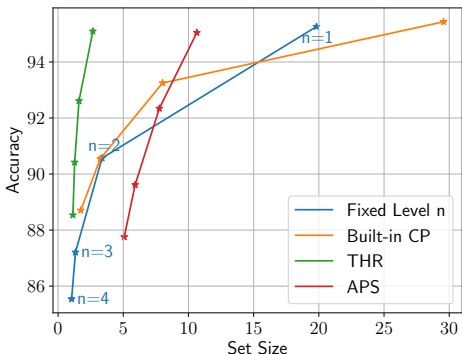

Figure 11: Results for QPM on CUB-2011 for different CP methods applied to it, comparable to Figure 3. CHiQPM significantly boots the efficiency of the built-in CP.

Table 6: Average Set Size $|\mathbb{Y}|$ of CHiQPM with Resnet50 on ImageNet-1K calibrated to reach various coverages $1 - \alpha$ comparing different conformal prediction methods. All methods are very close or reach the desired coverage.

| $|\mathbb{Y}|\downarrow$ | IMG | | | |
|---|---|---|---|---|
| Method | $\alpha$=0.22 | $\alpha$=0.2 | $\alpha$=0.175 | $\alpha$=0.15 |
| **Ours** | $1.10 \pm 0.01$ | $1.42 \pm 0.02$ | $3.25 \pm 0.07$ | $4.58 \pm 0.11$ |
| THR | $1.05 \pm 0.00$ | $1.16 \pm 0.01$ | $1.40 \pm 0.01$ | $1.87 \pm 0.03$ |
| APS | $16.7 \pm 0.33$ | $18.9 \pm 0.28$ | $22.1 \pm 0.49$ | $26.8 \pm 0.504$ |

## F  Fine-tuning

**Fine-tuning**  After solving the quadratic problem, the feature selection and binary assignment are set as fixed parameters of our model. Following Q-SENN, SLDD-Model and QPM, we continue training the model, so that the features adapt to the concept(s) that is or are shared by its assigned classes and also follow their normalization. Specifically, the mean $\boldsymbol{\mu}^{n_f^*} \in \mathbb{R}^{n_f^*}$ and standard deviation $\boldsymbol{\sigma}^{n_f^*} \in \mathbb{R}^{n_f^*}$ are computed on the the training set:

$$f_{j,d} = \frac{1}{w_M h_M} \sum_{l=1}^{w_M} \sum_{k=1}^{h_M} m_{d,l,k}^j \tag{26}$$

$$\boldsymbol{\mu}_d^{n_f^*} = \frac{\sum_{j=1}^{n_T} f_{j,d}}{n_T} \tag{27}$$

$$\boldsymbol{\sigma}_d^{n_f^*} = \sqrt{\frac{\sum_{j=1}^{n_T} (f_{j,d} - \boldsymbol{\mu}_d^{n_f^*})^2}{n_T}}, \tag{28}$$

where $m_{d,l,k}^j$ refers to the value of the feature maps at the spatial position $l, k$ for feature $d$ and sample $j$. This normalization for every feature $d$ is maintained for the fine-tuning:

$$\hat{f}_d = \frac{1}{w_M h_M} \frac{\sum_{l=1}^{w_M} \sum_{k=1}^{h_M} m_{d,l,k} - \boldsymbol{\mu}_d^{n_f^*}}{\boldsymbol{\sigma}_d^{n_f^*}} \tag{29}$$

While this was initially done due to the reliance of Q-SENN and SLDD-Model on *glm-saga* for sparsification, it has also proven effective for QPM [41]. While it is beneficial for the accuracy, as we show in Table 4, this reduces the global interpretability, as their absence, a sum of 0 across the entire feature map **M**, has an effect due to varying means $\boldsymbol{\mu}^{n_f^*}$. Hence, we additionally apply ReLU

Table 7: Average Set Size $|\mathbb{Y}|$ of CHiQPM with Resnet34 calibrated to reach various coverages $1 - \alpha$ comparing different conformal prediction methods. All methods are very close or reach the desired coverage.

| $\|\mathbb{Y}\|\downarrow$ | CUB | | | | CARS | | | |
|---|---|---|---|---|---|---|---|---|
| Method | $\alpha$=0.12 | $\alpha$=0.1 | $\alpha$=0.075 | $\alpha$=0.05 | $\alpha$=0.075 | $\alpha$=0.05 | $\alpha$=0.0025 | $\alpha$=0.001 |
| **Ours** | 1.45 ± 0.19 | 2.22 ± 0.37 | 6.65 ± 1.96 | 26.8 ± 22.5 | 1.05 ± 0.01 | 1.47 ± 0.18 | 12.6 ± 1.19 | 96.0 ± 4.36 |
| THR | 1.25 ± 0.04 | 1.44 ± 0.02 | 1.88 ± 0.05 | 2.9 ± 0.22 | 1.03 ± 0.02 | 1.22 ± 0.06 | 2.57 ± 0.09 | 9.50 ± 2.15 |
| APS | 6.97 ± 0.45 | 7.88 ± 0.39 | 9.80 ± 0.62 | 12.9 ± 0.63 | 6.13 ± 0.69 | 8.20 ± 1.00 | 12.0 ± 0.86 | 21.5 ± 3.18 |

Table 8: Average Set Size $|\mathbb{Y}|$ of CHiQPM with Inception-v3 calibrated to reach various coverages $1 - \alpha$ comparing different conformal prediction methods. All methods are very close or reach the desired coverage.

| $\|\mathbb{Y}\|\downarrow$ | CUB | | | | CARS | | | |
|---|---|---|---|---|---|---|---|---|
| Method | $\alpha$=0.12 | $\alpha$=0.1 | $\alpha$=0.075 | $\alpha$=0.05 | $\alpha$=0.075 | $\alpha$=0.05 | $\alpha$=0.0025 | $\alpha$=0.001 |
| **Ours** | 1.52 ± 0.05 | 2.34 ± 0.12 | 7.63 ± 0.15 | 61.5 ± 4.25 | 1.02 ± 0.01 | 1.47 ± 0.22 | 47.0 ± 22.7 | 81.8 ± 3.04 |
| THR | 1.24 ± 0.04 | 1.47 ± 0.07 | 2.09 ± 0.10 | 4.07 ± 0.38 | 1.00 ± 0.01 | 1.17 ± 0.04 | 3.17 ± 0.39 | 13.5 ± 2.68 |
| APS | 7.08 ± 0.44 | 7.94 ± 0.65 | 10.5 ± 0.96 | 13.8 ± 1.35 | 5.14 ± 0.30 | 6.77 ± 0.76 | 10.8 ± 0.77 | 21.1 ± 1.82 |

to the features after scaling, so that CHiQPM only reasons positively, and negligible activations are suppressed:

$$\mathbf{f}^* = \mathrm{ReLU}(\hat{\mathbf{f}}) \tag{30}$$

Note that this already makes CHiQPM a set-valued predictor, as *e.g.* when only the $n_{\mathrm{wc}} - 1$ features shared by 2 classes in $\mathbb{P}$ activate, both would be predicted. However, on the test datasets we used this happens very seldom and, as explained in Appendix A, we use $\mathrm{argmax}$ to break ties for evaluating point-predictive performance.

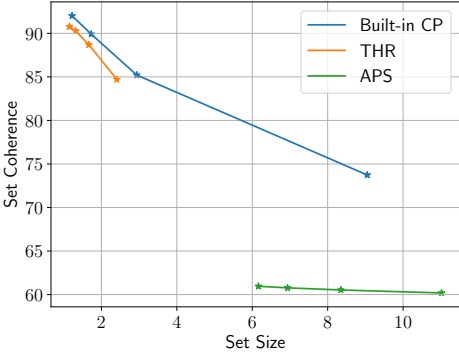

Figure 12: Average Similarity compared to other conformal prediction methods for the same CHiQPM. The stars denote different $\alpha \in \{0.12, 0.1, 0.075, 0.05\}$ used for calibration.

## F.1 Change during Fine-tuning

This section presents results on how much features change during the fine-tuning of CHiQPM. Table 11 quantifies the Pearson Correlation between the same features before and after the fine-tuning in relation to $\rho$. Evidently, a higher $\rho$ causes higher change in the features. As a consequence CHiQPM is less reliant on the perfect choice of initial features, as they undergo more changes regardless. For reference, QPM, has an average correlation of 0.855.

Table 9: Average Set Size $|\mathbb{Y}|$ of CHiQPM with Swin-Transformer-Small calibrated to reach various coverages $1 - \alpha$ comparing different conformal prediction methods. All methods are very close or reach the desired coverage.

| $|\mathbb{Y}|\downarrow$ | CUB | | | | CARS | | | |
|---|---|---|---|---|---|---|---|---|
| Method | $\alpha$=0.12 | $\alpha$=0.1 | $\alpha$=0.075 | $\alpha$=0.05 | $\alpha$=0.075 | $\alpha$=0.05 | $\alpha$=0.0025 | $\alpha$=0.001 |
| **Ours** | $1.14 \pm 0.05$ | $1.44 \pm 0.21$ | $2.44 \pm 0.32$ | $8.09 \pm 0.77$ | $1.28 \pm 0.24$ | $3.3 \pm 2.93$ | $20.2 \pm 25.6$ | $75.0 \pm 3.14$ |
| THR | $1.09 \pm 0.01$ | $1.22 \pm 0.03$ | $1.49 \pm 0.04$ | $2.24 \pm 0.13$ | $1.15 \pm 0.13$ | $1.42 \pm 0.21$ | $2.44 \pm 0.62$ | $7.62 \pm 2.36$ |
| APS | $1.73 \pm 0.05$ | $1.95 \pm 0.08$ | $2.49 \pm 0.16$ | $3.79 \pm 0.39$ | $2.02 \pm 0.32$ | $2.52 \pm 0.43$ | $4.64 \pm 0.94$ | $9.38 \pm 2.05$ |

Table 10: Accuracy and SID@5 of CHiQPM when increasing the weighting of the Feature Diversity Loss $\mathcal{L}_{\text{div}}$ $\beta$ beyond the default value $D$, used for Q-SENN, the SLDD-Model and QPM. Without reducing accuracy, SID@5 can be further increased for CHiQPM.

| Metric | CUB | | | IMG | | |
|---|---|---|---|---|---|---|
| | $\beta = D$ | $\beta = \sqrt{2}D$ | $\beta = 2D$ | $\beta = D$ | $\beta = 4D$ | $\beta = 12D$ |
| Accuracy | 85.3 | 85.3 | 85.1 | 75.3 | 75.3 | 75.1 |
| SID@5 | 88.1 | 92.7 | 96.5 | 42.9 | 70.3 | 77.0 |

## G  Relaxation of Hierarchical Constraint

This section presents the iterative relaxation of the hierarchical constraint introduced in Section 3.1. We relax the constraint of Equation (6) after finding the initial global solution for $\boldsymbol{W}$ and $\boldsymbol{s}$, $\boldsymbol{W}_{\text{init}}$ and $\boldsymbol{s}_{\text{init}}$. Specifically, we set $\boldsymbol{s} = \boldsymbol{s}_{\text{init}}$ to simplify the problem and allow the addition of $\mathbf{m} \in \{0,1\}^{|\mathbb{T}|}$ as an additional variable to optimize to the binary problem without introducing cubic constraints. It directly optimizes which pair of classes shall share $n_{\text{wc}} - 1$ features with the lowest reduction on the objective for the desired density $\rho$. Here $\mathbb{T}$ describes the set of pairs of classes that ever were represented sharing $n_{\text{wc}} - 1$ features during the iterative optimization process. Because the hierarchical constraint causes shared concepts to be represented by the shared features exclusively, other pairs in $\mathbb{P}$ not included in $\mathbb{K}$ emerge as other classes share this concept too. Extending how the QPM (Section 2.2) ensures no duplicates, $\mathbb{T}$ is continuously updated, and the optimum is found when all variables, including $\mathbb{T}$, have not changed for an iteration:

$$(\boldsymbol{w}_c \circ \boldsymbol{w}_{c'})^T \boldsymbol{s}_{\text{init}} \cdot m_i = (n_{\text{wc}} - 1) \cdot m_i \quad \forall (c, c') \in \mathbb{T} \tag{31}$$

$$\sum_{i=0}^{|\mathbb{T}|} m_i \geq |\mathbb{K}|, \tag{32}$$

## H  Impact of $\lambda_{\text{feat}}$

This section is concerned with the weighting factor $\lambda_{\text{feat}}$ of the Feature Grounding Loss $\mathcal{L}_{\text{feat}}$. As described, it is set to $\lambda_{\text{feat}} = 3$, as the ablation in Figure 13 demonstrates that a further increase harms accuracy, without significantly further improving sparsity and feature alignment. This ablation also highlights that features become more aligned with the attributes in CUB-2011, as the sparsity increases. This is likely due to the shared concept between the assigned classes being less abstract. For reference, QPM reaches a Feature Alignment of 1.9, half of CHiQPM, and thus learns significantly less grounded features than CHiQPM.

## I  Impact of ReLU

The necessity of the normalization and the improvements through our method are summarized in Table 4. First, it is clear that normalization is required to achieve state-of-the-art performance and ReLU further improves accuracy. However, having features with a shared fixed minimum, either without normalization or due to ReLU evidently increases Feature Alignment. Additionally, the ReLU significantly boosts Contrastiveness, as all the non-activating features are exactly 0, leading to

Table 11: Impact of $\rho$ on Average Pearson Correlation between features before and after fine-tuning.

| $\rho$ | Average Pearson Correlation |
|---|---|
| 0.0 | 0.825 |
| 0.1 | 0.823 |
| 0.3 | 0.805 |
| 0.5 | 0.776 |
| 0.7 | 0.743 |

a lower distribution $\mathcal{N}_1$ (Equation (18)) with very low standard deviation and therefore an almost perfect distinction between active and inactive features. However, the Class-Independence is slightly reduced with a shared minimum, as there are no minimal inactive activations that otherwise count towards Class-Independence . Regardless, a Class-Independence of around $95\%$ is indistinguishable from a perfect score, as a perfect feature, that equally activates on all images of its assigned classes and never on other images would have Class-Independence $= 1 - \frac{20}{200} = 90\%$. This calculation is based on the default configuration ($n_{\mathrm{wc}} = 5$ and $n_f^* = 50$) for CUB-2011, where every feature is on average assigned to 20 out of the 200 classes. Hence, the feature activation would be focussed on these $10\%$ of the class samples. A further increased value can be attributed to robust features that are only sometimes part of a classes appearance. One example are birds that change their color during their lifetime, causing these color features to activate even without being assigned to these classes.

## J  Impact of Dynamic Hierarchy

Table 12: Comparison of dynamic and static feature ordering across different levels in the hierarchy for CHiQPM with Resnet50 on CUB-2011.

| Method | Metric | Level 1 | Level 2 | Level 3 | Level 4 | Level 5 |
|---|---|---|---|---|---|---|
| Dynamic Hierarchy (CHiQPM) | Set Accuracy | 96.5 | 93.8 | 91.6 | 89.5 | 85.3 |
| | Set Size | 20.17 | 4.43 | 2.12 | 1.45 | 1.00 |
| Static Hierarchy (Avg. Order) | Set Accuracy | 89.7 | 87.4 | 86.2 | 85.6 | 85.2 |
| | Set Size | 18.52 | 2.85 | 1.30 | 1.09 | 1.00 |

This section evaluates the importance of using our sample specific hierarchy instead of a static one. Our used hierarchy based on order of the features of the predicted class has significant advantages compared to other choices. Its sample specific nature lends itself to a local explanation, as one can consider the order of the features as how dominant they appear in the image. Additionally, that enables traversing up the hierarchy in a meaningful way, since the least certain features get omitted first, causing accurate and efficient sets at each step. We believe that the main alternative is a class specific fixed order of features, as that would enable a global hierarchical explanation for each class. The comparison between a fixed order of features for each class, computed based on the average order on training data and our prediction up the hierarchy is shown in Table 12. Evidently, predicting with such a fixed order causes less accurate inefficient sets compared to our sample specific dynamic hierarchy. For example, our dynamic hierarchy reaches $91.6\%$ accuracy with an average set size of 2.1, while the fixed hierarchy never reaches that accuracy. We believe that this is likely due to the unique feature, e.g. the red eye of the Bronzed Cowbird, being on average quite important to that class and thus causing prediction sets that are too specific to the top-1 prediction.

## K  Localization Quality of Features

This section evaluates the localization quality of the feature maps. We evaluate a form of the Pointing Game, similar to the initial form in [66], using segmentation masks provided for CUB-2011 [12]. Specifically, we calculate the fraction of the activation of the GradCAM [55] saliency map that is focused on the segmented bird $S \in \{0,1\}^{w \times h}$:

$$\phi = \sum \left( \frac{\mathrm{GradCAM}}{\sum \mathrm{GradCAM}} \odot S \right) \tag{33}$$

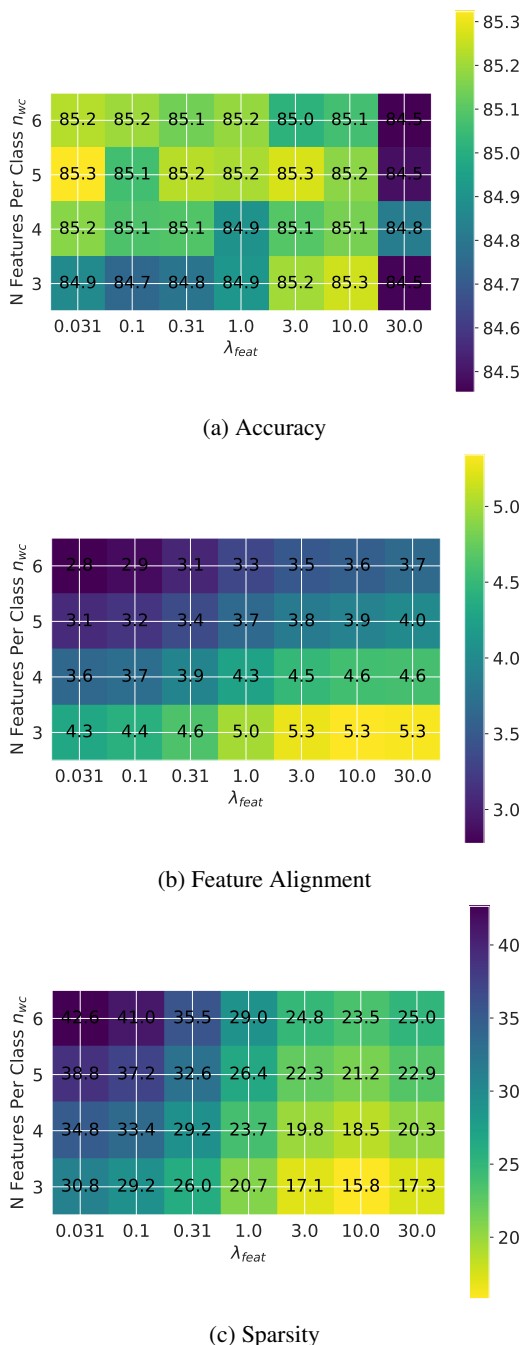

(a) Accuracy

(b) Feature Alignment

(c) Sparsity

Figure 13: Accuracy, Feature Alignment and Sparsity in relation to $\lambda_{\text{feat}}$ and the number of features per class $n_{\text{wc}}$ on CUB-2011

As the bird can be considered the region of the image responsible for the classification, a higher overlap with the segmentation indicates that the saliency maps localize more faithfully. The results are shown in Table 13. Importantly, one would not necessarily expect an overlap of 100%, as the edge region is relevant to describing the shape, causing activations both on and off the segmentation. Across the entire test dataset, CHiQPM's GradCAM focusses to 83.4% on the bird, whereas the Dense baseline only does so with 75.8%. Hence, CHiQPM has activation maps with improved faithfulness on CUB-2011, validating their use for saliency maps of individual interpretable features.

Table 13: Accuracy and faithfulness comparison on CUB-2011 with Resnet50. The final column measures the average GradCAM overlap with ground truth segmentations.

| Model | Accuracy | Avg. GradCAM Overlap |
|-------|----------|----------------------|
| Dense | 86.6% | 75.8% |
| CHiQPM | 85.3% | 83.4% |

## L    Limitations

This section discusses the limitations of the proposed CHiQPM. CHiQPM is a model that learns few general features which are used in a very interpretable way with a binary sparse assignment. The learnt features already enable improved interpretability, as they tend to localize consistently on the same concepts enabling an intuitive understanding of the features and predictable local behvaiour based on global explanations. For example, given the global explanation in Figure 1, a user would be able to predict that the image in Figure 2 would show little activation as the eye is different. Notably, this is different to most prototypical models, which learn class-specific features [41] that detect *red eye of Bronzed Cowbird* instead of *red eye* and thus lose human interpretability [20, 24]. However, for perfect human-understandable global interpretability all of the individual features have to be understandable for humans. Dependent on the dataset, there are 2 to 3 remaining obstacles:

1. As touched upon in Section 2.1, features can learn to detect multiple concepts, a phenomenon knows as polysemanticity. While the visualizations of features in Figures 31 to 34 indicate a consistent localization of the same feature across many classes on the same concept, proper metrics are missing to even measure that. However, we believe that a model like CHiQPM is very well suited to investitage this phenomen further, as superposition [11] can likely be ruled out in its final features. The issue of polysemanticity is exacerbated on ImageNet-1K, where every feature is assigned to $\approx 100$ classes. This connects to another limitation, where the number of selected features $n_f^*$ can not be set arbitrarily high, as the time it takes to solve the QP grows exponentially with it, as shown in [41].

2. While the activation maps of CHiQPM on CUB-2011 and Stanford Cars seem to localize very accurately, *e.g.* highlighting the red eye and not activating if it is not visible, the activation maps on ImageNet-1K seem to not always faithfully highlight the image region they respond to, *e.g.* Feature 23 in Figures 19, 29 and 30 consistently distributes a large portion of its activation on the same image patch. More elaborate saliency map methods or built-in methods like B-cos Networks [5] might be required for such activation maps. Finally, with the seemingly lacking faithfulness of the activation maps for ImageNet-1K, the use of SID@5 for this dataset can also be questioned.

3. CHiQPM learns general features that are well suited to classify the dataset given the training data. However, we do not restrict the features to be based on concepts that humans have noticed or named before. Therefore, there may exist a conceptual gap between the concepts learnt by CHiQPM and the ones known to humans. This Bi-directional Communication Problem [3] can however be reduced by examining the learnt general features of CHiQPM with its faithful global interpretability, as one can *e.g.* say that there is a distinctive pattern on the necks of the black birds, that can be used to differentiate them. Teaching humans the general shared features as neologisms [19] seems promising to bridge this gap. Again, the learnt features of CHiQPM are uniquely suited as they are individual neurons that already detect concepts. Finally, further work should aim at distinguishing polysemantic features from features that capture one concept unknown to humans.

The proposed method is evaluated quite thoroughly on multiple datasets with various architectures. Thus, CHiQPM can clearly be applied to other general vision datasets. However, these datasets need to have many classes that have shared concepts, as the method relies on learning shared concepts between the classes.

For assumptions, it is to note that the guaranteed coverage of Conformal Prediction only holds under the exchangeability assumption of test and calibration data. We designed our experiments to ensure this property. Specifically, splitting calibration off from the test data ensures i.i.d., which is a stronger

property than exchangeability. Additionally, we observe that all CP methods closely match the target coverage for which we calibrate on test data.

Finally, the built-in interpretable Conformal Prediction method of this work guarantees unconditional coverage under the given assumptions, *i.e.* achieving the desired accuracy across the entire test dataset, instead of for all test samples. Achieving unconditional instead of conditional coverage can relate to fairness, as some groups might be undercovered. Limiting the number of levels to ascend during the conformal prediction can theoretically further negatively affect the conditional coverage, as some classes are further away from other classes. However, conditional coverage is challenging to even measure, as one needs a fair assessment of difficulty. For the metrics we checked, using $n^{\text{limit}}$ did not negatively affect conditional coverage. However, as the set construction is interpretable for CHiQPM, the user can understand high uncertainty when all classes below the maximum number $n^{\text{limit}}$ are predicted, which may reduce the need for built-in conditional coverage.

## M  Future Work

This section discuses the future directions of this work. One direction is a formal analysis of the used QP. Understanding the properties in more detail might enable faster solving or scaling the model to even larger datasets that might require even more features.
Another very valuable direction is validating the effectiveness of the introduced explanations via human studies, which have been out of scope for this work.
Finally, we hypothesize that CHiQPM is well suited to handle class-imbalanced data, as the constraints on the QP ensure that every class can be predicted if their general features are present.

## N  Broader Impact

This paper presents work whose goal is to advance the field of Machine Learning towards more interpretability. Globally interpretable models enable a deeper understanding of the decision-making before and during deployment, and can therefore lead to increased safety, robustness, trustworthiness and potentially even scientific discovery. However, there are also potential negative consequences. Most notable, a bad actor can use a model with faithful global interpretability to deliberately and systematically ensure decision-making based on spurious or discriminating factors. Nevertheless, only such globally interpretable models offer the transparency to robustly ensure decision-making based on the correct concepts.

## O  Compute Ressources and Runtime

This section discusses the compute ressources used for the experiments in this paper. As GPU ressource, this work made use of an internal cluster composed of several NVIDIA RTX 2080 Ti. Every experiment fit on one GPU. As CPU ressource for solving the QP, this paper used an internal CPU cluster composed of primarily AMD EPYC 72F3 and up to 250GB of ram. This reference hardware with sufficient memory is assumed for all estimates.

Table 14: Rough time in minutes needed to obtain a CHiQPM, $n_{\text{wc}} = 5$ and $n_f^* = 50$, with Resnet50 on the three datasets used in this work.

| Dataset | Dense Training (GPU) | QP (CPU) | Fine-Tuning (GPU) | Total Time (Hours) |
|---|---|---|---|---|
| CUB-2011 | 176 | 200 | 82 | 458 (7.6) |
| Stanford Cars | 234 | 200 | 110 | 544 (9.1) |
| ImageNet-1K | 0 | 660 | 6300 | 6960 (116) |

All main experiments were done on four architectures. The total time it takes to obtain a CHiQPM dependes on the time spent training on the GPU, scaling with model and dataset size, and time spent solving the QP on the CPU, which scales with the dimensions of the variables and constraints. With $n_{\text{wc}} = 5$ and $n_f^* = 50$, the CPU time depends primarily on the number of classes in the dataset $n_c$ and the number of features of the black-box dense model $n_f$. As Stanford Cars has a very similar number of classes to CUB-2011 (196 to 200), we calculate with the same number of classes and

Table 15: Rough time for the 5 seeds in minutes (hours) spent on training each of the architectures for the main experiments on CUB-2011 and Stanford Cars (taking $\frac{4}{3}$ the time of CUB-2011 on GPU).

| Architecture | GPU | CPU |
|---|---|---|
| Resnet50 | $258 \cdot 5 \cdot \frac{7}{3}$ | $200 \cdot 5 \cdot 2$ |
| Resnet34 | $143 \cdot 5 \cdot \frac{7}{3}$ | $58 \cdot 5 \cdot 2$ |
| Inception-v3 | $76 \cdot 5 \cdot \frac{7}{3}$ | $200 \cdot 5 \cdot 2$ |
| Swin-Transformer-Small | $478 \cdot 5 \cdot \frac{7}{3}$ | $200 \cdot 5 \cdot 2$ |
| $\sum$ | 11141 (185) | 6580 (110) |

Table 16: Rough time in hours spent on training the additional models for the Ablations

| Ablation | Number of Models | GPU | CPU |
|---|---|---|---|
| Table 3 | 15 | 21 | 0 |
| Table 4 | 10 | 14 | 0 |
| Figure 9 | 70 | 75 | 230 |
| Table 10 | 12 (6 on IMG) | 352 | 0 |
| Figure 11 (QPM) | 5 | 4 | 17 |
| Figure 6 | 15 | 21 | 50 |
| $\sum$ | 127 | 487 | 297 |

$\frac{4}{3}$ times the number of training samples (8144 to 5994). The estimated time spent per dataset is shown in Table 14. How that varies between architectures is shown in Table 15. While there is some variance in cpu time due to the ablations, we assume even cpu runtime for our estimates in Table 16, which effectively results in an upper bound on the cpu time spent.

Generating the visualizations or explanations took negligible compute ressources as everything required is computed in one forward pass. Similarly, evaluating the models is also very fast in comparison.

Finally, most of the experiments with Resnet50 were ran on $3 - 5$ random seeds with CUB-2011 before starting training for the fixed seeds and across multiple architectures. Similarly, further experiments of roughly the same number were run to ultimately converge to the presented method. Also, starting from 61GB for solving the QP, very few seeds needed more than that, which we started after the inital ones crashed. For the three solved QPs on ImageNet-1K, we always allocated the available memory of 250G. Additionally, we had to retrain Q-SENN to include it in our tables. Each of these runs takes roughly half an hour more on the GPU than CHiQPM. Note that, Q-SENN did not always converge with Swin-Transformer-Small, which is why no result is reported there as one likely needs to tweak hyperparameters to ensure convergence. Similarly, Figure 11 required training QPM. We want to emphasize that we reused the same dense model or even QP solution where possible, *e.g.* for CHiQPM, QPM and Q-SENN or many of the ablations. This puts the total amount of GPU hours to roughly 1382 and CPU hours to 1040, shown in Table 17.

For storage, we temporarily save every model and their low-dimensional feature vectors to speed up metric calculations. While our internal clusters offers significantly more storage, the experiments of this paper needed less than 100 GB.

## P   Further Qualitative Examples

This section comments the extensive qualitative examples. For all examples, we always present explanations for the same model, i.e. all explanations for a CHiQPM on CUB-2011 with $n_{\mathrm{wc}} = 5$ and $n_f^* = 50$ explain the same model. Therefore, this section discusses results of 4 models, 3 per dataset with the default configuration and also visualizations of CHiQPM with $n_{\mathrm{wc}} = 3$ and $n_f^* = 30$ trained on CUB-2011 and explained in Figures 1 and 2. Figure 14 first contains the global class explanation between the Red-winged blackbird and Bronzed Cowbird, which are jointly predicted in Figure 2. The CHiQPM faithfully communicates that it differentiates them based on their unique attributes: The red eye or wing. Additionally, Figure 21 contrasts the explanation of Figure 2 with one for an

Table 17: Rough total time in hours spent on training with reference hardware NVIDIA RTX 2080 Ti as GPU and AMD EPYC 72F3 as CPU

| Experiment | GPU | CPU |
|---|---|---|
| Main Experiments (Table 15) | 185 | 110 |
| Ablations (Table 16) | 487 | 297 |
| ImageNet Main Experiments (Table 14) | 350 | 33 |
| Preliminary / Failed experiments | 300 | 600 |
| Q-SENN [40] training as competitor | 60 | 0 |
| $\sum$ | 1382 | 1040 |

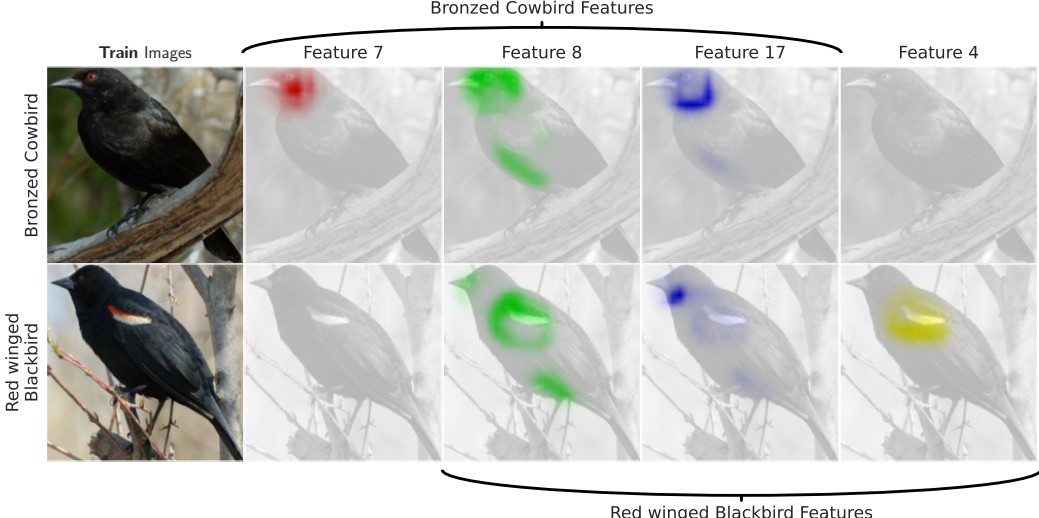

Figure 14: Global explanation comparing Bronzed Cowbird and Red Winged Blackbird. The CHiQPM trained on CUB-2011 with 3 features per class and 30 in total, explained in Figures 1 and 2, determined the differentiating factor, their red eye or wing, without any annotations and can communicate its behavior faithfully.

easy example. Further contrastive global class explanations with human understandable differences on CHiQPM trained on CUB-2011, Stanford Cars and ImageNet-1K are shown in Figures 15 to 20. Additionally, Figure 21 displays the extensive local explanations and an example where CHiQPM dynamically predicts the appropriate set of classes based on visible evidence, as globally explained by Figure 17. Supporting Figures 1, 2 and 21, Figures 23 and 24 show the corresponding graphs without the limitation of only including classes that share the top feature. Finally, Figures 25 to 30 show more exemplary local explanations of our CHiQPM, including the novel hierarchical explanations and saliency maps that also transport activation, as described in Appendix C.2. They demonstrate how CHiQPM can provide uniquely comprehensive local interpretability for a single test image, faithfully following its global explanations.

## P.1 Feature Generalization Visualizations

This section provides supplementary visualizations demonstrating that the features learned by our CHiQPM are not merely class-specific detectors but function as general concept detectors across a diverse range of images. Figures 31 to 34 visualize how the features of the probed CHiQPMs explained in Figures 1, 2, 14 to 16, 21 and 25 to 28 generalize across a huge range of images and classes. For that, we visualized all classes that share all but one feature with one class. We choose Shiny Cowbird for Figures 31 and 32, as it has a large neighborhood in the probed CHiQPM's class representations and is also used in the main paper in Figures 1 and 2. Additionally, we show the features of all classes similarly represented to Arctic Tern in Figures 33 and 34, as it is another class

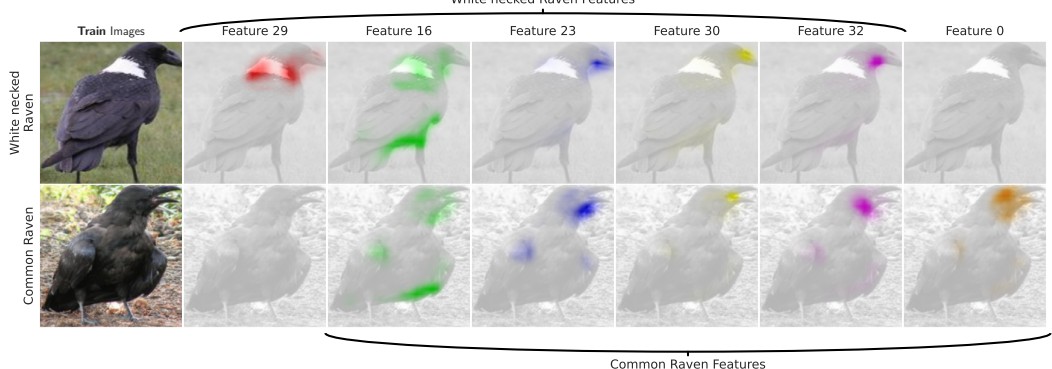

Figure 15: Global explanation comparing White necked and Common Raven. CHiQPM trained on CUB-2011 determined the differentiating factor, the white neck, without any annotations and can communicate its behavior faithfully. Local explanations of this model are shown in Figures 25 and 26.

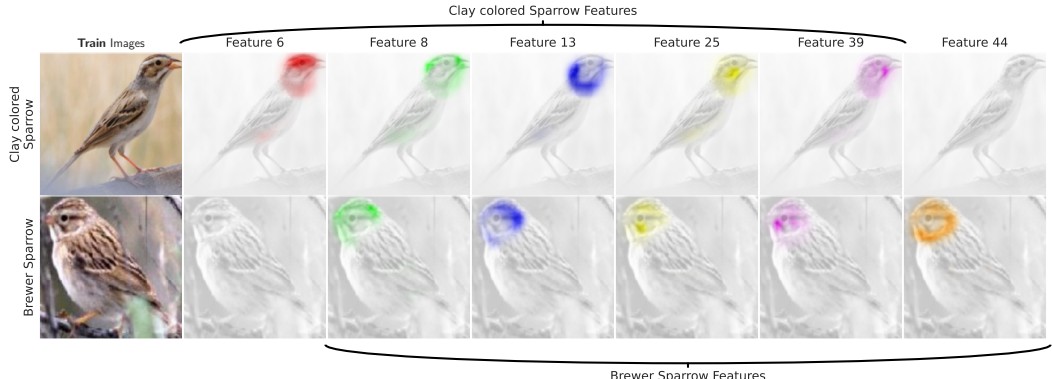

Figure 16: Global explanation comparing Clay colored and Brewer Sparrow. CHiQPM trained on CUB-2011 determined the differentiating factor, the distinct head patterns, *e.g.* the white crown stripe for Clay colored Sparrow, without any annotations and can communicate its behavior faithfully. Local explanations of this model are shown in Figures 27 and 28.

with a large neighborhood in both models. We further supplemented Figure 32 with the Raven classes from Figures 15, 25 and 26 to showcase that the features localize consistently, not just directly around the chosen class. These visualizations demonstrate that similarly represented classes by CHiQPM are indeed similar in reality, as measured by Structural Grounding, and that the features are general concept detectors.

Table 18: Accuracy and Compactness using Resnet50: CHiQPM shows the highest accuracy among interpretable models. Figure 9 shows the increasing gap when further raising compactness. Among more compact models, the best result is marked in bold, second best underlined.

| Method | Accuracy ↑ | | | Total Features↓ | | | Features / Class↓ | | |
| --- | --- | --- | --- | --- | --- | --- | --- | --- | --- |
| | CUB | CARS | IMGNET | CUB | CARS | IMGNET | CUB | CARS | IMGNET |
| Baseline Resnet50 | 86.6±0.2 | 92.1±0.1 | 76.1 | 2048 | 2048 | 2048 | 2048 | 2048 | 2048 |
| glm-saga$_5$ | 78.0±0.4 | 86.8±0.6 | 58.0±0.0 | 809±8 | 807±10 | 1627±1 | **5** | **5** | **5** |
| PIP-Net | 82.0±0.3 | 86.5±0.3 | - | 731±19 | 669±13 | - | 12 | 11 | - |
| ProtoPool | 79.4±0.4 | 87.5±0.2 | - | 202 | 195 | - | 202 | 195 | - |
| SLDD-Model | 84.5±0.2 | 91.1±0.1 | 72.7±0.0 | **50** | **50** | **50** | **5** | **5** | **5** |
| Q-SENN | 84.6±0.3 | _91.8±0.3_ | _74.3±0.0_ | **50** | **50** | **50** | **5** | **5** | **5** |
| QPM | _85.1±0.3_ | _91.8±0.3_ | 74.2±0.0 | **50** | **50** | **50** | **5** | **5** | **5** |
| CHiQPM (**Ours**) | **85.3±0.3** | **91.9±0.2** | **75.3±0.0** | **50** | **50** | **50** | **5** | **5** | **5** |

Table 19: Comparison on QPM [41] Interpretability metrics with Resnet50. Requiring annotations, Structural Grounding can only be computed for CUB-2011. Among more compact models, the best result is marked in bold, second best underlined.

| Method | SID@5 ↑ | | | Class-Independence ↑ | | | Contrastiveness↑ | | | Structural Grounding↑ |
| --- | --- | --- | --- | --- | --- | --- | --- | --- | --- | --- |
| | CUB | CARS | IMGNET | CUB | CARS | IMGNET | CUB | CARS | IMGNET | CUB |
| Baseline Resnet50 | 57.7±0.4 | 54.4±0.3 | 37.1 | 98.0±0.0 | 97.8±0.0 | 99.4 | 74.4±0.1 | 75.1±0.1 | 71.6 | 34.0±0.3 |
| glm-saga$_5$ | 55.4±0.5 | 51.8±0.3 | 35.8±0.0 | **97.8±0.0** | **97.6±0.0** | **99.4±0.0** | 74.0±0.1 | 74.5±0.1 | 71.7±0.0 | 2.5±1.0 |
| PIP-Net | **99.2±0.1** | **99.0±0.1** | - | 75.6±0.4 | 62.9±0.1 | - | _99.6±0.0_ | _99.7±0.0_ | - | 6.7±0.9 |
| ProtoPool | 24.5±0.8 | 30.7±3.4 | - | 96.9±0.1 | 96.0±0.5 | - | 76.7±1.0 | 78.9±2.0 | - | 13.9±0.9 |
| SLDD-Model | 88.2±0.2 | 88.6±0.6 | **64.7±0.7** | 96.2±0.1 | 95.5±0.1 | 98.6±0.0 | 87.3±0.2 | 89.7±0.3 | _93.4±0.1_ | 29.2±4.0 |
| Q-SENN | _93.2±0.4_ | _94.3±0.3_ | - | 95.5±0.1 | 94.8±0.1 | - | 93.0±0.3 | 93.9±0.2 | - | 20.8±4.1 |
| QPM | 90.1±0.3 | 89.6±0.4 | _64.1±0.7_ | _97.0±0.0_ | _96.5±0.0_ | _99.1±0.0_ | 96.0±0.4 | 97.7±0.4 | 89.3±0.1 | _47.9±2.7_ |
| CHiQPM (**Ours**) | 88.1±0.5 | 88.8±1.0 | 42.9±0.9 | 94.1±0.0 | 93.5±0.1 | 98.7±0.0 | **99.9±0.0** | **100.0±0.0** | **99.9±0.0** | **75.0±2.2** |

Table 20: Accuracy and Compactness using Resnet34: CHiQPM shows the highest accuracy among interpretable models. Among more compact models, the best result is marked in bold, second best underlined.

| Method | Accuracy ↑ | | Total Features↓ | | Features / Class↓ | |
| --- | --- | --- | --- | --- | --- | --- |
| | CUB | CARS | CUB | CARS | CUB | CARS |
| Baseline Resnet34 | 85.7±0.3 | 91.5±0.2 | 2048 | 2048 | 2048 | 2048 |
| glm-saga$_5$ | 72.0±1.0 | 82.0±0.6 | 442±5 | 453±6 | **5** | **5** |
| SLDD-Model | 83.2±0.3 | 90.7±0.3 | **50** | **50** | **5** | **5** |
| Q-SENN | **83.7±0.2** | _91.3±0.3_ | **50** | **50** | **5** | **5** |
| QPM | 83.0±0.2 | _91.3±0.0_ | **50** | **50** | **5** | **5** |
| CHiQPM (**Ours**) | **83.7±0.2** | **91.5±0.2** | **50** | **50** | **5** | **5** |

Table 21: Comparison on QPM [41] Interpretability metrics with Resnet34. Requiring annotations, Structural Grounding can only be computed for CUB-2011. Among more compact models, the best result is marked in bold, second best underlined.

| Method | SID@5 ↑ | | Class-Independence ↑ | | Contrastiveness↑ | | Structural Grounding↑ |
| --- | --- | --- | --- | --- | --- | --- | --- |
| | CUB | CARS | CUB | CARS | CUB | CARS | CUB |
| Baseline Resnet34 | 62.1±0.3 | 56.6±0.4 | 97.9±0.0 | 97.7±0.0 | 76.4±0.1 | 77.9±0.2 | 39.6±0.2 |
| glm-saga$_5$ | 59.9±0.4 | 55.3±0.3 | **97.9±0.0** | **97.7±0.0** | 76.5±0.0 | 77.8±0.2 | 7.6±2.2 |
| SLDD-Model | _90.1±0.8_ | 86.7±2.5 | _97.5±0.0_ | _97.6±0.2_ | 86.0±1.0 | 83.3±4.6 | 24.5±2.7 |
| Q-SENN | 86.6±1.4 | 80.1±2.8 | 96.6±0.1 | 96.0±0.2 | 95.2±1.0 | 94.3±1.4 | 25.7±2.0 |
| QPM | **90.5±0.5** | **89.1±1.1** | _97.5±0.0_ | 96.9±0.1 | _95.5±0.2_ | _94.7±1.1_ | _39.0±2.9_ |
| CHiQPM (**Ours**) | 87.9±0.5 | _88.4±1.0_ | 94.1±0.1 | 82.9±4.5 | **99.9±0.0** | **100.0±0.0** | **68.5±4.1** |

Table 22: Accuracy and Compactness using Inception-v3: CHiQPM shows the highest accuracy among interpretable models. Among more compact models, the best result is marked in bold, second best underlined.

| Method | Accuracy ↑ | | Total Features↓ | | Features / Class↓ | |
| --- | --- | --- | --- | --- | --- | --- |
| | CUB | CARS | CUB | CARS | CUB | CARS |
| Baseline Inception-v3 | 86.1±0.1 | 92.6±0.2 | 2048 | 2048 | 2048 | 2048 |
| glm-saga$_5$ | 79.2±0.5 | 89.3±0.3 | 814±9 | 795±8 | **5** | **5** |
| SLDD-Model | 83.1±0.4 | 91.1±0.2 | **50** | **50** | **5** | **5** |
| Q-SENN | 83.8±0.4 | 91.6±0.2 | **50** | **50** | **5** | **5** |
| QPM | 84.2±0.4 | 91.7±0.1 | **50** | **50** | **5** | **5** |
| CHiQPM (**Ours**) | **84.3**±0.3 | **92.1**±0.2 | **50** | **50** | **5** | **5** |

Table 23: Comparison on QPM [41] Interpretability metrics with Inception-v3. Requiring annotations, Structural Grounding can only be computed for CUB-2011. Among more compact models, the best result is marked in bold, second best underlined.

| Method | SID@5 ↑ | | Class-Independence ↑ | | Contrastiveness↑ | | Structural Grounding↑ |
| --- | --- | --- | --- | --- | --- | --- | --- |
| | CUB | CARS | CUB | CARS | CUB | CARS | CUB |
| Baseline Inception-v3 | 38.9±0.3 | 33.1±0.2 | 96.1±0.0 | 95.7±0.0 | 89.6±0.2 | 91.7±0.2 | 7.1±9.6 |
| glm-saga$_5$ | 39.3±0.2 | 34.0±0.4 | **95.4**±0.0 | **95.0**±0.0 | 91.3±0.3 | 93.4±0.2 | 0.3±0.4 |
| SLDD-Model | **58.1**±1.2 | **52.1**±1.5 | 92.6±0.1 | 92.1±0.1 | 93.0±0.3 | 94.4±0.2 | 24.4±2.3 |
| Q-SENN | 55.6±0.9 | 48.2±0.9 | 92.2±0.2 | 91.5±0.1 | 94.6±0.3 | 95.1±0.4 | 19.7±2.6 |
| QPM | 48.6±0.9 | 42.8±0.8 | 95.1±0.1 | 94.7±0.0 | 93.4±0.1 | 94.3±0.1 | 34.8±3.4 |
| CHiQPM (**Ours**) | 45.4±1.3 | 37.8±2.2 | 93.4±0.1 | 92.7±0.2 | **100.0**±0.0 | **100.0**±0.0 | **52.8**±4.5 |

Table 24: Accuracy and Compactness using Swin Transformer small: CHiQPM shows the highest accuracy among interpretable models. Among more compact models, the best result is marked in bold, second best underlined.

| Method | Accuracy ↑ | | Total Features↓ | | Features / Class↓ | |
| --- | --- | --- | --- | --- | --- | --- |
| | CUB | CARS | CUB | CARS | CUB | CARS |
| Baseline Swin Transformer small | 87.0±0.1 | 90.6±0.6 | 768 | 768 | 768 | 768 |
| glm-saga$_5$ | 76.5±0.4 | 75.5±1.2 | 572±4 | 559±8 | **5** | **5** |
| SLDD-Model | 85.3±0.4 | **89.1**±0.7 | **50** | **50** | **5** | **5** |
| QPM | 85.0±0.4 | 88.7±0.5 | **50** | **50** | **5** | **5** |
| CHiQPM (**Ours**) | **85.9**±0.3 | 89.0±1.6 | **50** | **50** | **5** | **5** |

Table 25: Comparison on QPM [41] Interpretability metrics with Swin Transformer small. Requiring annotations, Structural Grounding can only be computed for CUB-2011. Among more compact models, the best result is marked in bold, second best underlined.

| Method | SID@5 ↑ | | Class-Independence ↑ | | Contrastiveness↑ | | Structural Grounding↑ |
| --- | --- | --- | --- | --- | --- | --- | --- |
| | CUB | CARS | CUB | CARS | CUB | CARS | CUB |
| Baseline Swin Transformer small | 26.4±0.1 | 26.0±0.1 | 96.8±0.0 | 96.6±0.0 | 98.3±0.1 | 98.8±0.1 | 24.5±0.4 |
| glm-saga$_5$ | 26.4±0.2 | 26.1±0.1 | **96.6**±0.0 | **96.4**±0.0 | 99.1±0.1 | 99.6±0.0 | 8.8±2.8 |
| SLDD-Model | 38.0±0.5 | 35.6±1.1 | 93.4±0.1 | 93.3±0.2 | 99.0±0.2 | 99.4±0.2 | 37.2±3.4 |
| QPM | 33.6±0.4 | 32.0±0.3 | 95.2±0.0 | 94.7±0.0 | 98.5±0.3 | 99.1±0.2 | 45.1±3.2 |
| CHiQPM (**Ours**) | **46.1**±0.9 | **45.8**±4.0 | 94.2±0.1 | 93.7±0.1 | **99.9**±0.0 | **99.9**±0.1 | **65.4**±3.6 |

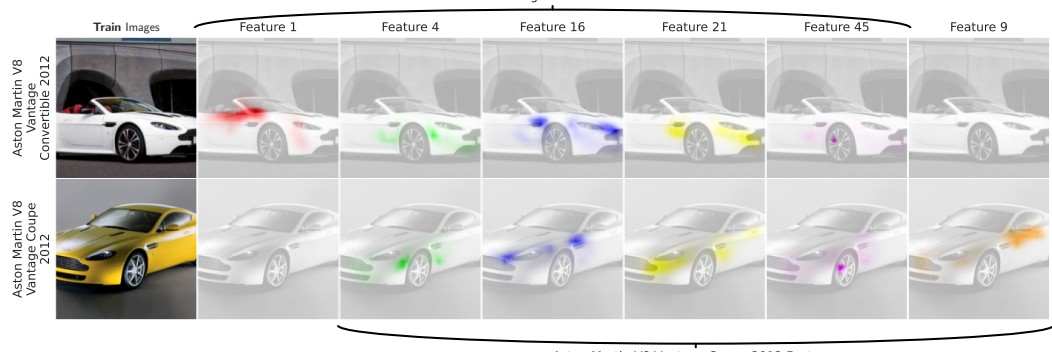

Figure 17: Contrastive global Explanation, comparing the class representations of two cars that only differ in Coupe or Convertible for CHiQPM trained on Stanford Cars that represents every class with 5 of 50 features. They are differentiated based on features activating on where the windows would be.

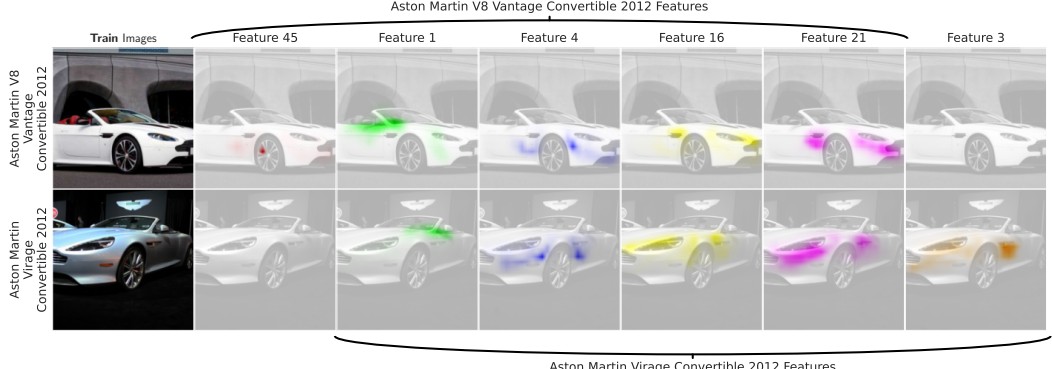

Figure 18: Contrastive global Explanation, comparing the class representations of two Convertible Aston Martins for CHiQPM trained on Stanford Cars that represents every class with 5 of 50 features. They are differentiated based on human perceivable deviating features like the fender vent.

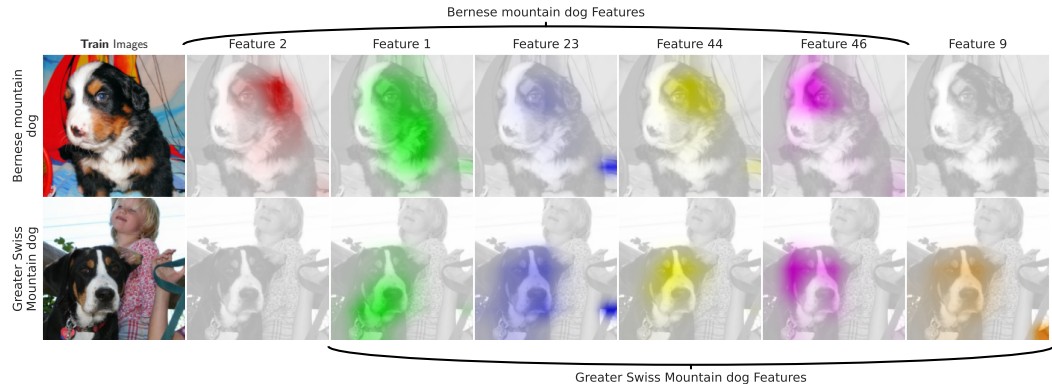

Figure 19: Contrastive global Explanation, comparing the class representations of two mountain dogs for CHiQPM trained on ImageNet-1K that represents every class with 5 of 50 features. They are differentiated based on human perceivable deviating features like the different ear fur. Local explanations for this model are shown in Figures 29 and 30.

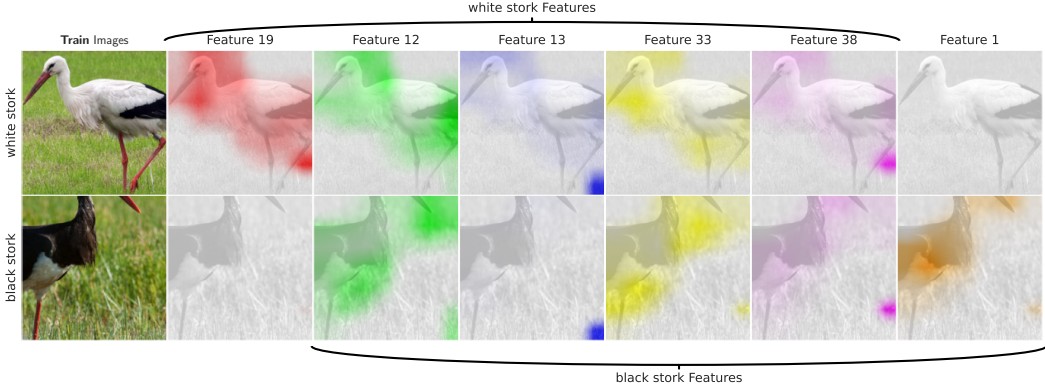

Figure 20: Contrastive global Explanation, comparing the class representations of white and black stork for CHiQPM trained on ImageNet-1K that represents every class with 5 of 50 features. They are differentiated based on one broadly activating feature.

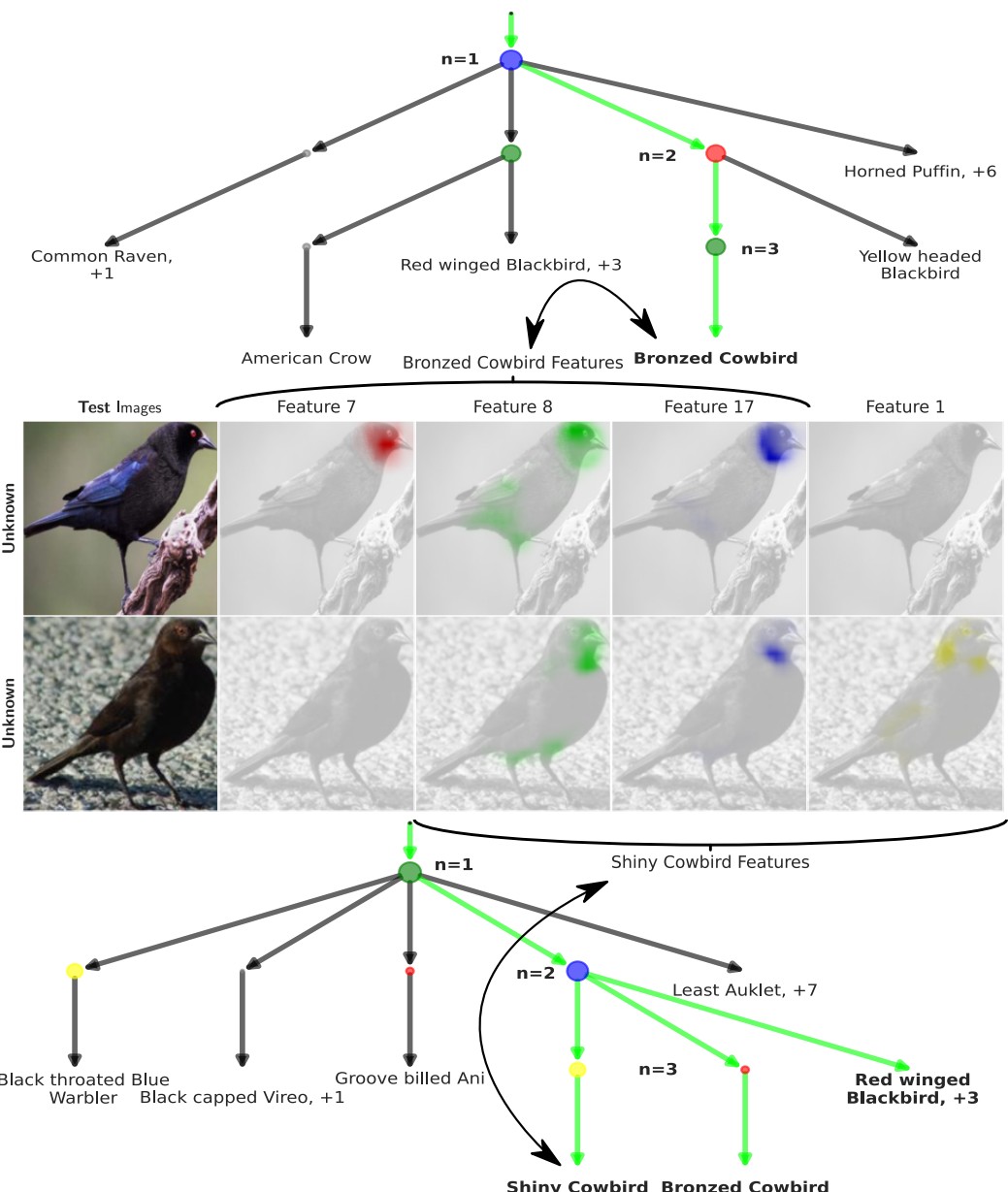

Figure 21: Exemplary local explanations provided by our CHiQPM, with the global explanation in Figure 1 for two test images of Bronzed Cowbird. The first row is an easy example where all 3 features are found, including the red eye. Therefore, our calibrated model only predicts Bronzed Cowbird. The red eye is not visible in the second image, leading to the reasoning of our CHiQPM along its dynamic class hierarchy identifying it as one of the black bird species and predicting all of them, including the Bronzed Cowbird.

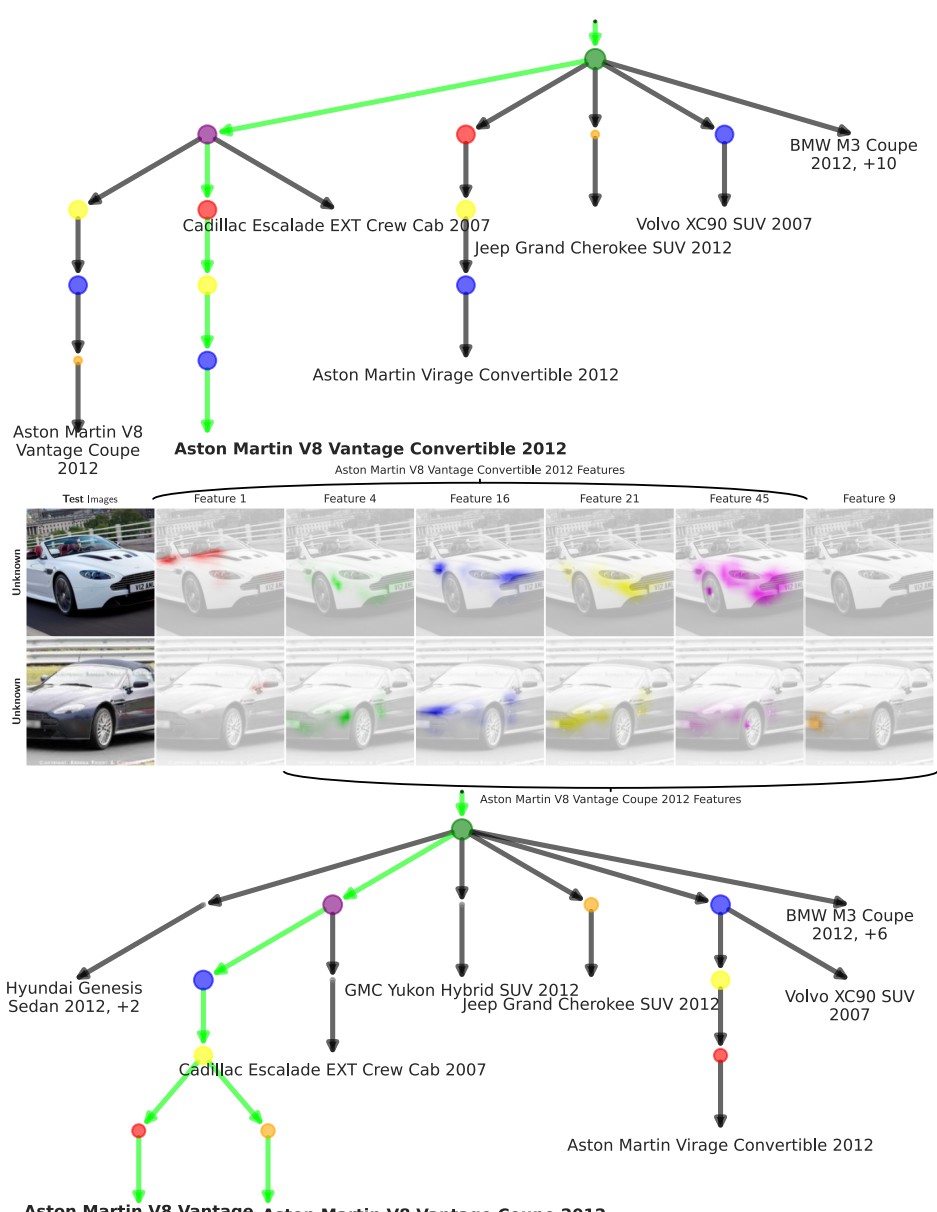

Figure 22: Exemplary local explanations provided by our CHiQPM, with the global explanation in Figure 17 for two test images of the Convertible. The first row is an easy example where all 5 features are found, because the top is down. Therefore, our calibrated model only predicts the Convertible. The top is up in the second image, leading to the reasoning of our CHiQPM along its dynamic class hierarchy identifying it as one of the cars, either Coupe or Convertible because both window features are only barely activating. As the global explanation in Figure 17 explains, the probed CHiQPM does not rely on the fabric top, and hence predicts the set of both cars interpretably.

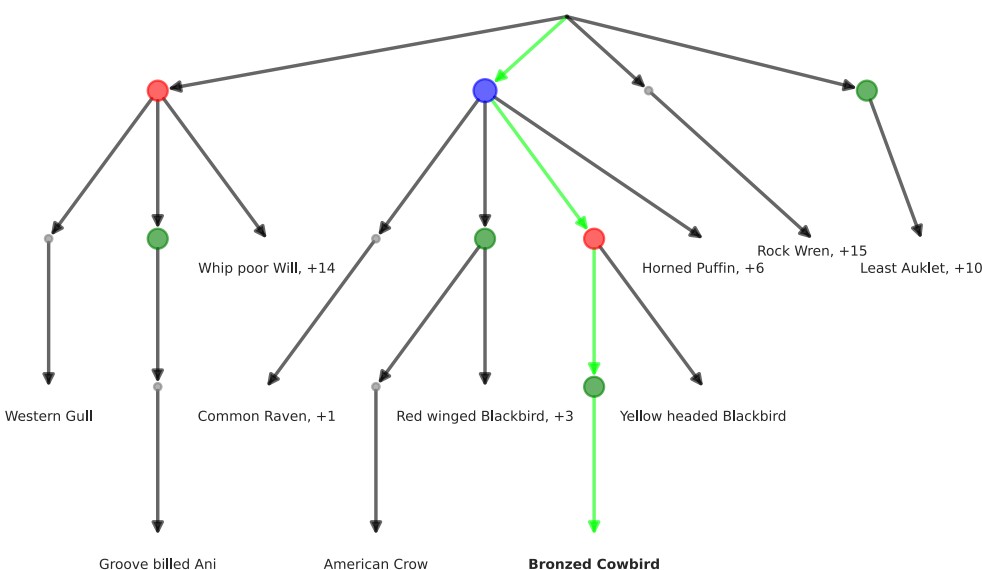

Figure 23: Full graph including all activations for top graph in Figure 21.

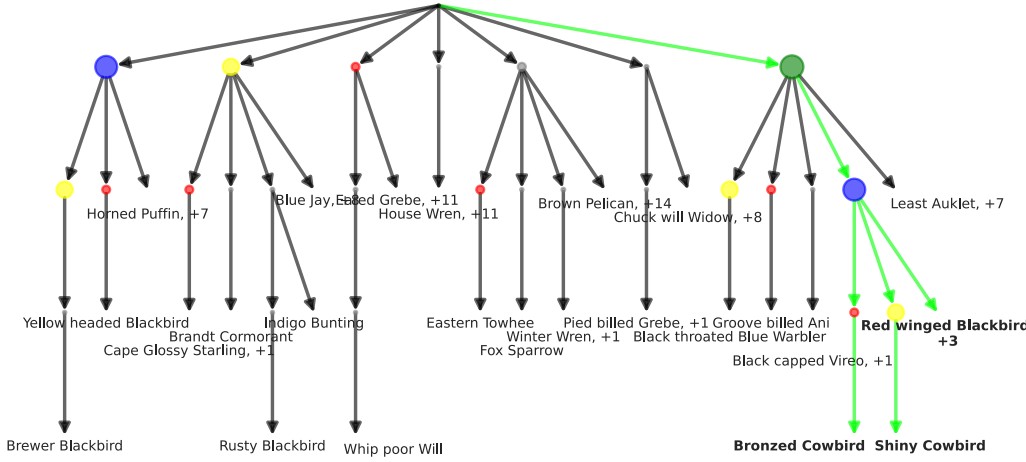

Figure 24: Full graph including all activations for graph in Figure 2.

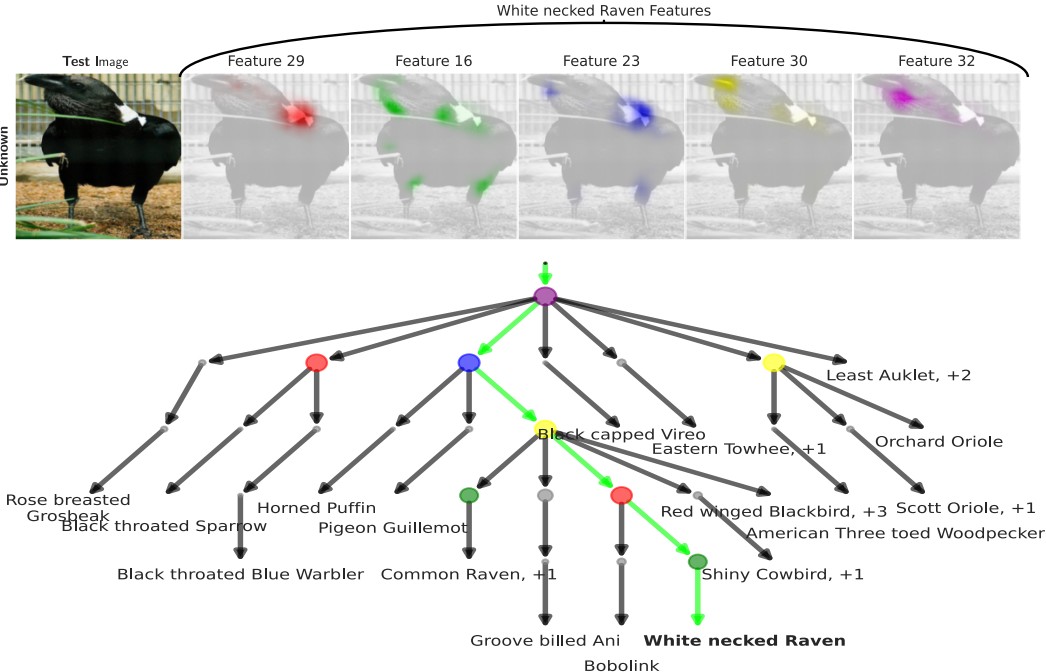

Figure 25: Exemplary local explanation for White necked Raven labeled test sample of CHiQPM with global explanation in Figure 15. All 5 features are recognized, hence only the true label is predicted and the features of the predicted class visualized. The tree further visualizes the learned class similarities, with violet and red leading to black throated birds, whereas violet and yellow seem to indicate *Oriole*.

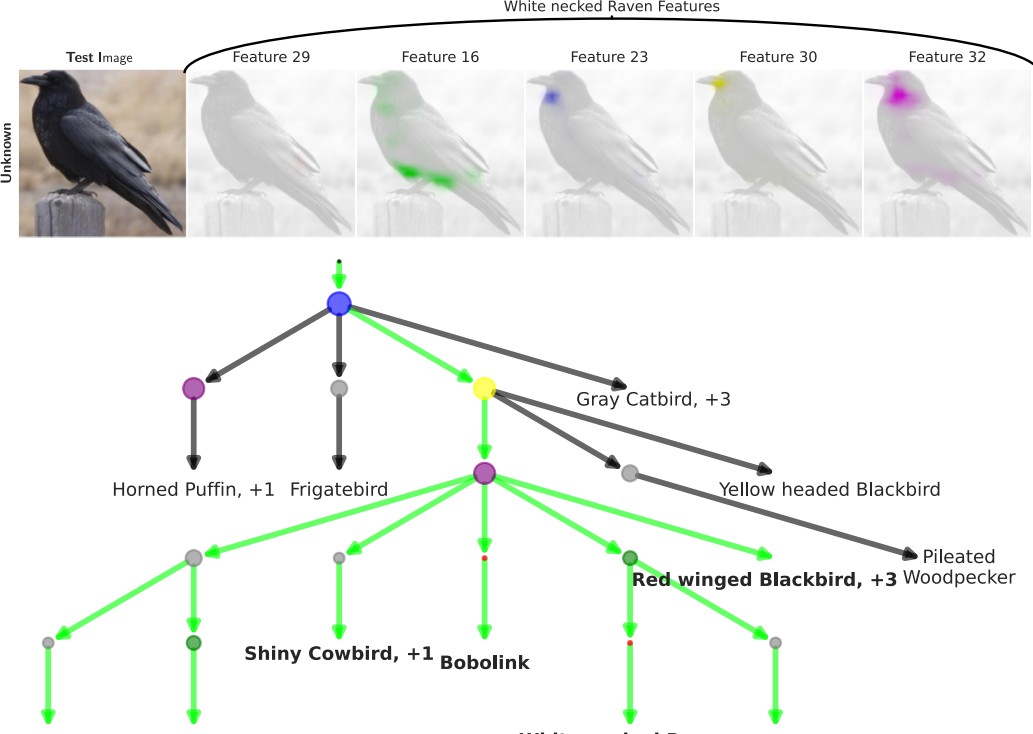

Figure 26: Exemplary local explanation for White necked Raven labeled, presumably mislabelled, test sample of CHiQPM with global explanation in Figure 15. Due to missing evidence, especially the white neck, our calibrated CHiQPM predicts a set of black birds. We show saliency maps for features of White necked Raven as the reader can have learned about its class representation already. Note that the gray feature for Common Raven is shown in Figure 15 too. When CHiQPM determines that it needs to predict a set, one might want to visualize all the features of all classes that are predicted in the coherent set. Alternatively, all active features could be visualized. This is ultimately up to the level of detail desired.

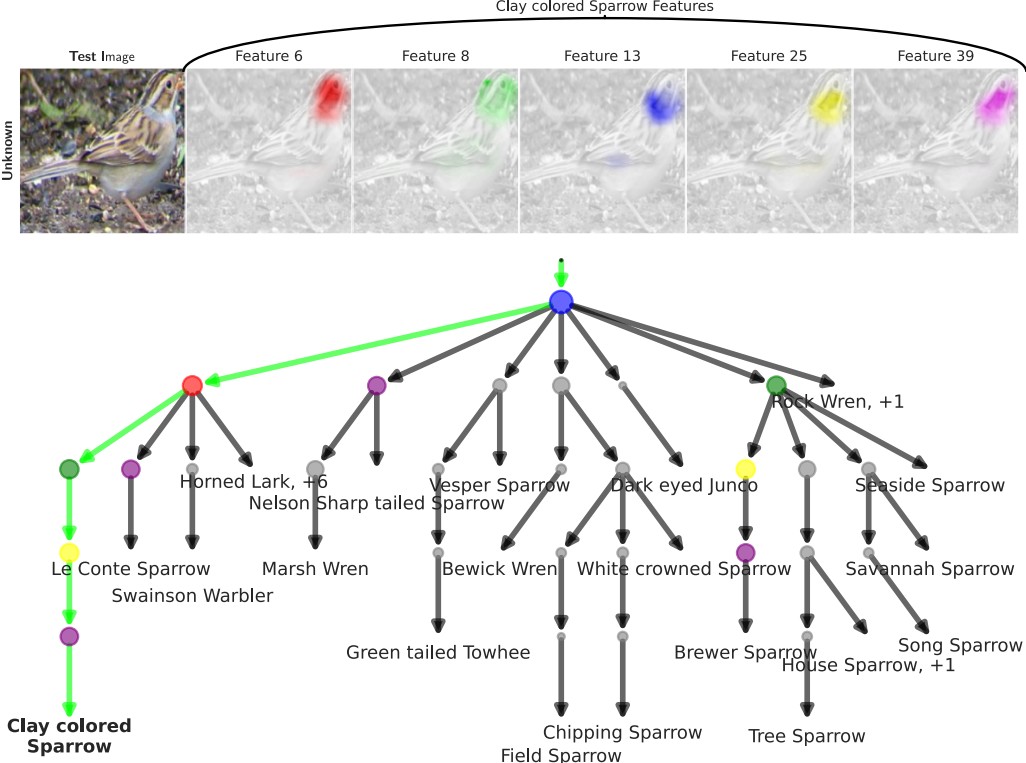

Figure 27: Exemplary local explanation for Clay colored Sparrow labeled test sample of CHiQPM with global explanation in Figure 16. All 5 features are recognized, hence only the true label is predicted and the features of the predicted class visualized. Interestingly, the clearly visible white crown stripe, detected by the red feature, is sufficient for CHiQPM to distinguish it from most other sparrows early on in the hierarchy, as opposed to Figure 28, where blue and green determine a set of sparrows first.

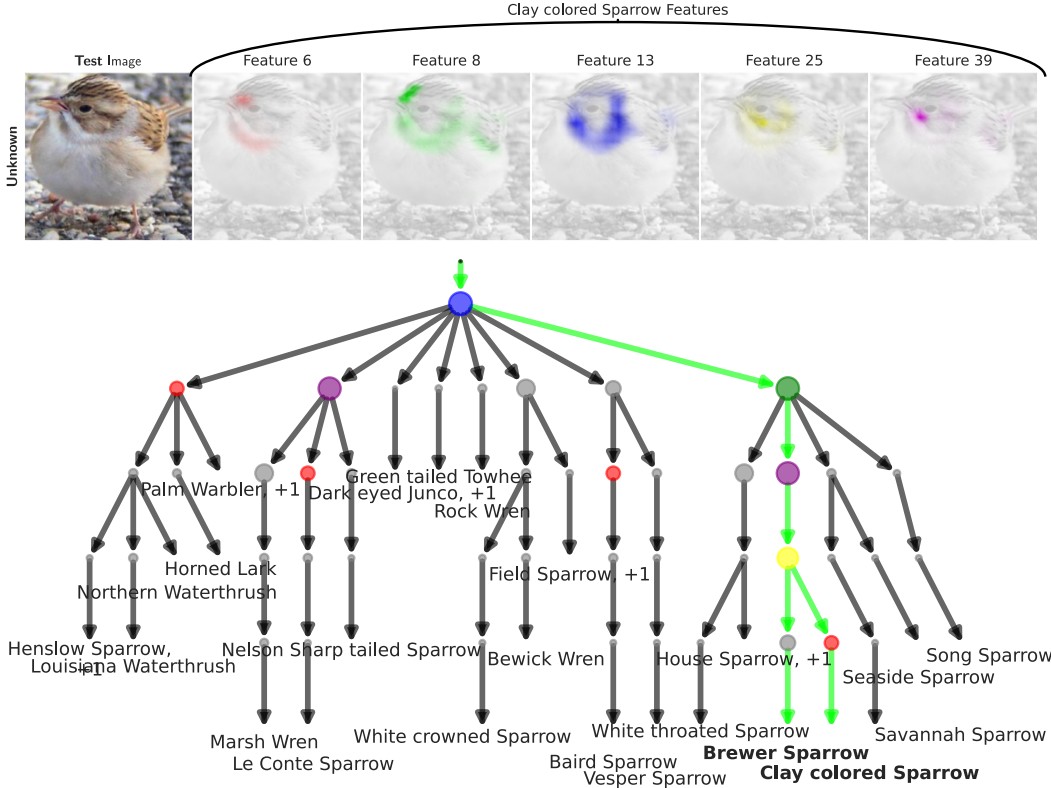

Figure 28: Exemplary local explanation for Clay colored Sparrow labeled test sample of CHiQPM with global explanation in Figure 16. Both head features activate evenly, thus both sparrows are predicted jointly.

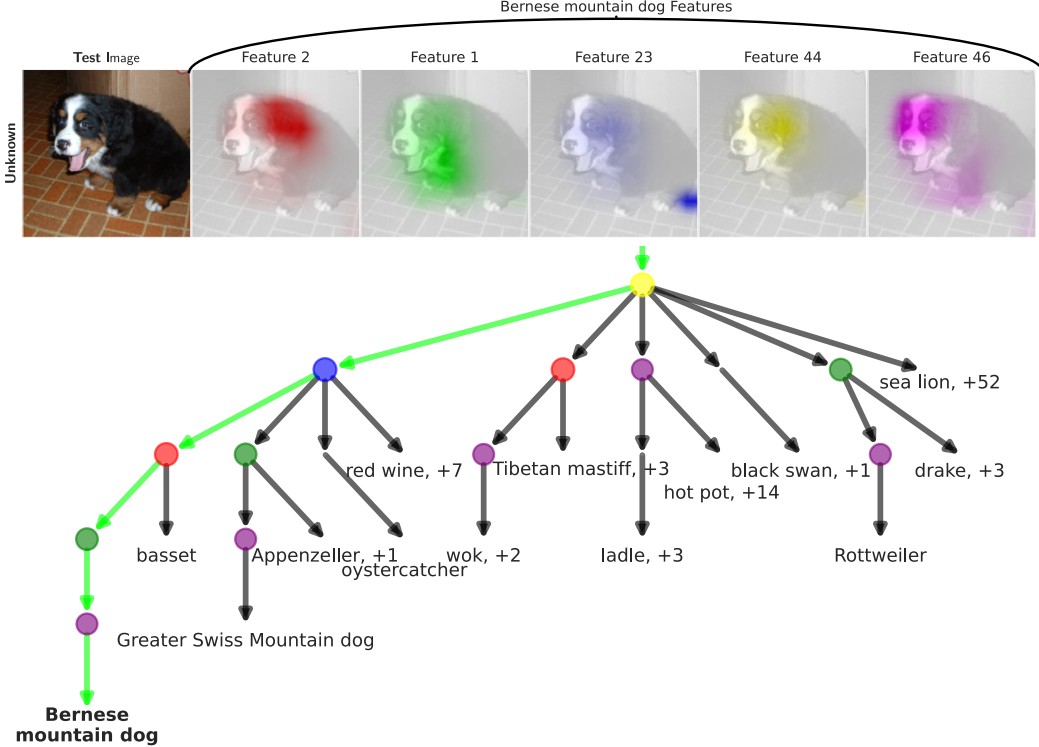

Figure 29: Exemplary local explanation for Bernese mountain dog labeled test sample of CHiQPM with global explanation in Figure 19. All 5 features are recognized, hence only the true label is predicted. The tree further gives a glimpse into how the 1000 classes of ImageNet-1K are organized.

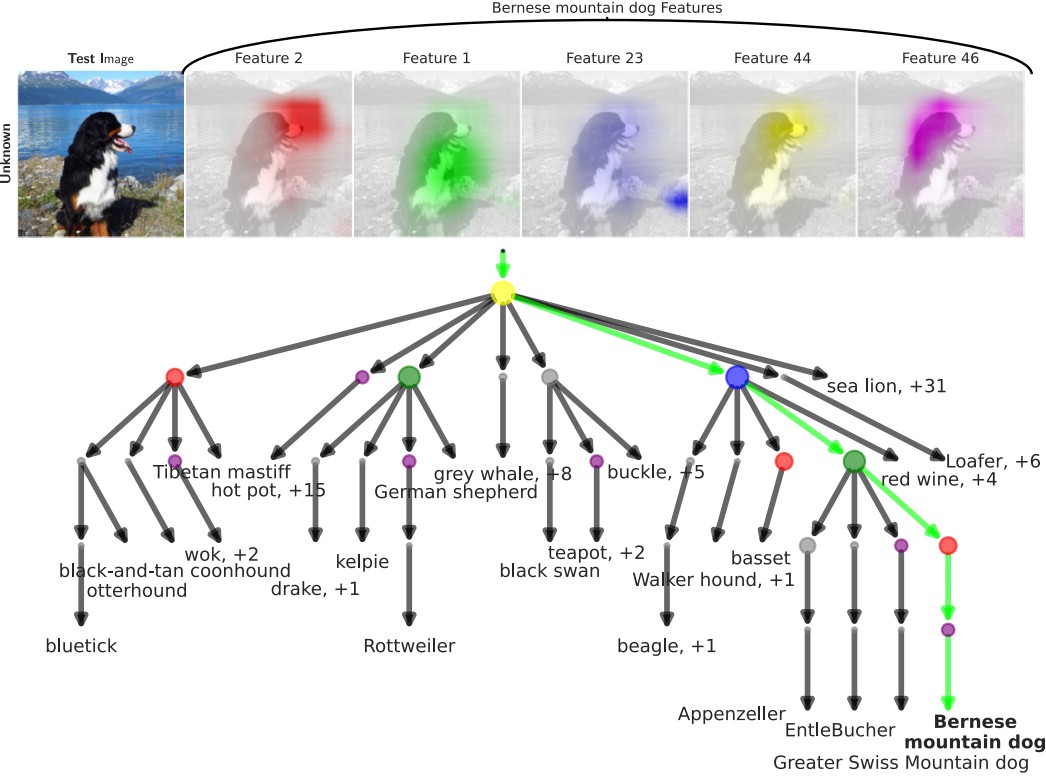

Figure 30: Exemplary local explanation for Bernese mountain dog labeled test sample of CHiQPM with global explanation in Figure 19. All 5 features are recognized, hence only the true label is predicted. However, due to the sideways pose, feature 46, focusing on the frontal face, is less activated. The tree further gives a glimpse into how the 1000 classes of ImageNet-1K are organized.

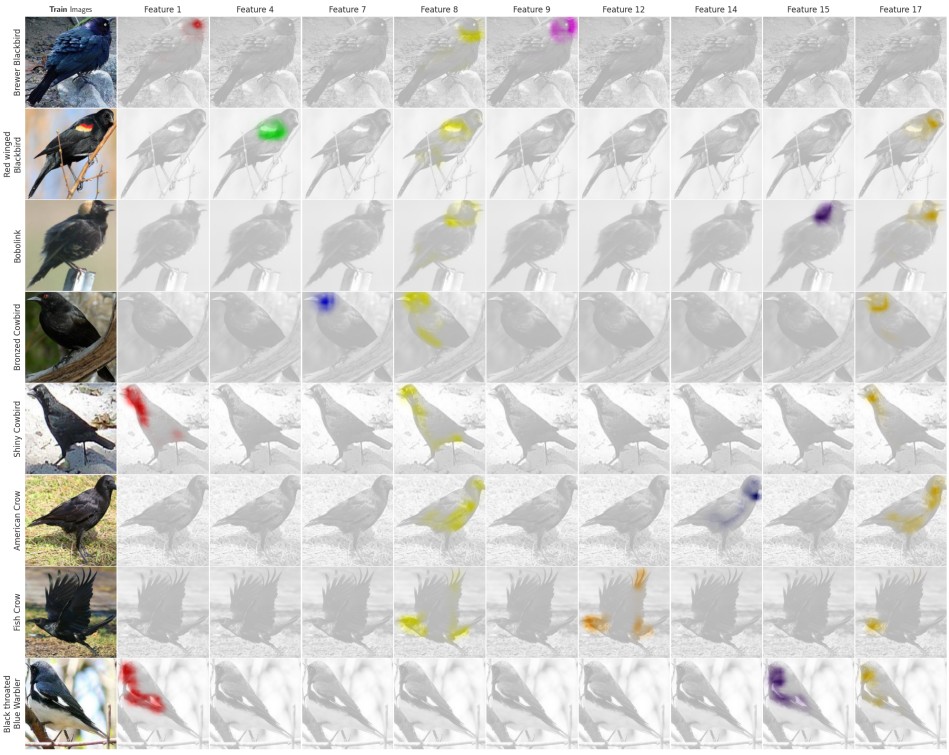

Figure 31: Visualizations for classes similar to Shiny Cowbird using the model with 3 features per class, *e.g.* explained in Figures 1, 2, 14 and 21.

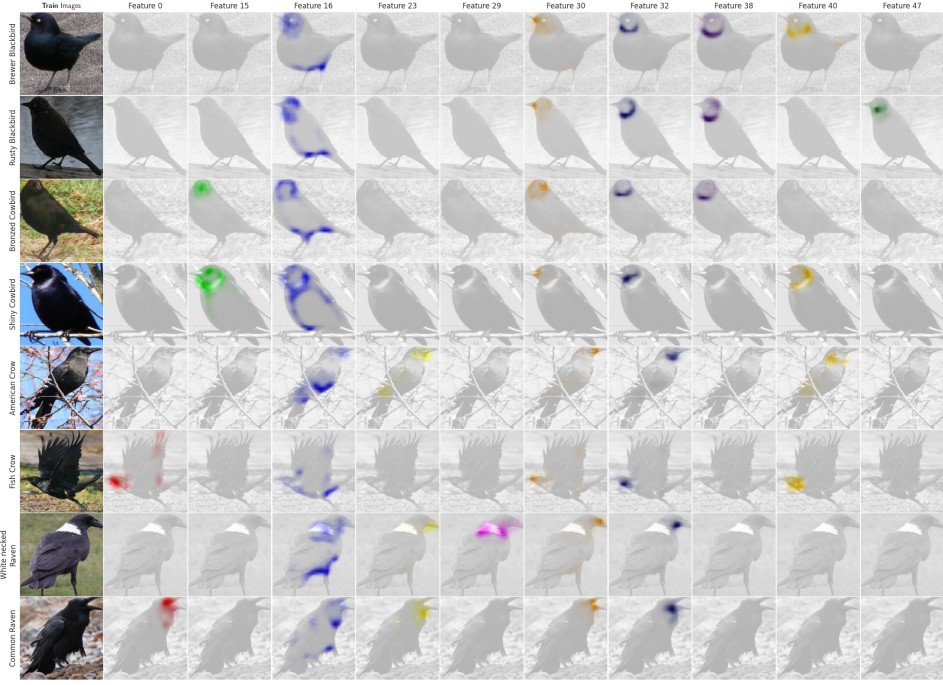

Figure 32: Visualizations for classes similar to Shiny Cowbird using the model with $n_{\mathrm{wc}} = 5$ features per class, *e.g.* explained in Figures 15, 16 and 25 to 28. We included the Raven classes from Figures 15, 25 and 26 for reference.

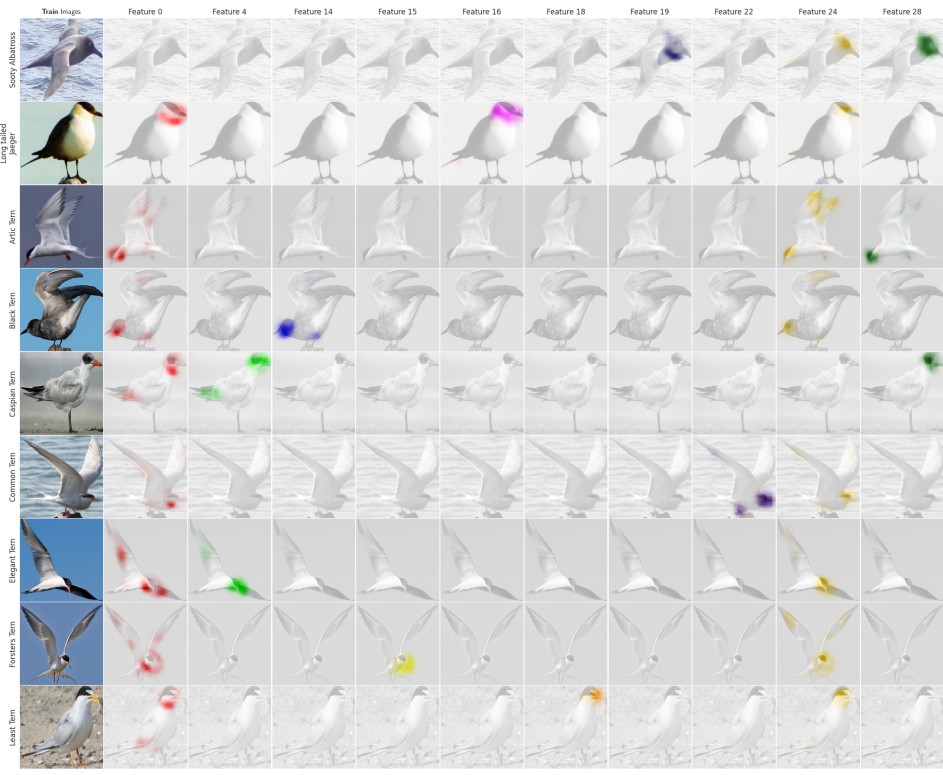

Figure 33: Visualizations for classes similar to Arctic Tern using the model with $n_{\mathrm{wc}} = 3$ features per class, *e.g.* explained in Figures 1, 2, 14 and 21.

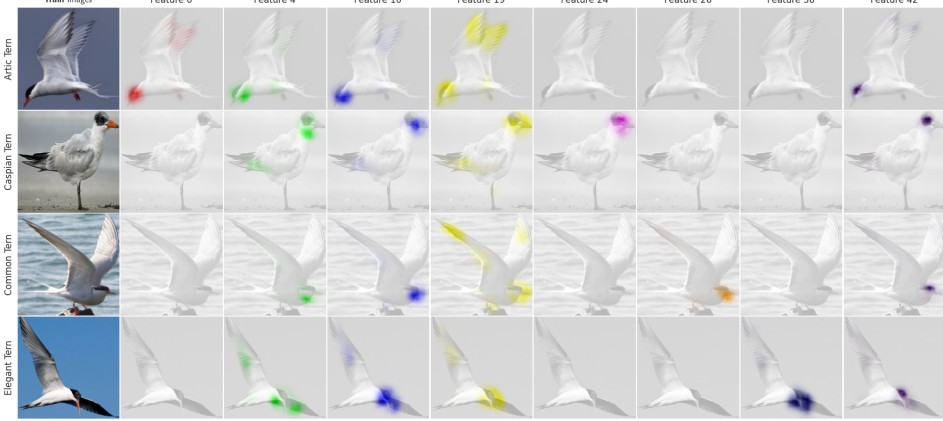

Figure 34: Visualizations for classes similar to Arctic Tern using the model with $n_{\mathrm{wc}} = 5$ features per class, *e.g.* explained in Figures 15, 16 and 25 to 28.

