# OpenReview forum: "CHiQPM: Calibrated Hierarchical Interpretable Image Classification"
_NeurIPS.cc/2025/Conference — NeurIPS 2025 poster_

### Official Review · Reviewer_7DuL · 2025-07-01

**Clarity:** 2
**Significance:** 2
**Originality:** 2
**Rating:** 3
**Confidence:** 4

**Summary:**

This paper proposes a globally and locally interpretable image classification model that builds upon QPM with improved contrastive class representations and hierarchical explanations. CHiQPM encourages class-wise feature sharing and introduces a built-in interpretable conformal prediction mechanism by traversing explanation trees formed from shared feature activations. Experiments on CUB, Stanford Cars, and ImageNet-1K demonstrate that CHiQPM maintains high accuracy while offering calibrated predictions.

**Questions:**

1. How is the interpretability of local features ensured? While contrastiveness is enforced, it does not imply semantic consistency. How do we know the activated features correspond to concepts meaningful to humans?
2. What is the generalization ability of the model structure? If applied to a complex dataset with highly abstract or noisy classes, how would CHiQPM perform? What happens if class pairs do not share enough features to construct a meaningful hierarchy?
3. How does CHiQPM compare with prototype methods like ProtoTree [Nauta et al., 2021]? What are the key advantages of your approach?

**Ethical Concerns:**

["NO or VERY MINOR ethics concerns only"]

**Final Justification:**

While CHiQPM offers a more flexible hierarchy for model explanations, it provides no guarantee on what these features actually represent. Therefore, I remain unconvinced and maintain my borderline reject assessment.

**Limitations:**

Yes.

**Paper Formatting Concerns:**

None.

**Quality:**

2

**Strengths And Weaknesses:**

**Strengths**
1. The paper aims to provide global explanations that are encouraged, especially the design of interpretable models rather than relying on post-hoc explanations.
2. The idea of inducing hierarchical explanation paths by enforcing shared features between similar classes is novel.
3. The paper is easy to follow, making the objectives accessible to a broad audience.

**Weaknesses**
1. The model heavily relies on learned mid-level features, which are not guaranteed to correspond to human-interpretable concepts. While the authors introduce a contrastive objective and grounding loss, these only encourage sparsity but not semantic alignment. This leaves the interpretation paths inherently fragile and possibly misleading. Moreover, relying on a small set of local features per class may hurt performance, especially if global context is important.
2. The proposed framework assumes that similar classes will share a few features. However, for more abstract or complex classification tasks, a larger number of features per class may be necessary. As the feature pool expands, feature overlap between classes may decrease, undermining the construction of hierarchical structures and weakening the interpretability claims.
3. The paper lacks analysis of how the depth and width of the explanation trees vary under different configurations (e.g., feature pool size or class granularity). Such ablations are crucial to understand how robust or shallow the constructed explanation paths truly are.

---

> ### Author Rebuttal · Authors · 2025-07-29
>
> Thank you for your review, acknowledging several strengths and bringing up important points to discuss. As some arguments build on each other, we would like to answer the questions with answers building on each other.
> > How does CHiQPM compare with prototype methods like ProtoTree [Nauta et al., 2021]? What are the key advantages of your approach?
>
> The distinction to prototypical methods is a crucial point so we would like to elaborate. CHiQPM offers global interpretability as it has interpretable class representations that are built from shared, general, contrastive and diverse features which empirically align well with human attributes on CUB-2011. Therefore, one can understand the class representations of a CHiQPM globally.
> For example, the CHiQPM explained in Figure 1 represents a Bronzed Cowbird as:
> * A Bird with a red eye
> * A Bird with head and belly of black birds
> * A Bird with the neck of black birds
>
> In contrast, Prototypical methods represent classes via prototypes which typically (barring PIP-Net) are exact patches from the training data. The crucial difference is that these prototypes are class specific. This originates from training the feature (or prototype) extractor with classification loss, while allowing it to learn at least one prototype per class. Consequently, the class representation of a prototypical method for a Bronzed Cowbird might be:
> * A Bronzed Cowbird
> * … (They typically have significantly more prototypes assigned per class. Depending on the model, the prediction either depends on just one class-specific feature, or all/most assigned prototypes are class-specific.)
>
> When judging prototype visualizations, humans typically suffer from confirmation bias and think that the model indeed represents the class with the human concept, while it is a class detector. This can be quantified via Class-Independence [35], with many prototypes of PIP-Net being only (> 99%) similar to samples of just one class. Additionally, human studies have shown that humans cannot predict the presence of such prototypes [16,19]. Figure 7 in the ProtoTree paper [31] itself illustrates this very effectively: When the height of the ProtoTree enables one prototype per class (height = 8; $2^8 = 256 > 200$ = number of classes in CUB-2011), the accuracy jumps to a competitive level and does not increase further with increasing height. Notably, even with more allowed prototypes at a height of 9, allowing $2^9 = 512$, ProtoTree still just learns about one per class with 201.6 (see Table 2 in the paper [31]). Finally, ProtoTree learns a deep decision tree, which might seem similar to our hierarchical explanations. However, as its features are not general, the usefulness is very limited.
>
> CHiQPM follows the line of Work of Q-SENN [34] and QPM [35], where classes are represented with shared general features instead of class-specific ones to enable global interpretability. Crucial elements for that are to learn less features than there are classes in the dataset and to ensure that features are equally important for multiple classes. Therefore, typically no more than 50 features are learned, while their weights to classes are binary in QPM and CHiQPM. That way, one feature on CUB has to activate on general, shared concepts of its assigned classes and cannot detect class-specific properties. As shown in [35], this strong restriction on the space of learnable features already causes the features to become class-independent and contrastive.
>
> We thank the reviewer for highlighting this crucial point. We agree that clarifying the distinction to prototypical methods is essential for understanding the significance of our work. We will add this detailed comparison to the appendix and emphasize the core differences in the main paper.
> > The model heavily relies on learned mid-level features, which are not guaranteed to correspond to human-interpretable concepts. While the authors introduce a contrastive objective and grounding loss, these only encourage sparsity but not semantic alignment. This leaves the interpretation paths inherently fragile and possibly misleading.
> >
> > How is the interpretability of local features ensured? While contrastiveness is enforced, it does not imply semantic consistency. How do we know the activated features correspond to concepts meaningful to humans?
>
> As we discussed in the limitations Section J, it can indeed be considered a limitation of CHiQPM that it is not enforced to learn human-interpretable concepts. Instead, it by-design learns general and contrastive features that classify the dataset well. However, the feature alignment metric quantifies that CHiQPM empirically does learn features that correspond to concepts meaningful to humans, with an average increased feature value of 3.8 times its mean when the most aligned concept is present (Table 3). It seems that the concepts that humans have annotated for the birds in CUB-2011 are among the most effective general and contrastive concepts, which is why CHiQPM learns to detect them.
> Notably, this metric also verifies the qualitative impression that the saliency maps of individual classes give. For example, the features 7,8 and 17 of Figure 1 indeed return high alignment values of 2.2, 3.9 and 2.5 for the attributes Red Eye Color, Black Belly Color and Black Throat Color their saliency maps indicate. Please note that the novel Feature Grounding Loss improves the alignment with these attributes, but as you correctly mentioned does not generally prefer the hard-to-define category of human-interpretable concepts, but rather further encourages general contrastive features.
>
> Finally, we would like to point out that restricting the learnt features to relevant concepts that are known to mankind a priori is not necessarily an advantage, as discussed in Point 3 of Section J. The open setting of learning general features that are shared (or the main difference) between classes could lead to scientific discovery, as one can learn where patterns are that are suited for differentiation.
> > Moreover, relying on a small set of local features per class may hurt performance, especially if global context is important.
>
> We acknowledge that it might be possible that the few features per class become too few in certain conditions. However, previous and this work show that the accuracy on already quite challenging datasets tends to saturate after 5 features per class. For reference, on CUB-2011, 3 features per class already seem sufficient for CHiQPM:
>
> | $n_{wc}$ | Accuracy |
> | :---: | :---: |
> | 3 | 85.2 |
> | 4 | 85.1 |
> | 5 | 85.3 |
> | 6 | 85.0 |
>
> We further want to clarify that we do not enforce local features, but rather bias the model towards learning such features via the Feature Diversity Loss and feature selection bias in the QP. However, if a more global feature is required, it is still learnt, such as Feature 8 in Figure 1. We would also like to point out that CHiQPM performs remarkably well on ImageNet, where the scale of features is drastically varying across images and prototypical models with prototypes restricted to one patch are generally not even converging. We will include this ablation in the appendix and add your point as limitation.
> > The proposed framework assumes that similar classes will share a few features. However, for more abstract or complex classification tasks, a larger number of features per class may be necessary. As the feature pool expands, feature overlap between classes may decrease, undermining the construction of hierarchical structures and weakening the interpretability claims.
> >
> > What is the generalization ability of the model structure? If applied to a complex dataset with highly abstract or noisy classes, how would CHiQPM perform? What happens if class pairs do not share enough features to construct a meaningful hierarchy?
>
> We consider ImageNet as already quite complex with very noisy labels [1] and deduce that CHiQPM’s predictive performance generalizes very well to such a complex dataset with just 5 features per class and 50 in total. However, as described in Section J, the interpretability of the features does not scale as well. For a more thorough discussion, please see our first response to Reviewer7M7N.
>
> As discussed in Section 3.1, 4.3 and D, the tree density should ideally be set using the calibration data, as it is hard to know how many class pairs share sufficient concepts. If that analysis determines that $\rho$ should be set to 0, the resulting CHiQPM will lose some of its interpretability, but will still be the most accurate model with global interpretability. Nevertheless, as mentioned in the Limitations (lines 750-753), if classes truly share nothing or are very few in total, CHiQPM is not applicable.
> > The paper lacks analysis of how the depth and width of the explanation trees vary under different configurations (e.g., feature pool size or class granularity). Such ablations are crucial to understand how robust or shallow the constructed explanation paths truly are.
>
> The depth of the explanation tree generally equals the number of features assigned to each class, as every activating feature is part of the tree. We understand the width of the tree as the number of leaf nodes in the tree and have measured that width of the total explaining tree as in Figure 23 and the tree below Level 1, as e.g. in Figure 2. As one expects, with increasing number of features, the resulting trees become smaller, as fewer classes are assigned to each individual feature. We will add this ablation to the appendix.
>
> | Number of Features  | 20 | 30 | 40 | 50 | 60 |
> | :--- | :---: | :---: | :---: | :---: | :---: |
> | # Leafs in total Tree | 73.9 | 64.8 | 57.9 | 53.5 | 50.2 |
> | # Leafs in Tree below Level 1 | 24.9 | 17.8 | 14.1 | 11.9 | 10.3 |
>
> We hope these clarifications and new results address your concerns.
>
> [1] Shankar, Vaishaal, et al. "Evaluating machine accuracy on imagenet." International Conference on Machine Learning. PMLR, 2020.

---

> ### Comment · Reviewer_7DuL · 2025-08-03
>
> Thanks to the authors for their detailed rebuttal. However, several concerns remain.
>
> 1. While I appreciate the clarification regarding the distinction between CHiQPM and prototype-based methods such as ProtoTree, the response revolves the depth and structure of the hierarchical explanations. However, I have found that many prototype networks have already explored hierarchical structures for global interpretability [1, 2, 3, 4]. Many of these approaches also construct semantic trees or hierarchies, and I struggle to see a clear and significant advantage of CHiQPM over these alternatives.
>
> 2. I fully agree with the authors that learning human-understandable features should not be seen as an inherent advantage. However, hierarchical structures are precisely valuable because they simplify the explanatory process for humans. Therefore, I find it contradictory that the paper highlights hierarchical interpretability as a major contribution, yet downplays the necessity of human interpretability when discussing feature semantics. If we argue that interpretability for human users is no longer essential, then the motivation for building structured and traversable explanations weakens. One may then question whether interpretability-focused frameworks are justified in the first place, or whether we should instead focus on purely robust and reliable predictive models.
>
> 3. I appreciate the additional results and clarifications. However, they seem to raise further concerns. For instance, the fact that prediction accuracy remains stable across different numbers of features suggests that the learned structure behaves more like a general-purpose feature learning module rather than a model with tightly constrained, explanation-driven representations. This undermines the core interpretability guarantee. While I acknowledge the general challenge of scaling such methods to complex datasets (e.g., ImageNet), this should also be explicitly discussed or solved.
>
> Overall, I encourage the authors to pursue this promising line of work further, and would appreciate more efforts that strike a good trade-off between model performance and strengthen interpretability guarantees, as well as more clearly show advantages over existing methods.
>
> [1] Hase, Peter, et al. "Interpretable image recognition with hierarchical prototypes." Proceedings of the AAAI conference on human computation and crowdsourcing. Vol. 7. 2019.
>
> [2] Yang, Peiyu, Zeyi Wen, and Ajmal Mian. "Multi-grained interpre table network for image recognition." 2022 26th International Conference on Pattern Recognition. IEEE, 2022.
>
> [3] Gulshad, Sadaf, Teng Long, and Nanne van Noord. "Hierarchical explanations for video action recognition." Proceedings of the IEEE/CVF Conference on Computer Vision and Pattern Recognition. 2023.
>
> [4] Yu, Zhen, et al. "Hierarchical skin lesion image classification with prototypical decision tree." npj Digital Medicine 8.1 (2025): 26.

---

> > ### Author Response · Authors · 2025-08-04
> > **Response from Authors (1/3)**
> >
> > Thank you for appreciating our effort and promptly raising your concerns to allow us to clarify.
> >
> > > While I appreciate the clarification regarding the distinction between CHiQPM and prototype-based methods such as ProtoTree, the response revolves the depth and structure of the hierarchical explanations. However, I have found that many prototype networks have already explored hierarchical structures for global interpretability [1, 2, 3, 4]. Many of these approaches also construct semantic trees or hierarchies, and I struggle to see a clear and significant advantage of CHiQPM over these alternatives.
> >
> > Thank you for pointing us to several relevant papers. We will follow your suggestion and add a discussion on models with a **fixed hierarchy** in the related work section. However, we would like to clarify that CHiQPM’s **dynamic hierarchy** is fundamentally different and has significant advantages over these alternatives while offering globally interpretable class representations:
> >
> > **Class Representations for Global Interpretability**
> > * CHiQPM offers global interpretability through its interpretable class representations. Every class is represented exclusively using the assignment of $n_{wc}$ features that quantifiably show several desired properties for interpretability.
> > * None of these papers offers such globally interpretable class representations. [1, 2] use fully-connected layers to do the actual classification, hence not learning a compact interpretable class representation. [4] learns one prototype per class. Similarly, [3] learns 10 prototypes per class on each level of the hierarchy. Thus, [3, 4] represent classes with themselves.
> >
> > To summarize the difference with an example:
> > CHiQPM can globally explain that it differentiates a Bronzed Cowbird from a Shiny Cowbird via its red eye, while an accurate prototypical method recognizes a Bronzed Cowbird by it being a Bronzed Cowbird or worse for fully-connected layers.
> >
> > **Hierarchy and Conformal Prediction**
> > * CHiQPM does not use any fixed hierarchy. Instead, it represents classes via equally weighted features. CHiQPM then provides local explanations with dynamic hierarchies, that are specific to a concrete test sample and answer an unprecedented range of questions (as also explained in rebuttal to Reviewer kBM9):
> >     1.  What meaningful features of which classes are found in an image? (Quantified by multiple metrics including Structural Grounding and Feature Alignment in Tables 1, 3, & 16.)
> >     2.  How does each feature narrow down the set of potential predictions into increasingly similar classes? (Visualized in our hierarchical explanation and measured by Set Coherence in Figure 6.)
> >     3.  Which set shall be predicted to guarantee a configurable average accuracy? (Guaranteed by our conformal prediction framework, with results in Table 2.)
> >     4.  Which features would have needed to activate stronger in order to predict a smaller set with sufficient certainty? (Demonstrated by the paths in our hierarchical explanations.) This notion of counterfactual interpretability has recently gotten more attention [A] and is included by-design.
> > * This work is the first to introduce a notion of interpretable conformal prediction, as CHiQPM traverses the hierarchical local explanation to predict a coherent set of classes with guarantees while being competitively efficient to conventional conformal prediction methods. We are not aware of any comparable work and want to emphasize that points 2 to 4 are only enabled through the dynamic hierarchy.
> >
> > * In contrast, [1, 2, 3, 4] organize classes in a fixed hierarchy that is either annotated [1, 3, 4] or constructed from data [2] and aim to provide global interpretability as you rightfully pointed out. The decision nodes in their fixed hierarchies are of the form “Is it a Bronzed Cowbird?” or “Is it any of these Seabirds?”, while CHiQPMs dynamic hierarchical explanation has inner nodes that mean: “This image has a red eye and therefore only this subset of classes is predicted with a red eye”.
> >
> > In summary: The dynamic hierarchical local explanations of CHiQPM are fundamentally different to the fixed class hierarchies of the papers you mentioned. CHiQPM’s local explanations transport certainty and answer several questions that can only be answered with them, most notably enabling the first form of efficient interpretable conformal prediction.
> >
> > [A] Dominici, Gabriele, et al. "Counterfactual Concept Bottleneck Models." ICLR. 2025.

---

> > > ### Author Response · Authors · 2025-08-04
> > > **Response from Authors (2/3)**
> > >
> > > > I fully agree with the authors that learning human-understandable features should not be seen as an inherent advantage. However, hierarchical structures are precisely valuable because they simplify the explanatory process for humans. Therefore, I find it contradictory that the paper highlights hierarchical interpretability as a major contribution, yet downplays the necessity of human interpretability when discussing feature semantics. If we argue that interpretability for human users is no longer essential, then the motivation for building structured and traversable explanations weakens. One may then question whether interpretability-focused frameworks are justified in the first place, or whether we should instead focus on purely robust and reliable predictive models.
> > >
> > > Thank you for agreeing with our argument of not having to restrict learning to prior known features. We would like to clarify that we do see human interpretability as an essential property of transparent models in deployment and agree that our hierarchical explanations can be valuable by simplifying the explanation process. CHiQPM’s hierarchical explanation uses its features as nodes. These features have the desired quantities (Diversity, Contrastiveness, Generality) that are generally considered beneficial for interpretability [29]. As discussed in Section J, Point 3, this makes them perfectly suitable to be thoroughly analyzed and cause the diverse, general and contrastive patterns they detect to become a concept that is interpretable for humans, as the neologisms discussed in [15]. Nevertheless, we would like to reiterate that on the CUB-2011 dataset, where quantitative measurement is possible, the learnt features are significantly more aligned with known human attributes than QPM’s features, showing that human understandable features are learnt without explicit supervision.
> > >
> > > To answer your more meta point of whether interpretability-focused frameworks are justified: We believe that work like ours with an elevated level of transparency and trustworthiness is crucial to responsibly enable certain high-stakes applications, with some even requiring transparent models under recent legislation. One perfect application for this work would be to support experts in high-stakes decision-making, e.g. doctors, because:
> > > * The budget would exist to thoroughly analyze the learnt general features.
> > > * Explanations are often required.
> > > * Global interpretability can ensure being right for the right reasons before deployment.
> > > * Conformal prediction is ideal for supporting experts [47,48].
> > > * Coherent Sets are preferred [8].
> > >
> > > Holistically, we follow [B], aim to build interpretable models by-design for high-stakes decisions and consider CHiQPM the frontier as it moves towards actual faithful interpretability rather than class-specific prototypes.
> > >
> > > [B] Rudin, Cynthia. "Stop explaining black box machine learning models for high stakes decisions and use interpretable models instead." Nature machine intelligence 1.5 (2019): 206-215.

---

> > > > ### Author Response · Authors · 2025-08-04
> > > > **Response from Authors (3/3)**
> > > >
> > > > > I appreciate the additional results and clarifications. However, they seem to raise further concerns. For instance, the fact that prediction accuracy remains stable across different numbers of features suggests that the learned structure behaves more like a general-purpose feature learning module rather than a model with tightly constrained, explanation-driven representations. This undermines the core interpretability guarantee. While I acknowledge the general challenge of scaling such methods to complex datasets (e.g., ImageNet), this should also be explicitly discussed or solved.
> > > >
> > > > You correctly note that CHiQPM is stable across different sparsities. However, we consider this a significant strength, as it achieves that while being massively constrained for interpretability. For example, at $n_{wc}=3$, only 50 features are sufficient to classify the dataset with each of them being equally important to on average 12 classes. For reference, QPM [35] drops below 84% accuracy with these constraints.
> > > >
> > > > In order to refute the consideration as “general-purpose feature learning module”, we want to emphasize that the model with $n_{wc}=3$ uses an entirely distinct assignment between features and classes, as the QP is solved optimally with a different constraint, resulting in different features being selected with a different assignment, which in turn results in different features emerging during fine-tuning as the shared general concept between the assigned classes.
> > > >
> > > > Finally, Feature Alignment increases (5.3 for $n_{wc}=3$, from 3.8 for $n_{wc}=5$) with decreasing number of features per class, as shown in our response to Reviewer TSHT. This quantifies that features even become more aligned, likely due to the concept shared between the assigned classes being less abstract.
> > > >
> > > > Regarding scalability, we would like to point out that we have already discussed it in Section J, but will add a dedicated Section as promised in our rebuttal to Reviewer 7M7N.
> > > >
> > > > We again thank you for your engagement and hope that these clarifications address your concerns.

---

> > > > > ### Comment · Reviewer_7DuL · 2025-08-09
> > > > >
> > > > > Thank you for the response. However, my main concerns remain. While CHiQPM offers a more flexible hierarchy for model explanations, it provides no guarantee on what these features actually represent. In contrast, tree-based prototype networks, though structurally fixed, can constrain interpretations within a human-interpretable scope. For example, in the case of distinguishing a Bronzed Cowbird from a Shiny Cowbird by its red eye, what evidence shows that CHiQPM’s feature corresponds to the red eye? The only support provided is activation maps between images, which, as noted in your cited work [B], are known to be less explainable and unreliable. Consequently, the model may rely on spurious or abstract patterns rather than meaningful, verifiable features. This undermines the validity of the explanations, especially in high-stakes applications where understanding what is being used to make a decision is critical. If the learned features themselves are opaque, then constructing a feature-based hierarchy on top of them raises further concerns. Moreover, prototype trees can also provide conformal prediction sets by leveraging similarity to parent prototypes (e.g., prototypical features like "red eye") to justify subclass predictions.
> > > > >
> > > > > While I agree that stability across different feature settings is beneficial for model performance, it does not address the core concern of an explainability guarantee. If the learned features are not semantically grounded, the resulting hierarchy lacks true interpretability. Without guarantees or concrete evidence for what the features represent, I remain unconvinced and maintain my borderline reject assessment.

---

> > > > > > ### Author Response · Authors · 2025-08-09
> > > > > > **Response from Authors**
> > > > > >
> > > > > > Thank you for your detailed final thoughts.
> > > > > > > For example, in the case of distinguishing a Bronzed Cowbird from a Shiny Cowbird by its red eye, what evidence shows that CHiQPM’s feature corresponds to the red eye?   The only support provided is activation maps between images, which, as noted in your cited work [B], are known to be less explainable and unreliable.
> > > > > >
> > > > > > We would like to point out the provided Feature alignment metric as evidence which exactly quantifies that specific features correspond to specific attributes without considering activation maps. This metric directly measures how much a feature's activation increases when a ground-truth human concept is present.  For the “red eye” example, feature 7 has an alignment of 2.2 with the ground-truth attribute.
> > > > > > Furthermore, as shown in our rebuttal to Reviewer 7M7N, CHiQPM’s activation maps are quantifiably more faithful than those of the black-box baseline, focusing 83.4% of their Grad-CAM explanation on the relevant object vs. 75.8% for the baseline.
> > > > > >
> > > > > >
> > > > > >
> > > > > >
> > > > > >
> > > > > > > Moreover, prototype trees can also provide conformal prediction sets by leveraging similarity to parent prototypes (e.g., prototypical features like "red eye") to justify subclass predictions.
> > > > > >
> > > > > > To our knowledge, no existing prototype-based method is able to generate coherent and efficient prediction sets with coverage guarantees like ours.
> > > > > > Additionally, we must emphasize this fundamental difference again: CHiQPM learns general concepts, like “red eye”, while prototypical methods rather learn “(eye of a) Bronzed Cowbird”.

---

### Official Review · Reviewer_TSHT · 2025-07-02

**Clarity:** 3
**Significance:** 3
**Originality:** 3
**Rating:** 5
**Confidence:** 4

**Summary:**

QPM introduce a way to make convolutional networks globally interpretable by representing each class as the logical "or" of a very small, fixed set of feature maps.

This yields ultra‐compact, class‐wise explanations but lacks fine‐grained, instance‐level detail and makes no provision for calibrated uncertainty in predictions. CHiQPM, builds directly on QPM:

(a) The discrete quadratic program is augmented to force many class pairs to share nwc – 1 features, creating a tree‐like hierarchy of explanations rather than a flat, per‐class OR.

(b) After solving the QP, the model is fine-tuned with a novel Feature Grounding Loss and ReLU-based regularization that:

(i) equalizes feature activation strength within each class representation and

(ii) suppresses spurious activations, dramatically improving how well features align to human‐understandable concepts.


(c) By traversing the learned hierarchy and using a tailored nonconformity score, CHiQPM produces calibrated prediction sets whose size meets a desired coverage target while keeping the returned classes semantically coherent (e.g. birds of similar shape), a capability missing in both QPM and flat set predictors.

Experiments on CUB‑2011, Stanford Cars and ImageNet‑1K across four backbones show (a) accuracy kept similar to a dense CNN (b) higher structural grounding than QPM.

**Questions:**

1. Are there other hierarchy construction choices?
2. Grounding is measured via concept annotations; however, do humans actually find the hierarchical paths helpful? A small-scale user study (e.g 20 domain experts evaluating 100 samples) could validate the claimed usability of the explanations.
3. What is the sensitivity to λ_feat and maximum depth of the hierarchy? Please add a grid.
4. Are there limitations regarding class imbalance?

**Ethical Concerns:**

["NO or VERY MINOR ethics concerns only"]

**Final Justification:**

Authors response is great, my rating doesn't change.

**Limitations:**

yes

**Quality:**

3

**Strengths And Weaknesses:**

Strengths:
1. CHiQPM builds on QPM’s adds a hierarchical tree structure, giving both a concise global summary and sample-specific decision paths.
2. The discrete QP formulation forces most class pairs to share nwc – 1 features, yielding an explanation hierarchy rather than a flat feature assignment.
3. Introducing L₍feat₎ plus a ReLU-based fine-tuning step sharply improves alignment between learned features and human concepts.
4. Strong experiments

Weaknesses:
1. Point‐accuracy improvements over QPM are very small and even negative on some backbones, limiting appeal outside interpretability‐focused audiences.
2. Dense notation (e.g. reusing “M” for multiple matrices) and scattered rationale for key design decisions (why ReLU vs. soft-threshold) hinder readability.

---

> ### Author Rebuttal · Authors · 2025-07-29
>
> Thank you for your positive review, acknowledging several strengths and posing interesting questions. We will of course try to improve readability at the points mentioned and answer each question separately.
>
> > Are there other hierarchy construction choices?
>
> Our used hierarchy based on order of the features of the predicted class has significant advantages compared to other choices. Its sample specific nature lends itself to a local explanation, as one can consider the order of the features as how dominant they appear in the image. Additionally, that enables traversing up the hierarchy in a meaningful way, since the least certain features get omitted first, causing accurate and efficient sets at each step. We believe that the main alternative is a class specific fixed order of features, as that would enable a global hierarchical explanation for each class. The comparison between a fixed order of features for each class, computed based on the average order on training data and our prediction up the hierarchy is shown below. Evidently, predicting with such a fixed order causes less accurate inefficient sets compared to our sample specific dynamic hierarchy. For example, our dynamic hierarchy reaches 91.6% accuracy with an average set size of 2.1, while the fixed hierarchy never reaches that accuracy. We believe that this is likely due to the unique feature, e.g. the red eye of the Bronzed Cowbird, being on average quite important to that class and thus causing prediction sets that are too specific to the top-1 prediction. This experiment showcases the superiority of using our sample specific dynamic hierarchy and will therefore be added to the appendix, as it further validates our method. Many thanks for this suggestion which is further strengthening our contribution.
>
> Comparison to fixed order of features per class based on average order of Features:
>
> **Sample Specifc Dynamic Feature order as used by CHiQPM:**
>
> | | Level 1 | Level 2 | Level 3 | Level 4 | Level 5 |
> | :--- | :---: | :---: | :---: | :---: | :---: |
> | **Set Accuracy** | 96.5   | 93.8   | 91.6  | 89.5   | 85.3   |
> | **Set Size** | 20.17| 4.43 | 2.12 | 1.45 | 1.00 |
>
> **Static Feature Order based on Average Order of Features on Training Data for each class.**
>
> | | Level 1 | Level 2 | Level 3 | Level 4 | Level 5 |
> | :--- | :---: | :---: | :---: | :---: | :---: |
> | **Set Accuracy** | 89.7 | 87.4|86.2 |85.6 | 85.2 |
> | **Set Size** | 18.52 | 2.85 | 1.30 | 1.09 | 1.00 |
>
> > Grounding is measured via concept annotations; however, do humans actually find the hierarchical paths helpful? A small-scale user study (e.g 20 domain experts evaluating 100 samples) could validate the claimed usability of the explanations.
>
> While we agree with the reviewer that such a user study would be very valuable, we are unfortunately unable to accommodate that in this timeframe. We will, however, add it to a newly added future Work section in the appendix.
>
> > Are there limitations regarding class imbalance?
>
> Thank you for this intriguing question. We believe that CHiQPM can be more robust to class imbalance than baseline methods for two reasons:
>
> 1. The quadratic problem is solved with no weighting based on prevalence of the class in the training data.
> 2. As underrepresented classes get the same number of feature assignments, the class will be predicted if all its general features are activating.
>
> However, for that to work, the assignment between features and classes has to be good. When some classes are underrepresented, the constants in the QP might not be good enough to obtain an assignment well suited for fine-tuning. Empirically, these effects seem to balance out, indicating no quantifiable limitation regarding class imbalance. We calculated the Pearson correlation coefficient between number of training samples and test accuracy for each class on the slightly imbalanced datasets Stanford Cars and ImageNet. As expected, there is a positive correlation between the number of training samples on both datasets, with a negligible reduction for CHiQPM. We will add this discussion and experiment to the appendix. Additionally, we add robustness to class imbalance as a direction for future work as we believe in the aforementioned potential. Future work might include considering the number of samples per class during the QP as a form of certainty or use some additional form of supervision, e.g. via the class names, to make the constants more robust.
>
> **Pearson Correlation Coefficient between number of training samples and test accuracy per class:**
>
> | Model | Stanford Cars | ImageNet |
> | :--- | :---: | :---: |
> | **Dense** | 0.12 | 0.18 |
> | **CHiQPM** | 0.11 | 0.17 |
>
> > What is the sensitivity to $\lambda_{feat}$ and maximum depth of the hierarchy? Please add a grid.
>
> Thank you for that question, as we indeed missed including the ablation. The maximum depth of the hierarchy is the number of features assigned to each class $n_{wc}$. Therefore, we swept $n_{wc}$ and $\lambda_{feat}$ and report accuracy, sparsity and alignment. As generally expected, $\lambda_{feat}$ improves alignment and sparsity across various sparsities. Additionally, features become sparser and more aligned with decreasing $n_{wc}$, as they must activate on fewer samples and can therefore learn less abstract shared concepts. We gladly add these grids for their valuable insights to the appendix.
>
> **Accuracy:**
>
> | $n_{wc}$ \ $\lambda_{feat}$ | 0.031 | 0.1 | 0.31 | 1.0 | 3.0 | 10.0 | 30.0 |
> | :--- | :---: | :---: | :---: | :---: | :---: | :---: | :---: |
> | **6** | 85.2 | 85.2 | 85.1 | 85.2 | 85.0 | 85.1 | 84.5 |
> | **5** | 85.3 | 85.1 | 85.2 | 85.2 | 85.3 | 85.2 | 84.5 |
> | **4** | 85.2 | 85.1 | 85.1 | 84.9 | 85.1 | 85.1 | 84.8 |
> | **3** | 84.9 | 84.7 | 84.8 | 84.9 | 85.2 | 85.3 | 84.5 |
>
> **Sparsity:**
>
> | $n_{wc}$ \ $\lambda_{feat}$ | 0.031 | 0.1 | 0.31 | 1.0 | 3.0 | 10.0 | 30.0 |
> | :--- | :---: | :---: | :---: | :---: | :---: | :---: | :---: |
> | **6** | 42.6 | 41.0 | 35.5 | 29.0 | 24.8 | 23.5 | 25.0 |
> | **5** | 38.8 | 37.2 | 32.6 | 26.4 | 22.3 | 21.2 | 22.9 |
> | **4** | 34.8 | 33.4 | 29.2 | 23.7 | 19.8 | 18.5 | 20.3 |
> | **3** | 30.8 | 29.2 | 26.0 | 20.7 | 17.1 | 15.8 | 17.3 |
>
> **Feature Alignment:**
>
> | $n_{wc}$ \ $\lambda_{feat}$ | 0.031 | 0.1 | 0.31 | 1.0 | 3.0 | 10.0 | 30.0 |
> | :--- | :---: | :---: | :---: | :---: | :---: | :---: | :---: |
> | **6** | 2.8 | 2.9 | 3.1 | 3.3 | 3.5 | 3.6 | 3.7 |
> | **5** | 3.1 | 3.2 | 3.4 | 3.7 | 3.8 | 3.9 | 4.0 |
> | **4** | 3.6 | 3.7 | 3.9 | 4.3 | 4.5 | 4.6 | 4.6 |
> | **3** | 4.3 | 4.4 | 4.6 | 5.0 | 5.3 | 5.3 | 5.3 |

---

### Official Review · Reviewer_7M7N · 2025-07-02

**Clarity:** 1
**Significance:** 3
**Originality:** 3
**Rating:** 5
**Confidence:** 3

**Summary:**

This work introduces the Calibrated Hierarchical QPM (CHiQPM). The strength of CHiQPM lies in its ability to provide both global and local explanations. The model employs contrastive explanations for majority classes and offers hierarchical explanations that align with human reasoning; a unique feature that also enables interpretable conformal prediction (CP) through traversable decision paths. Empirical results demonstrate that CHiQPM achieves near-state-of-the-art accuracy (retaining 99% of non-interpretable models' performance) while maintaining interpretability.

**Questions:**

1. Given the proposed method can do both global and local explanations; if the author consider to compare with some common XAI benchmarks, e.g., OpenXAI.

2. Please provide some discussions around scalability of the proposed approach.

3. In practice, can the authors give some examples of how a conformal set of features is better than, say, the use of top-1 or top-k features for decision-making.

**Ethical Concerns:**

["NO or VERY MINOR ethics concerns only"]

**Final Justification:**

The authors addressed my concerns in their response and thus I increase my score.

**Limitations:**

yes

**Paper Formatting Concerns:**

the line spaces between paragraphs seems to be changed; e.g., line 57-58.

**Quality:**

3

**Strengths And Weaknesses:**

Pros:
1. the idea of integrating global, local XAI and CP apears to be novel.
2. the paper is technically sound.

Cons:
1. while the overall idea is novel, the work is incremental to the original QPM work.
2. the writing is a bit hard to follow if without prior knowldege of QPM (i.e., not self-contained), I have to quick read the QPM paper as well.
3. the evaluation miss some common metrics for local XAI, e.g., delete/insert scores, Pointing Game;

---

> ### Author Rebuttal · Authors · 2025-07-29
>
> Thank you for your review and appreciating the novelty and soundness of our paper. We also thank you for the valuable feedback regarding clarity. We will revise the paper to make it more self-contained, particularly by providing more background on QPM [1] and how it differs from prototypical methods, as also suggested by Reviewer 7DuL. Regarding the questions:
>
> > Please provide some discussions around scalability of the proposed approach.
>
> We gladly elaborate on the scalability of CHiQPM. As shown in Table 1, our model scales exceptionally well to the large-scale ImageNet dataset, achieving a remarkable accuracy of just 0.8 percentage points less than the dense, non-interpretable baseline. This performance is achieved using just 50 features for all 1000 classes.
> As we described in Section J, scaling to a dataset of this magnitude introduces certain limitations for interpretability. For instance, the learned concepts become more abstract, saliency maps less faithful, and some features exhibit polysemanticity.  Nevertheless, our results show that CHiQPM successfully identifies the high-level features needed for accurate classification.
> Finally, the time it takes to solve the QP scales exponentially with the number of features to select [1]. But remarkably, just 50 features are sufficient for the very complex ImageNet dataset, making this a very manageable limitation in practice.
> In conclusion, CHiQPM demonstrates robust scalability. While its interpretability does not scale as excellently as its accuracy, it still offers the best global interpretability on ImageNet. We will summarize these limitations in scaling in a dedicated section in Section J.
>
> > the evaluation miss some common metrics for local XAI, e.g., delete/insert scores, Pointing Game;
> >
> > Given the proposed method can do both global and local explanations; if the author consider to compare with some common XAI benchmarks, e.g., OpenXAI.
>
> We would like to point out that we do not introduce a new saliency map method whose faithfulness should certainly be evaluated using delete/insert or similar [2] scores, that alleviate some OOD Problems of the former metrics. Instead, the proposed method mainly offers improved local interpretability as one can provide saliency maps via any of the available methods for each of the individually meaningful, general and diverse features separately, as well as via the novel hierarchical explanations.
> However, as visualizations of the activations of individual features are shown, it is a good idea to evaluate their faithfulness. Following your suggestion, we evaluate a form of the Pointing Game, similar to the initial form in [3], using segmentation masks provided for CUB200 [4]. Specifically, we calculate the fraction of the activation of the GradCAM saliency map that is focused on the segmented bird S.
>
> $\phi = \frac{GradCAM}{\sum GradCAM} \odot S$
>
> As the bird can be considered the region of the image responsible for the classification, a higher overlap with the segmentation indicates that the saliency maps localize more faithfully. Importantly, one would not necessarily expect an overlap of 100%, as the edge region is relevant to describing the shape, causing activations both on and off the segmentation. Across the entire test dataset, CHiQPM’s GradCAM focusses to 83.4% on the bird, whereas the Dense baseline only does so with 75.8%. Hence, CHiQPM has activation maps with improved faithfulness on CUB-2011, validating their use for saliency maps of individual interpretable features. We will include this experiment in the appendix.
>
> | Model  | Accuracy | Average $\phi$ = GradCAM Overlap with Segmentation |
> |:-------|:--------:|:--------------------------------------------------:|
> | Dense  | 86.6%    | 75.8%                                              |
> | CHiQPM | 85.3%    | 83.4%                                              |
>
> > In practice, can the authors give some examples of how a conformal set of features is better than, say, the use of top-1 or top-k features for decision-making.
>
> We gladly elaborate on why using our approach is preferential to using top-k features for decision-making. We understand “conformal set of features” as the set of features $\hat{F}_i^c$ (Eq. 8), from which the descending classes in the hierarchical explanation are predicted, e.g. {Green, Blue} in Figure 2. There are two main differences to just using top-k features for decision-making.
>
> 1. The conformal set of features is determined via conformal prediction which is considered an effective tool for supporting experts [47,48]. The level in the hierarchy is determined dynamically for each test sample via our novel interpretable conformal prediction method to guarantee that the target coverage is met. That way, the conformal set of features also transports certainty via its set size. In contrast, predicting with a fixed k, e.g. top-1, also returns large inefficient sets, when the test sample can be easily classified.
> 2. Practically, using top-k features is not suitable for building a hierarchy, as those top-k features might not belong to any one class together. This also causes top-k to be slightly less efficient than predicting with a fixed level n in the sample specific hierarchy. To demonstrate this, we compare between predicting with top-k features and at level n.
>
> |                | Level 1  | Level 2 | Level 3 | Level 4 | Level 5 |
> |:---------------|:--------:|:-------:|:-------:|:-------:|:-------:|
> | **Set Accuracy** | 96.5   | 93.8   | 91.6  | 89.5   | 85.3   |
> | **Set Size** | 20.17  | 4.43  | 2.12  | 1.45 | 1.00  |
>
> |                  | Top-1   | Top-2  | Top-3  | Top-4  | Top-5  | Top-6  | Top-7  | Top-8  |
> |:-----------------|:-------:|:------:|:------:|:------:|:------:|:------:|:------:|:------:|
> | **Top-K Accuracy** | 96.3  |93.2 | 90.8 | 89.0 | 85.5 | 85.4 | 85.3 |85.3 |
> | **Set Size** | 20.17 | 4.48 | 2.19 | 1.51 | 1.08 | 1.02 | 1.01| 1.01 |
>
> Note that predicting with top-1 feature will always return the large set of all classes with that feature which is even less accurate than predicting at Level 1 as there simply might be no class that has the remaining dominant features apart from Top-1. A concrete example can illustrate this failure case: A male bird of the species “Scarlet Tanager” is red, while females are olive-green. The feature "red" might be the top activated feature for a male, but CHiQPM may not use "red" to represent the class Scarlet Tanager because it's not a consistent feature. Thus, relying on the top-1 feature "red" would lead to an incorrect prediction. We will add this experiment and discussion to the appendix.
>
> We again thank you for your review and suggestions that strengthen our paper.
>
> [1]: Norrenbrock, Thomas, et al. "QPM: Discrete Optimization for Globally Interpretable Image Classification." The Thirteenth International Conference on Learning Representations. 2025.
>
> [2]: Zheng, Xu, et al. "F-Fidelity: A Robust Framework for Faithfulness Evaluation of Explainable AI." The Thirteenth International Conference on Learning Representations. 2025.
>
> [3]: Zhang, Jianming, et al. "Top-down neural attention by excitation backprop." International Journal of Computer Vision 126.10 (2018): 1084-1102.
>
> [4]: Farrell, R. (2022). CUB-200-2011 Segmentations (1.0) [Data set]. CaltechDATA.

---

### Official Review · Reviewer_kBM9 · 2025-07-03

**Clarity:** 3
**Significance:** 2
**Originality:** 2
**Rating:** 4
**Confidence:** 2

**Summary:**

The paper introduces Calibrated Hierarchical QPM (CHiQPM), an interpretable image classification framework that merges the compact global explanations of Quadratic Prototype Models (QPM) with *hierarchical* local explanations and built-in *conformal prediction* (CP).
At training time, a dense backbone network is first diversified, after which a discrete quadratic program with new hierarchy constraints selects a small, fixed set of concept-like features per class. Those features are then fine-tuned with a Feature Grounding Loss and an additional ReLU so that each activates only on a single, human-aligned visual concept. Finally, the model is *calibrated* by CP, enabling it to traverse its sample-specific hierarchy and output coherent class sets.

**Questions:**

## Questions
- Can you provide formal statements for the proposed method as raised in the above?
- Justification of hyper-parameter selection is needed. Can you give more discussion?
- Can you state whether the reported experimental settings satisfy the assumption? and can you provide several counterexamples?

## Additional suggestions
- Matrix \textit{M} denotes a feature map, but the letter $M$ is reused for class-class similarity, risking confusion.
- L655: “calibrated cohrerent set predictions” → coherent
- L839: “These visuzalizations demonstrate…” → visualizations
- “we temporarily save every model … to speed up metric **calculation**.” The preceding sentence says *metrics* (plural). Singular/plural mismatch.
- “auxiliary Feature Diversity Loss $\mathcal{L}_{\text{div}}$ [33]” – no definition or formula is given in the current manuscript. Readers must look up ref. [33].

**Ethical Concerns:**

["NO or VERY MINOR ethics concerns only"]

**Final Justification:**

Although theoretical concerns remain, other points were addressed by the rebuttal.

**Limitations:**

yes

**Quality:**

2

**Strengths And Weaknesses:**

## Strengths
- The proposed pipeline is easy to follow.
- Non-trivial empirical gains while keeping backbone accuracy.
- Interpretable hierarchy visualizations and qualitative examples. Figures illustrate that the learned tree edges correspond to semantic super-/sub-classes and that prediction sets indeed trace these edges upward when uncertainty is high.

## Weaknesses
- Essential steps lack any formal analysis. There is no approximation bound, convergence argument, or sample-complexity discussion. Theoretical novelty is therefore thin.
- Since the proposed method appears to be somewhat sensitive to a hyper-parameter $\rho$, theoretical discussions on the optimality of hyper-parameter selection strengthen this study. Moreover, the authors set $\lambda_{\text{feat}} = 3$ in their numerical experiments, but there are no discussion on how to choose this value, and no justification.
- Exchangeability is assumed in the manuscript, but there is no discussion as to when this assumption is satisfied and when it is violated (how realistic an assumption it is). At least there should be a discussion of the assumptions made with respect to the experimental setup reported by the authors.

---

> ### Author Rebuttal · Authors · 2025-07-29
>
> Thank you for your review, appreciating several strengths and raising several questions that allow us to clarify our contributions. We also thank you for the detailed suggestions on the text and will of course correct the typos, rename the class-class similarity to an unused symbol, and include the formula for the Feature Diversity Loss in the revised manuscript.
>
> > Essential steps lack any formal analysis. There is no approximation bound, convergence argument, or sample-complexity discussion. Theoretical novelty is therefore thin.
> >
> > Can you provide formal statements for the proposed method as raised in the above?
>
> Our work follows other work in interpretable machine learning, such as PIP-Net, ProtoPool, Q-SENN or QPM, where the primary contribution is a model that offers novel forms of interpretability that are thoroughly empirically validated. As such, the paper’s main contribution is not theoretical. Instead, our main novelty is the introduction of hierarchical explanations and the adaptation of the theoretically well-grounded theory of conformal prediction to predict along them. This enables the first form of built-in interpretable conformal prediction and local explanations that answer an unprecedented range of questions simultaneously while having non-trivial empirical gains (as rightly pointed out):
>
> 1.  What meaningful features of which classes are found in an image? (Quantified by multiple metrics including Structural Grounding and Feature Alignment in Tables 1, 3, & 16.)
> 1. How does each feature narrow down the set of potential predictions into increasingly similar classes? (Visualized in our hierarchical explanation and measured by Set Coherence in Figure 6.)
> 1. Which set shall be predicted to guarantee a configurable average accuracy? (Guaranteed by our conformal prediction framework, with results in Table 2.)
> 1. Which features would have needed to activate stronger in order to predict a smaller set with sufficient certainty? (Demonstrated by the paths in our hierarchical explanations.) This notion of counterfactual interpretability has recently gotten more attention [1] and is included by-design.
>
> Note that questions 2.-4. are only enabled through our novel contributions, while the answer to 1. is greatly improved compared to what QPM offers.
> Alongside the improved global interpretability, we consider answering these questions in an explanation a significant step to advance the field of interpretable machine learning. We will include this phrasing and the recent counterfactual paper as we believe that it clarifies our contribution and why it is significant without novel formal statements.
>
> > Exchangeability is assumed in the manuscript, but there is no discussion as to when this assumption is satisfied and when it is violated (how realistic an assumption it is). At least there should be a discussion of the assumptions made with respect to the experimental setup reported by the authors.
> >
> > Can you state whether the reported experimental settings satisfy the assumption? and can you provide several counterexamples?
>
> You correctly pointed out that we assume exchangeability between calibration data and test data for our conformal prediction experiments, which is required for the guarantees of Split Conformal Prediction to hold. We argue that exchangeability is certainly holding for our evaluation, as we
>
> 1. Calibrate on the first 10 samples per class of the test data (lines 325-326). This ensures the calibration and test samples are i.i.d., a stronger condition than exchangeability.
> 2. Very closely match the coverage on the test data for which we calibrate, indicating that the test was exchangeable with the calibration data.
>
> We will explicitly state that we assume exchangeability in the main paper and reference an expanded section on exchangeability in Section J, which contains the surrounding clarification. As our experiments are designed to satisfy exchangeability, there is no counterexample in our experiments. However, there exists work [2, 3, 4] on applying split conformal prediction under conditions where exchangeability does not hold. Investigated scenarios without exchangeability are time series datasets [2, 3] or calibrating on ImageNet and then testing on datasets with distribution shift [4], such as ImageNetR or ImageNetA.
>
> > Since the proposed method appears to be somewhat sensitive to a hyper-parameter $\rho$, theoretical discussions on the optimality of hyper-parameter selection strengthen this study. Moreover, the authors set $\lambda_{feat}=3$ in their numerical experiments, but there are no discussion on how to choose this value, and no justification.
> >
> > Justification of hyper-parameter selection is needed. Can you give more discussion?
>
> We thank the reviewer for the question about selecting $\rho$ and are happy to clarify our reasoning. As described in Sections 3.1, 4.3 and D, $\rho$ should be set using the calibration data and depends on several factors, most notably how many classes are similar in the dataset and if a specific coverage is targeted. Varying coverage goals relate to real-world scenarios, since a bird classifying app can be calibrated for a lower coverage, while a medical application should certainly guarantee smaller error rates.
> We selected $\rho=0.5$. As shown in our ablation studies (Figures 7 and 8), this value represents a 'sweet spot' for the CUB-2011 dataset. It is high enough to induce a rich semantic hierarchy that improves set prediction performance, yet low enough that it does so without sacrificing point-prediction accuracy. However, we will extend Section D to emphasize that there is no single optimal $\rho$ for all CHiQPMs.
>
> You are right to point out that an ablation on $\lambda_{feat}$ is missing. It was chosen based on the initial experiment on CUB-2011 shown below showcasing Accuracy, Sparsity and Feature Alignment with respect to the weighting. Evidently, alignment and sparsity improve with increasing $\lambda_{feat}$. The hyperparameter was set to the highest value for which the accuracy does not decrease, hence resulted in $\lambda_{feat}=3$. We will include this ablation in the appendix.
>
> | Metric \ $\lambda_{feat}$ | 0.031 | 0.1  | 0.31 | 1.0  | 3.0  | 10.0 | 30.0 |
> | :-------------------- | :---: | :--: | :--: | :--: | :--: | :--: | :--: |
> | Accuracy              | 85.3  | 85.1 | 85.2 | 85.2 | 85.3 | 85.2 | 84.5 |
> | Feature Sparsity      | 38.8  | 37.2 | 32.6 | 26.4 | 22.3 | 21.2 | 22.9 |
> | Feature Alignment     | 3.1   | 3.2  | 3.4  | 3.7  | 3.8  | 3.9  | 4.0  |
>
>
> We thank you again for pointing out valid points of criticism and believe that the resulting changes will improve the manuscript.
>
> [1] Dominici, Gabriele, et al. "Counterfactual Concept Bottleneck Models." ICLR. 2025.
>
> [2] Barber, Rina Foygel, et al. "Conformal prediction beyond exchangeability." The Annals of Statistics 51.2 (2023): 816-845.
>
> [3] Oliveira, Roberto I., et al. "Split conformal prediction and non-exchangeable data." Journal of Machine Learning Research 25.225 (2024): 1-38.
>
> [4] Alijani, Shadi, and Homayoun Najjaran. "WQLCP: Weighted Adaptive Conformal Prediction for Robust Uncertainty Quantification Under Distribution Shifts." Proceedings of the Computer Vision and Pattern Recognition Conference. 2025.

---

> > ### Comment · Reviewer_kBM9 · 2025-08-04
> >
> > Thank you for the carefully written rebuttal. Below I comment point-by-point on the clarifications you provided.
> > ### 1. Lack of formal analysis / theoretical novelty:
> > You state that CHiQPM, like prior prototype models, is primarily an engineering contribution. I understand the historical positioning, yet without a proof that ascending the tree preserves coverage, it lacks justification of the proposed method.  Moreover, the core optimization problem introduces new hierarchy constraints whose feasibility and approximation quality remain unanalyzed. I therefore still regard the absence of even a minimal formal treatment as a serious shortcoming.
> > ### 2. Exchangeability assumption:
> > The clarification that calibration samples are the first ten images after shuffling resolves my main practical concern. Please state it explicitly in the manuscript.
> > ### 3. Hyper-parameter selection:
> > The new ablation table is helpful and should be added to the appendix.
> >
> > Again, I appreciate the clarifications by the authors.

---

> > > ### Author Response · Authors · 2025-08-05
> > > **Response from Authors (1/2)**
> > >
> > > Thank you for your engagement and giving us the opportunity to clarify one crucial point regarding our conformal prediction contribution. We are glad our clarifications on exchangeability and hyperparameters were helpful and will incorporate them following your suggestions.
> > >
> > >
> > >
> > > > ### 1. Lack of formal analysis / theoretical novelty:
> > > > You state that CHiQPM, like prior prototype models, is primarily an engineering contribution. I understand the historical positioning, yet without a proof that ascending the tree preserves coverage, it lacks justification of the proposed method. Moreover, the core optimization problem introduces new hierarchy constraints whose feasibility and approximation quality remain unanalyzed. I therefore still regard the absence of even a minimal formal treatment as a serious shortcoming.
> > >
> > >
> > >
> > > Thank you for this critical question, as it allows us to clarify that our conformal prediction method that ascends the hierarchical local explanation guarantees coverage by **inheriting guarantees from the proven framework of Split Conformal Prediction [37]**.
> > >
> > > We understand
> > >
> > > > ascending the tree preserves coverage
> > >
> > >
> > >
> > > to question whether our conformal prediction method actually guarantees that the specific prediction sets in the hierarchical explanations reach the coverage CHiQPM is calibrated for.
> > > To clarify this crucial point: For any given sample, ascending the explanation tree simply enlarges the prediction set. Therefore, this action cannot lead to undercoverage, as no label, including a potential ground truth label, is removed. The significance of our contribution is that the exact classes in the tree to predict are determined via conformal prediction, so that the coverage can be guaranteed without needlessly predicting too many classes.
> > >
> > >
> > >
> > >
> > >
> > > As described in Section 2.3, Split Conformal Prediction only needs exchangeable data with test data for calibration and the definition of a non-conformity score that captures some notion of conformity and has sufficiently many distinct values, so that the Quantile in Eq. 3 can be clearly found. With this quantile, following proofs in [37] or more elaborate in [1], Eq. 3 returns prediction sets that guarantee the desired coverage (e.g. the ground truth will be included in 95% of exchangeable test data samples).
> > >
> > >
> > >
> > > Our contribution is to adapt Conformal Prediction to predict along the introduced hierarchical explanation via a novel non-conformity score (Eq. 12). This score relates to the distance in the hierarchy between top-1 prediction and each class and restricts the order of classes that are added to the prediction set to go up the tree. **By only introducing a different non-conformity score, our proposed method maintains all the guarantees that come with Conformal Prediction**, while enabling us to provide hierarchical local explanations that automatically highlight the coverage-guaranteeing predicted coherent set in the tree. Note that the non-conformity score returns different values for every sample as each is the sum of typically $n_{wc}=5$ real values, hence also ensuring that requirement is met.
> > >
> > >
> > >
> > > We will clarify that our work leverages the existing proven CP theory to provide guarantees and include a dedicated section in the appendix explaining why our conformal prediction method offers these guarantees. That section will include this explanation and the exchangeability points of the previous answer.
> > >
> > > [1] Lei, Jing, et al. "Distribution-free predictive inference for regression." Journal of the American Statistical Association 113.523 (2018): 1094-1111.

---

> > > ### Author Response · Authors · 2025-08-05
> > > **Response from Authors (2/2)**
> > >
> > > Regarding the new constraint in the QP: We agree that a formal analysis is an interesting direction and will add it as future work. However, our empirical analysis shows why a formal optimality guarantee for the QP is not as critical for our method's success.
> > >
> > > In line with QPM [35], we find that our method does not rely on finding the exact global optimum. In fact, it is at least as robust, showing no degrading performance when configuring our solver to terminate at 1% gap to optimality, as described in Section A. We attribute this robustness to the hierarchical constraint making the assignment more grounded (as measured by Structural Grounding), which in turn makes every feature be assigned to classes that share a less abstract concept. This forces the features to then change significantly more during fine-tuning to align with these clearer, shared concepts, ultimately achieving higher Feature Alignment.
> > >
> > > We have quantified this impact via an ablation study on CUB-2011, where we measured the average Pearson correlation coefficient between the activations of the same feature before and after fine-tuning across the entire training dataset while altering $\rho$.
> > >
> > > | $\rho$ | Avg. Pearson Correlation  |
> > > |--------|---------------------|
> > > | 0.0    | 0.825               |
> > > | 0.1    | 0.823               |
> > > | 0.3    | 0.805               |
> > > | 0.5    | 0.776               |
> > > | 0.7    | 0.743               |
> > >
> > > For reference, QPM has an average correlation of 0.855, indicating higher reliance on the most optimal choice of features. This evidence shows that the need for finding the exact optimal solution is reduced, as the features undergo significant changes regardless due to the more grounded assignment. We will add this discussion and experiment to the appendix, too.
> > >
> > > We again thank you for asking this critical question and hope that our answer resolves the primary concern.

---

> > > > ### Comment · Reviewer_kBM9 · 2025-08-07
> > > >
> > > > Thanks for the detailed comments by authors. I will update my score accordingly.

---

### Author Response · Authors · 2025-08-09
**Thank You for the Constructive Discussion**

As the discussion phase concludes, we would like to thank the reviewers for their time and detailed feedback. It has been very valuable and the resulting clarifications, discussions and additional experiments, which we will incorporate into the camera-ready version, will strengthen the paper.


We are particularly grateful that our responses were able to address the concerns of Reviewers kBM9 and 7M7N, and we appreciate their positive reassessment.


The discussion also allowed us to better articulate the novelty of our work, particularly the distinction of our dynamic, sample-specific hierarchies and their role in our interpretable conformal prediction framework.

---

### Decision · Program_Chairs · 2025-09-17

**Decision:**

Accept (poster)

**Comment:**

This paper received mixed reviews. After discussion, reviewers agreed to accept this paper.  The authors are encouraged to address the major concerns in the camera-ready version.
1. Add formal analysis for essential steps.
2. Improve the writing to make it easier to follow
3. Accuracy improvement over QPM is incremental.
4. Streamline notations.

The authors’ rebuttal and messages were carefully ready, discussed, and considered.